# Discovery of an allosteric binding site for anthraquinones at the human P2X4 receptor

Jessica Nagel [1], Vigneshwaran Namasivayam [1], Stephanie Weinhausen [1], Juan Sierra-Marquez [2], Younis Baqi [1,3], Hashem Ali M. Al Musawi [1], Aliaa Abdelrahman[1], Victoria J. Vaaßen [1], Jonathan G. Schlegel [1], Lisa Taplick[1], Jane Torp [4], Jan Kubicek[5], Barbara Maertens[5], Matthias Geyer [4], Tobias Claff [1], Annette Nicke [2], Gregor Hagelueken [4] ✉ & Christa E. Müller [1] ✉

P2X receptors are trimeric ATP-gated ion channels. The P2X4 receptor subtype is a promising drug target for the treatment of inflammatory diseases, neuropathic pain, and cancer. Here, the water-soluble anthraquinone derivative Cibacron Blue, previously described as a P2X4 receptor modulator, is selected as a lead structure, and structure-activity relationships are investigated. A chimeric receptor approach, combined with mutagenesis and docking studies, is applied to identify the allosteric binding site and the interacting amino acid residues. We discover that Glu307, located in the upper body of the receptor, is prone to form an intermolecular "ionic lock" with basic amino acid residues, thereby preventing high-affinity binding of anthraquinone derivatives. Exchange of Glu307 for threonine leads to a dramatic potency increase for anthraquinones in blocking P2X4 receptor function. The structure of the human P2X4-E307T receptor in complex with the anthraquinone derivative PSB-0704 is determined by cryo-electron microscopy at a resolution of 3.35 Å. This reveals an allosteric binding site in the upper body at the interface of the receptor trimer subunits, which differs from previously described allosteric sites on P2X receptors. Our results provide a rational basis for structure-based drug design towards potent and selective P2X4 receptor antagonists.

Adenosine 5′-triphosphate (ATP), the primary intracellular energy source, has emerged as a key extracellular signaling molecule, in particular in inflammatory conditions[1]. ATP-induced signaling cascades are mediated by ionotropic P2X and metabotropic P2Y receptors[2]. P2X receptors comprise seven subtypes, P2X1-P2X7, which are activated by ATP[2]. They form functional homotrimeric receptors, but heterotrimeric receptor complexes, e.g., P2X2/P2X3, have also been described[3,4]. Each protein subunit consists of a large extracellular domain harboring the agonist binding site and two transmembrane domains forming a cation pore that is permeable for $Na^+$, $K^+$, and $Ca^{2+}$ ions[5,6].

P2X receptors are widely expressed in neuronal and non-neuronal cells[7]. They contribute to a number of physiological and pathological processes, including synaptic transmission, smooth muscle contraction, pain sensation, and acute and chronic inflammation[7–9]. Thus, P2X receptors have emerged as attractive drug targets; in fact, the P2X3 receptor antagonist Gefapixant has recently been approved for the treatment of chronic cough[10,11].

[1]Pharmaceutical & Medicinal Chemistry, PharmaCenter Bonn, Pharmaceutical Institute, University of Bonn, Bonn, Germany. [2]Walther Straub Institute of Pharmacology and Toxicology, Faculty of Medicine, Ludwig-Maximilians-Universität München, Munich, Germany. [3]Department of Chemistry, College of Science, Sultan Qaboos University, Muscat, Oman. [4]Institute of Structural Biology, University of Bonn, Bonn, Germany. [5]Cube Biotech, Monheim, Germany. ✉e-mail: hagelueken@uni-bonn.de; christa.mueller@uni-bonn.de

The P2X4 receptor is mainly expressed in the central nervous system, particularly on neurons and microglia, and on peripheral immune cells, namely B- and T-lymphocytes, monocytes, neutrophils, and mast cells[12,13]. The inhibition of P2X4 receptors has shown promise in preclinical studies for various pathological conditions, including neuropathic pain, epilepsy, and further peripheral and central nervous system diseases associated with inflammation[14–18]. Moreover, several studies have demonstrated a role for P2X4 receptors in different types of cancer, e.g., in breast, prostate, and colon cancer proliferation and aggressiveness[19–21].

Thus, considerable efforts have been made to develop potent and selective P2X4 receptor antagonists; however, only a few compounds have been described so far[4]. These include the benzodiazepine derivatives 5-BDBD[4], NP-1815-PX[22], and MRS4719[23], the urea derivative BX430[24], its analog "9o"[25], and the sulfonamide derivative BAY-1797[26] (for structures, see Supplementary Fig. 1). NC-2600 (structure undisclosed) was the first P2X4 receptor antagonist to enter phase I clinical

trials for neuropathic pain, but no further development has been reported[4]. In the past years, our group has developed several series of allosteric P2X4 receptor antagonists, e.g., the indole derivatives PSB-15417 and PSB-OR-2020[27–29] (for structures see Supplementary Fig. 1). The anthraquinone derivative Cibacron Blue (Fig. 1a) was described as a non-selective allosteric modulator of P2X2, P2X3, and P2X4 receptors[30–32]. Anthraquinones are a major class of natural products and synthetic compounds used as therapeutic drugs, e.g., as laxatives, anti-inflammatory and antiarthritic drugs, and as anti-cancer agents[30–34]. However, the binding site of anthraquinones on P2X receptors has remained unknown.

The elucidation of high-resolution structures of different P2X receptor subtypes has significantly advanced our understanding of agonist and antagonist recognition, channel architecture and opening mechanisms, and processes involved in desensitization[35–51]. The first high-resolution P2X receptor structures were determined by X-ray crystallography of the zebrafish P2X4 receptor in its apo state[35], and its

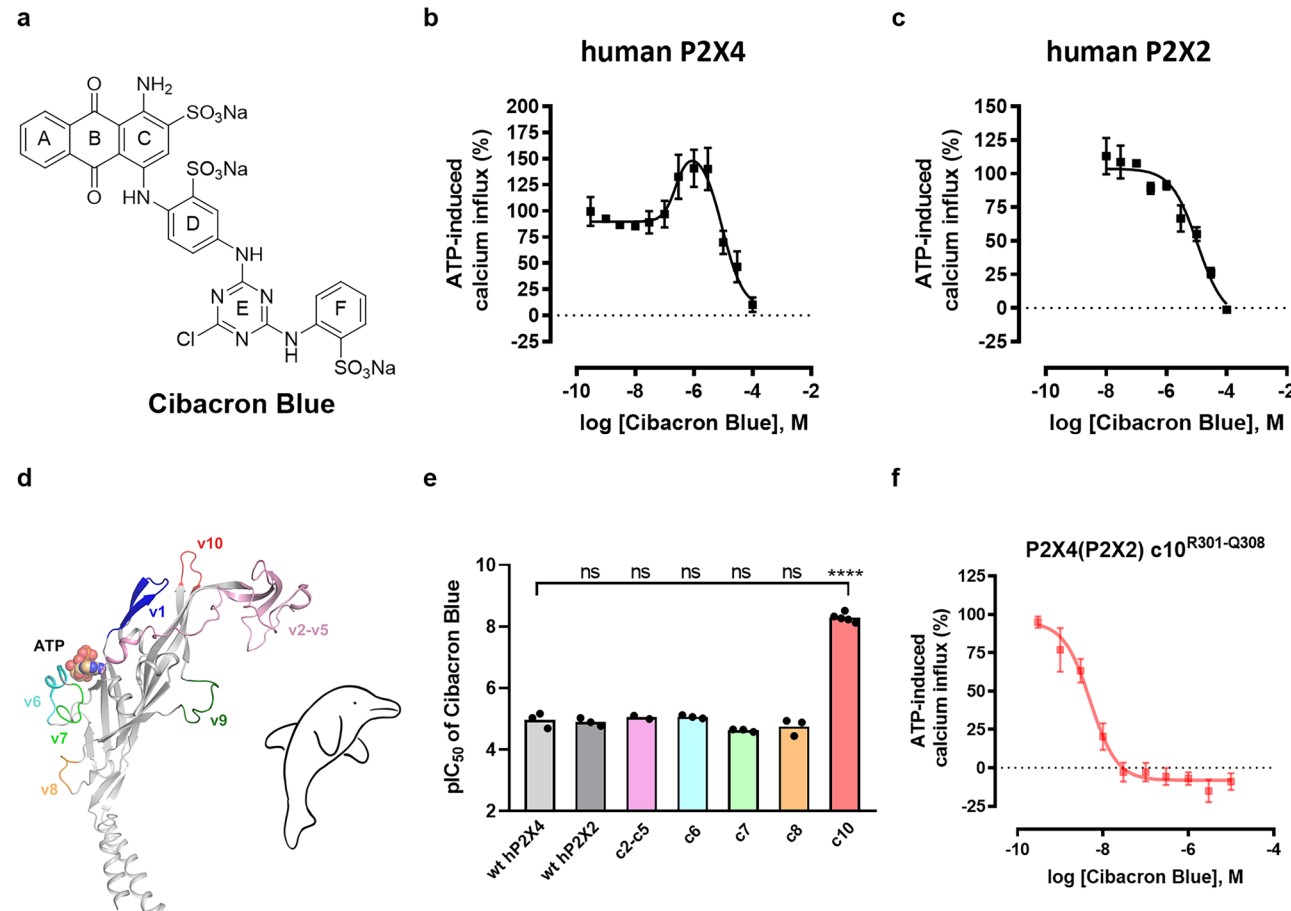

**Fig. 1 | Pharmacological characterization of Cibacron Blue at the human wt P2X4, the human wt P2X2, and chimeric P2X4(P2X2) receptors.** All receptors were stably expressed in human 1321N1 astrocytoma cells. Concentration-dependent inhibition was determined in the presence of the $EC_{80}$ of ATP. Data represent means ± SEM of at least three biological replicates, performed in technical duplicates. **a** Chemical structure of Cibacron Blue. **b** Biphasic modulation of the human wt P2X4 receptor by Cibacron Blue, determined by ATP-induced calcium influx assays (*n* = 4 independent experiments). Low concentrations of Cibacron Blue increased the ATP-induced activation of the human wt P2X4 receptor, while higher concentrations of Cibacron Blue inhibited it. **c** Concentration-dependent inhibition of the human wt P2X2 receptor by Cibacron Blue determined in ATP-induced calcium influx assays (*n* = 3 independent experiments). **d** Human P2X4 receptor subunit. Sequences that differ in the human P2X2 receptor (termed variable regions v1-v10) and the orthosteric ATP-binding site (shown as spheres) are

indicated. The figure was reproduced from Weinhausen et al.[56] based on PDB ID 4dw1[36]. Dolphin illustration was hand-drawn based on Hattori et al.[36] **e** $pIC_{50}$ values of Cibacron Blue at the human wt P2X4, the human wt P2X2, and the chimeric P2X4(P2X2) receptors c2-c5[C116-T186], c6[N208-S216], c7[I218-D224], c8[R265-L269], and c10[R301-Q308]. The chimeric receptor c1[V71-I83] showed only minor activation by ATP (maximal effect of 4% compared to that of the human wt P2X4 receptor)[56], and the effect of Cibacron Blue could therefore not be evaluated at c1. The level of significance was calculated by a one-way ANOVA with Dunnett's multiple comparisons test, with the human wt P2X4 receptor serving as the control group. Significance was expected if the *P* value was ≤0.05. The significance levels were defined as follows: *$P < 0.05$, **$P < 0.01$, ***$P < 0.001$, ****$P < 0.0001$, ns not significant. **f** Concentration-dependent inhibition of the chimeric P2X4(P2X2) receptor c10[R301-Q308] mutant by Cibacron Blue (*n* = 5 independent experiments). Source data are provided as a Source data file.

ATP-bound state[36]. The results confirmed the previously proposed trimeric structure of P2X receptors[52,53], with three extracellular ATP-binding sites, one on each receptor monomer. The receptor subunits resemble a leaping dolphin, where the transmembrane helices correspond to the flukes and the extracellular region to the upper body[35,36]. Recently published cryo-electron microscopy (cryo-EM) structures of the zebrafish P2X4 receptor in complex with the negative allosteric modulators BX430 and BAY-1797 identified a common binding site for both antagonists at the top of the extracellular domain near the subunit interfaces[38]. However, significant P2X4 receptor species differences are known to exist[54], and very recently, the first cryo-EM structure of the human P2X4 receptor bound to BAY-1797 was published[51]. Further high-resolution structures of P2X receptors, either determined by cryo-EM or X-ray crystallography, include those of the human P2X3 receptor, the panda, chicken, and rat P2X7 receptor, and the human P2X1 receptor (see Supplementary Table 1)[39–44,46–51,55]. Each of the determined structures has provided important and often unpredictable insights.

The present study reports the structure of the human P2X4 receptor, containing an E307T mutation, in complex with an allosteric anthraquinone-derived P2X4 receptor antagonist, elucidated by cryo-EM. We identify an allosteric binding site for anthraquinone derivatives, which differs from previously described allosteric sites. Our findings are based on initial computational and mutagenesis studies, supported by calcium influx and radioligand binding assays, in addition to two-electrode voltage clamp (TEVC) studies on wild-type (wt) and mutant P2X4 receptors expressed in *Xenopus laevis* oocytes. These insights advance our molecular understanding of P2X receptors in general and of the human P2X4 receptor in particular. They are expected to provide a basis for the development of future drugs.

## Results

### Receptor chimeras, docking, and mutagenesis

In our efforts aimed at the development and characterization of P2X4 receptor antagonists, we determined concentration-dependent effects of the anthraquinone derivative Cibacron Blue on ATP-induced calcium influx in 1321N1 astrocytoma cells stably transfected with the human wt P2X4 receptor. Cibacron Blue displayed a biphasic modulation of the P2X4 receptor, enhancing the ATP effect at low concentrations (up to 141%; $EC_{50}$ 0.243 μM) while inhibiting it completely at higher concentrations ($IC_{50}$ 12.1 μM) (Fig. 1b). At the human P2X2 receptor, only an inhibitory effect was observed ($IC_{50}$ 13.1 μM) (Fig. 1c). To further study the different behavior of the allosteric modulator Cibacron Blue at P2X4 as compared to P2X2 receptors, and directed at the identification of the allosteric binding site for the anthraquinone derivative, we investigated the compound at a set of previously developed P2X4/P2X2 receptor chimeras[56]. Ten variable regions (v1-v10) of the human P2X4 receptor, showing large differences to those of the P2X2 receptor, were selected based on a sequence alignment of both receptors (Fig. 1d, Supplementary Fig. 2)[56]. This led to the design of chimeric (c) human P2X4 receptors in which specific variable sequences were exchanged for the corresponding sequences of the human P2X2 receptor[56], designated c1[V71-I83], c2-c5[C116-T186], c6[N208-S216], c7[I218-D224], c8[R265-L269], c9[T281-P290], and c10[R301-Q308]. Except for chimeric receptors c1[V71-I83] and c9[T218-P290], all chimeric receptors could be activated by ATP (Supplementary Fig. 3 and Supplementary Table 2).

Pharmacological characterization of the chimeras led to an unexpected discovery. The inhibitory potency of Cibacron Blue at the chimeric P2X4(P2X2) receptor c10[R301-Q308] was dramatically increased as compared to the wt P2X4 receptor, from an $IC_{50}$ value of 12.1 μM to an $IC_{50}$ value of 4.76 nM (2,500-fold potency increase; Fig. 1e, f). In this chimeric receptor, the amino acids ranging from Arg301 to Gln308 of the human P2X4 receptor had been exchanged for the corresponding amino acids present in the human P2X2 receptor (Fig. 2). The $EC_{50}$

value of the endogenous agonist ATP was found to be similar at the chimeric P2X4(P2X2) receptor c10[R301-Q308] (0.862 μM) as compared to the wt P2X4 receptor (0.357 μM, Supplementary Fig. 3, Supplementary Table 2). At chimeric P2X4(P2X2) receptors c2-c5[C116-T186], c7[I218-D224], and c8[R265-L269], Cibacron Blue showed inhibitory potency in the micromolar range (Supplementary Table 3 and Supplementary Figs. 4 and 5). The antagonist displayed a biphasic modulation of the chimeric P2X4(P2X2) receptor c6[N208-S216], similar to the modulation observed at the P2X4 receptor (Supplementary Table 3 and Supplementary Fig. 5), while it showed monophasic inhibition at the other investigated chimeric receptors.

To understand the observed effect, an AlphaFold 3 (AF3)[57] model of the apo human P2X4 receptor was built, which predicted that Glu307 of one receptor subunit forms ionic interactions with both Arg82 and Lys298 of the adjacent subunit (Fig. 2a, left). These predicted interactions were recently confirmed by the cryo-EM structure of the human P2X4 receptor[51]. Arg82 is located in variable region v1 of the P2X4 receptor (Fig. 1d), while Arg301 and Glu307 are located in variable region v10 (Figs. 1d and 2b). Based on the AF3 model, electrostatic interactions between Glu307 and basic amino acid residues Arg82, as well as Lys298 of an adjacent protein subunit, result in the formation of an intra- and intermolecular "ionic lock," which is likely responsible for the moderate potency of Cibacron Blue and related anthraquinone derivatives at the wt P2X4 receptor.

The high potency of Cibracon Blue at the chimeric P2X4(P2X2) receptor c10[R301-Q308] may result from the replacement of the acidic residue Glu307 in variable region v10 of the P2X4 receptor (Fig. 2b). Exchange of glutamate for threonine in position 307 prevents the putative ionic interaction and would make Arg82 available for direct ligand interaction, which we predicted to be the reason for the 2,500-fold increased potency of Cibacron Blue observed at the chimeric receptor P2X4(P2X2) c10[R301-Q308].

To provide further evidence for the proposed binding interactions, P2X4 receptor mutants were generated and characterized, harboring mutations in v10 (E307T, D302I, and Q308T). The D302I and Q308T mutations essentially served as controls. All mutants could be activated by ATP, and $EC_{50}$ values were similar to those determined at the human wt P2X4 receptor (Supplementary Table 2 and Supplementary Fig. 4). For the P2X4-E307T receptor mutant, Cibacron Blue showed a significantly higher inhibitory potency than for the wt P2X4 receptor (680-fold; $IC_{50}$ 0.0178 μM (P2X4-E307T); $IC_{50}$ 12.1 μM (wt P2X4)) (Fig. 3a, b, Supplementary Table 3), supporting our hypothesis that "ionic lock" formation, i.e., the formation of a salt bridge involving Glu307 and basic amino acid residues, prevents high-affinity binding to the wt P2X4 receptor. The inhibitory potency of Cibacron Blue at the P2X4-D302I receptor was moderately increased by about 9-fold compared to the wt P2X4 receptor (Fig. 3a, Supplementary Table 3). At the P2X4-Q308T mutant, the potency of Cibacron Blue was similar to that determined at the wt P2X4 receptor (Fig. 3a, Supplementary Table 3).

These results support our hypothesis that the amino acid residue Thr307, instead of glutamate at that position, is particularly important for high-affinity binding of Cibacron Blue, while mutation of the other selected residues did not (Q308T) or only moderately contribute (D302I). It has to be mentioned that D302 in the P2X4 receptor does probably not align with Ile in the P2X2 receptor due to a gap in the sequence (see Fig. 2b).

To validate these results and to identify a potential binding mode, further anthraquinone derivatives were tested at the wt P2X4 receptor, the P2X4-E307T receptor mutant, and the chimeric P2X4(P2X2) c10[R301-Q308] receptor (Table 1). Structure-activity relationship (SAR) studies showed that at least one acidic function is required, as PSB-25041, a neutral molecule lacking any acidic group, is inactive. For all active compounds, a large potency increase is observed at the P2X4-E307T mutant and also at the chimeric P2X4(P2X2) receptor c10 (as far as investigated), in comparison to the wt P2X4 receptor. Besides Cibacron

**a**

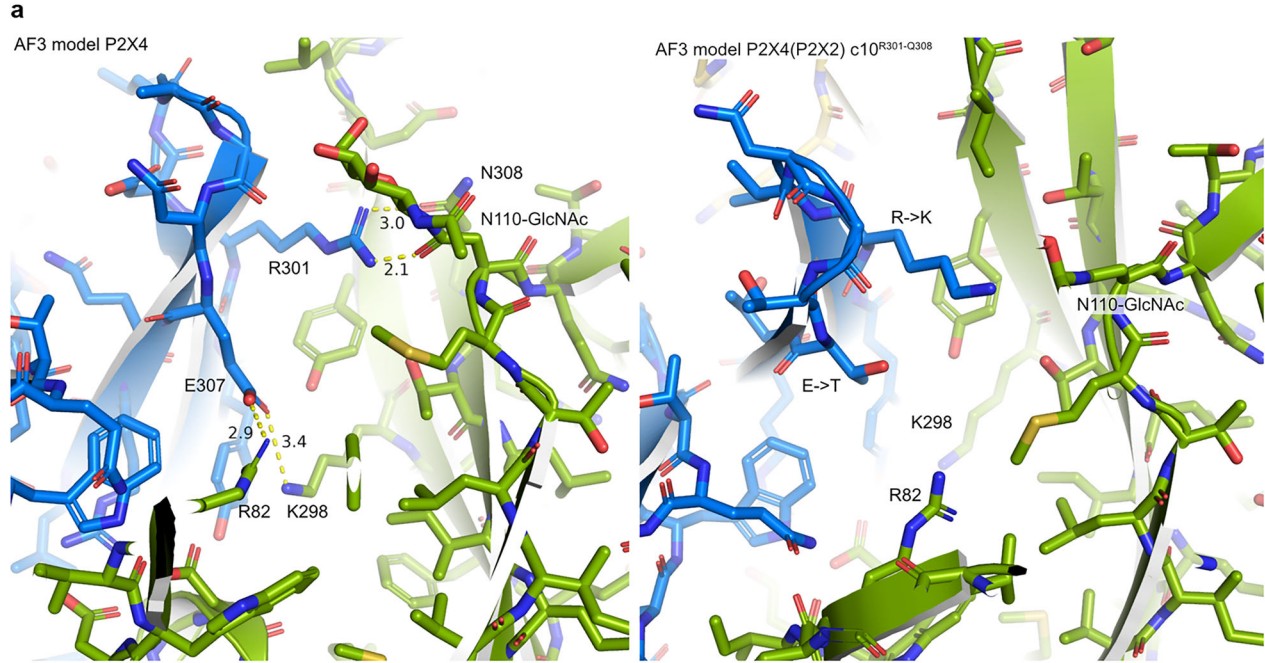

**b**

```
hP2X4_gi-116242696   IGWVFVWEKGYQETDSVV-SSVTTKVKGVAVTNTSKLGFRIWDVADYVIPAQEENSLFVM   102
hP2X2_gi-25092719    VWYVFIVQKSYQESETGPESSIITKVKGITT-----SEHKVWDVEEYVKPPEGGSVFSII   111
                     : :**: :*.***::.   **: *****::.        .::*** :** * : . : ::

hP2X4_gi-116242696   TNVILTMNQTQGLCPEIPD-ATTVCKSDASCTAGSAGTHSNGVSTGRCVAF-NGSVKTCE   160
hP2X2_gi-25092719    TRVEATHSQTQGTCPESIRVHNATCLSDADCVAGELDMLGNGLRTGRCVPYYQGPSKTCE   171
                     *.*  * .**** ***    .::* ***.*.**. .  .**: ***** : :* ****

hP2X4_gi-116242696   VAAWCPVEDDTHVPQPAFLKAAENFTLLVKNNIWYPKFNFSKRNILPNITTTYLKSCIYD   220
hP2X2_gi-25092719    VFGWCPVEDGASVSQF-LGTMAPNFTILIKNSIHYPKFHFSKGNIA-DRTDGYLKRCTFH   229
                     * .******.: * *  : . * ***:*:**.* ****:*** **  : *  *** * :.

hP2X4_gi-116242696   AKTDPFCPIFRLGKIVENAGHSFQDMAVEGGIMGIQVNWDCNLDRAASLCLPRYSFRRLD   280
hP2X2_gi-25092719    EASDLYCPIFKLGFIVEKAGESFTELAHKGGVIGVIINWDCDLDLPASECNPKYSFRRLD   289
                     :* :*****:** ***:**.** :* :**::*: :****:** ** * *:*******

                                          |--v10-|
hP2X4_gi-116242696   TRDVEHNVSPGYNFRFAKYYRDLAGNEQRTLIKAYGIRFDIIVFGKAGKFDIIPTMINIG   340
hP2X2_gi-25092719    PKHVP--ASSGYNFRFAKYYK-INGTTTRTLIKAYGIRIDVIVHGQAGKFSLIPTIINLA   346
                     :.*   .* **********: : *. **********:*:**.*:****.:***:**:.
```

| | variable region v10 | Cibacron Blue IC$_{50}$ (μM) |
|---|---|---|
| human wt P2X4 | R̲DLAG̲NE̲Q<br>301 307 | 12.1 |
| chimeric P2X4(P2X2) c10$^{R301-Q308}$ | K̲-INGT̲T̲T | 0.00476 |

**Fig. 2 | Models of native human and chimeric P2X4 receptor and sequence comparison. a** AlphaFold 3 (AF3) model of the trimeric apo human P2X4 receptor (left) and of the chimeric P2X4(P2X2) c10$^{R301-Q308}$ receptor (right). The three chains of the trimer are color-coded in blue, green, and gold. In the wild-type receptor (left), Glu307 is predicted to form ionic interactions with Arg82 and Lys298. Arg301 may interact with the N-glycosylated Asn110. The numbers are distances in Å. **b** Sequence alignment of the P2X4 and P2X2 receptor. Top: sequence alignment of human wt P2X4 (gi-116242696) and the wt P2X2 receptor (gi-25092719) using Clustal Omega (1.2.4). Variable region v10 is colored in red. Amino acid residues predicted to be important for Cibacron Blue binding are underlined. Symbols: (*) indicates conserved residues, (:) and (.) indicate conservative and semi-conservative exchanges, and blank spaces point out non-conservative exchanges in residues between aligned sequences. Bottom: comparison of amino acid residues in variable region v10 (red) in the human wt P2X4 and the chimeric P2X4(P2X2) receptor c10$^{R301-Q308}$. Amino acid residues predicted to be important for Cibacron Blue binding are underlined. Inhibitory potency of Cibacron Blue (IC$_{50}$ (μM)) at the human wt P2X4 receptor and the chimeric P2X4(P2X2) receptor c10$^{R301-Q308}$, determined in calcium influx assays.

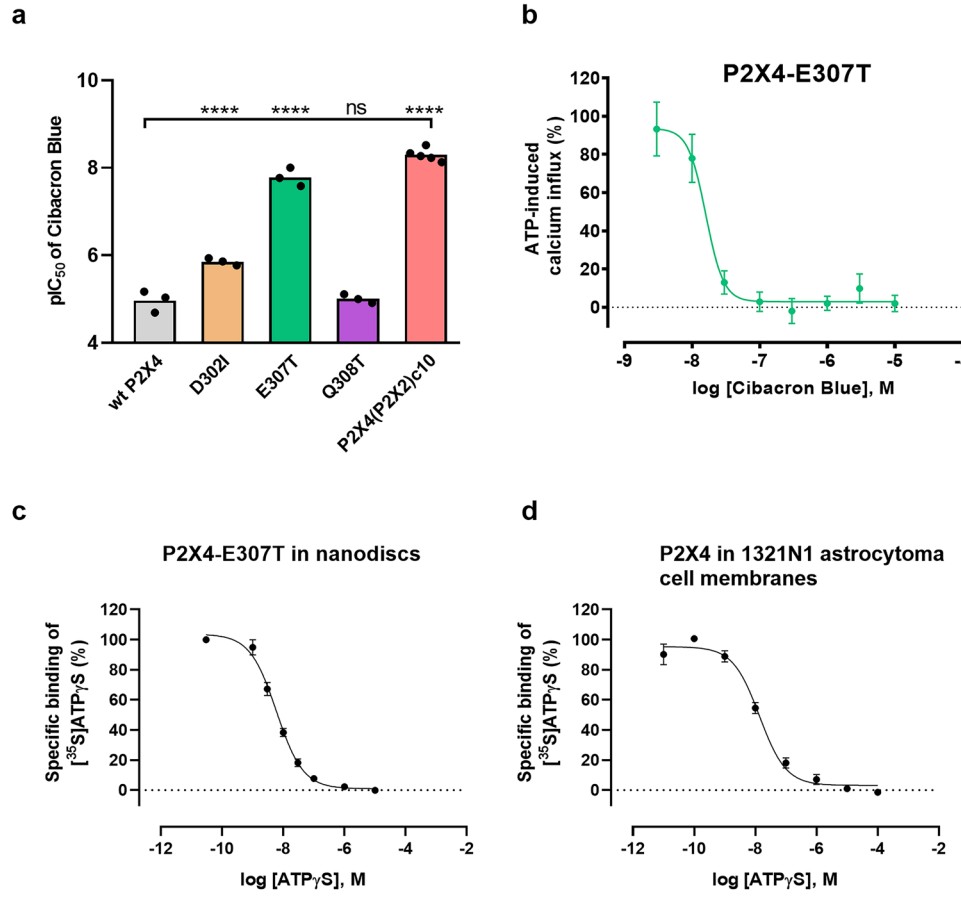

**Fig. 3 | Potency of Cibacron Blue at P2X4 receptor mutants and chimeric P2X4(P2X2) receptor c10$^{R301-Q308}$ and characterization of the P2X4-E307T receptor mutant in comparison to the wt receptor by radioligand binding assays. a** pIC$_{50}$ values of Cibacron Blue in blocking the human wt P2X4 receptor, the chimeric P2X4(P2X2) receptor c10$^{R301-Q308}$, and several P2X4 receptor mutants in the presence of an EC$_{80}$ of ATP (1–3 µM). pIC$_{50}$ values were determined in fluorimetric calcium influx assays induced by ATP (at its EC$_{80}$). Single data points from 3–5 biological replicates, performed in technical duplicates, are shown. (wt P2X4: $n = 3$; D302I: $n = 3$; E307T: $n = 3$; Q308T: $n = 3$; P2X4(P2X2)c10: $n = 5$ independent experiments). The level of significance was calculated by a one-way ANOVA with Dunnett's multiple comparisons test, with the human wt P2X4 receptor serving as the control group. Significance was expected if the $P$ value was ≤0.05. The significance levels were defined as follows: *$P < 0.05$, **$P < 0.01$, ***$P < 0.001$, ****$P < 0.0001$; ns not significant. **b** Concentration-dependent inhibition by

Cibacron Blue of ATP-induced activation of the human P2X4-E307T receptor mutant determined in the presence of an EC$_{80}$ of ATP. Data represent means ± SEM of $n = 3$ biological replicates, performed in technical duplicates. **c** Homologous competition binding experiments of a range of ATPγS concentrations versus 0.2 nM of [$^{35}$S]ATPγS (1250 Ci/mmol; $4.63 \times 10^{13}$ Bq/mmol) using the human solubilized and purified P2X4-E307T receptor mutant reconstituted into nanodiscs. Radioligand binding was determined in the presence of various concentrations of unlabeled ATPγS in different fractions collected during gel filtration (1:25 dilutions) of the P2X4 receptor reconstituted into nanodiscs. Data represent means ± SEM of $n = 3$ biological replicates. **d** Homologous competition of ATPγS vs. [$^{35}$S]ATPγS (0.2 nM) at the human wt P2X4 receptor recombinantly expressed in 1321N1 astrocytoma cells (15 µg of protein). Data represent means ± SEM of $n = 3$ biological replicates. Source data are provided as a Source data file.

Blue, the most potent compounds of the present series at the mutant receptors are PSB-0739, bearing two sulfonate groups, one in position 2 (ring C), and one on ring D, and its derivative lacking the amino group in the 1-position (PSB-24039). Sulfonate or carboxylate groups at ring C and D are similarly well tolerated (Table 1). SARs are somewhat different at the wt P2X4 receptor as compared to its E307T mutant. The most potent antagonist at the wt P2X4 receptor is PSB-0826, bearing one sulfonate group (IC$_{50}$ 1.35 µM). The smaller compounds PSB-0711 and PSB-25012, lacking ring E, are more potent at the wt P2X4 receptor than many of the larger derivatives, but less potent at the mutant receptor. The quite strong interaction of PSB-0826 and PSB-25012 with the wild-type receptor may be explained by the absence of a negatively charged substituent on ring D. This would allow an interaction with the same balance of charges and the ionic lock still being partially intact. In contrast to Cibacron Blue, none of the other investigated anthraquinone derivatives showed a biphasic curve at the wt P2X4 receptor in calcium influx assays under the applied conditions.

Thus, we had strong indications for an allosteric binding site of the P2X4 receptor that had not been previously described. To obtain unambiguous evidence, we set out to determine a cryo-EM structure.

## P2X4 receptor protein expression and purification

A construct for P2X4 receptor expression was designed that would enable cryo-EM studies. We chose the human P2X4 receptor sequence containing a C-terminal polyhistidine tag (10x-His tag) and three point mutations to remove potential glycosylation sites (N75R and N184R)[35,36] and to increase the receptor's potency for anthraquinone derivatives (E307T). The glycosylation sites were removed since we initially considered X-ray crystallography, which requires high protein homogeneity. The receptor was overexpressed in *Spodoptera frugiperda (Sf9)* insect cells, solubilized with dodecyl-β-ᴅ-maltoside (DDM) and cholesteryl hemisuccinate (CHS), and purified by immobilized metal affinity chromatography (IMAC). The purified protein was reconstituted into short and flexible amphipathic polymers that

**Table 1 | Potency of selected anthraquinone derivatives determined by measurement of calcium influx at the human wt P2X4 receptor, the P2X4-E307T receptor mutant, and the chimeric receptor P2X4(P2X2) c10$^{R301-Q308}$, stably expressed in 1321N1 astrocytoma cells**

| Compound | Structure | IC$_{50}$ ± SEM (µM)[a] | | |
|---|---|---|---|---|
| | | wt P2X4 | P2X4-E307T | P2X4(P2X2) c10$^{R301-Q308}$ |
| Cibacron Blue | | 12.1 ± 4.2 | 0.0178 ± 0.0047 | 0.00476 ± 0.00062 |
| PSB-0704 | | 12.5 ± 1.9 | 0.0724 ± 0.0102 | n.d.[b] |
| PSB-0739 | | 8.91 ± 0.67 | 0.0353 ± 0.0047 | 0.00341 ± 0.00049 |
| PSB-24039 | | 11.4 ± 0.3 | 0.0404 ± 0.0090 | n.d.[b] |
| PSB-0826 | | 1.35 ± 0.22 | 0.164 ± 0.042 | 0.0922 ± 0.021 |
| PSB-25041 | | >10 | >10 | >10 |
| PSB-0801 | | 25.3 ± 0.8 | 0.180 ± 0.007 | 0.0429 ± 0.0033 |
| PSB-0711 | | 4.33 ± 1.29 | 0.289 ± 0.044 | 0.0619 ± 0.0103 |
| PSB-25012 | | 3.28 ± 0.80 | 0.204 ± 0.027 | 0.0448 ± 0.0073 |
| PSB-09059 | | 20.0 ± 8.45 | 0.195 ± 0.014 | 0.266 ± 0.018 |
| BX430 | | 0.127 ± 0.045[c] | 0.0249 ± 0.0054 | n.d. |
| BAY-1797 | | 0.108 [d] | 0.0102 ± 0.0021 | n.d. |
| 5-BDBD | | 0.990 ± 0.112[e] | 0.901 ± 0.131 | n.d. |

**Table 1 (continued) | Potency of selected anthraquinone derivatives determined by measurement of calcium influx at the human wt P2X4 receptor, the P2X4-E307T receptor mutant, and the chimeric receptor P2X4(P2X2) c1OR3O1-Q3O8, stably expressed in 1321N1 astrocytoma cells**

| Compound | Structure | IC$_{50}$ ± SEM (µM)$^a$ | | |
|---|---|---|---|---|
| | | wt P2X4 | P2X4-E307T | P2X4(P2X2) c1O$^{R3O1-Q3O8}$ |
| | | | | |

$^a$Data represent means ± SEM of at least three biological replicates, performed in technical duplicates. The inhibition of anthraquinone derivatives was determined in the presence of ATP at its respective EC$_{80}$.
$^b$n.d. not determined.
$^c$Published value[56].
$^d$Published value[26].
$^e$Published value[54].

stabilize membrane proteins (A8-35 amphipol)[58,59], and subsequently analyzed through single-particle cryo-EM.

To characterize the human P2X4 receptor construct used for cryo-EM studies, the solubilized fraction of the same receptor construct, but with an additional N-terminal green fluorescent protein (GFP) fusion attached for visualization, and a C-terminal Rho1D4 tag, was reconstituted into nanodiscs and evaluated by radioligand binding studies using the agonist radioligand [$^{35}$S]ATPγS (Fig. 3c), an ATP analog with increased stability[30,60]. A concentration-dependent inhibition of [$^{35}$S]ATPγS binding by the unlabeled agonist ATPγS was observed, indicating high affinity of the ligand to the P2X4 receptor reconstituted into nanodiscs, with a dissociation constant ($K_D$ value) of 6.01 ± 0.81 nM (Fig. 3c). A maximum binding capacity ($B_{max}$ value) of 1056 ± 110 pmol/mg of protein was calculated, indicating a very high level of conformationally intact P2X4 receptors. For comparison, we performed homologous competition binding experiments using 1321N1 astrocytoma cell membranes stably expressing the human wt P2X4 receptor (Fig. 3d). A $K_D$ value of 14.1 ± 2.7 nM was determined, which is in the same range as the value determined for the purified, reconstituted P2X4-E307T receptor construct. Our results show that a high receptor density can be achieved by P2X4 receptor reconstitution into nanodiscs while retaining its bioactive conformation.

For the determination of a cryo-EM structure, we utilized amphipol A8-35 instead of nanodiscs. Both approaches can be expected to provide a similar environment, and A8-35 has been successfully employed for structure determination of ion channels[59,61]. In contrast to nanodiscs, reconstitution using A8-35 does not require membrane scaffold proteins, which simplifies the procedure[59,62].

**Anthraquinone derivative PSB-0704 selected for structure determination**

For structural studies, the potent and water-soluble anthraquinone derivative PSB-0704 was selected (Table 1, Fig. 4a). This compound is more drug-like than Cibacron Blue due to its smaller molecular weight and the exchange of sulfonate groups for carboxylate functions. PSB-0704 (Fig. 4a) showed a concentration-dependent inhibition of the human wt P2X4 receptor stably expressed in 1321N1 astrocytoma cells, with an IC$_{50}$ value of 12.5 µM, similar to that of Cibacron Blue (Figs. 1b and 4b). At the P2X4-E307T receptor mutant, the potency of PSB-0704 was also strongly increased (>170-fold, IC$_{50}$ 0.0724 µM, Fig. 4b).

**Cryo-EM reconstruction of the human P2X4 receptor**

A cryo-EM dataset (Supplementary Table 4) of the P2X4(E307T)-PSB-0704-complex reconstituted into amphipols, was collected and processed as shown in Supplementary Fig. 6. Initial processing clearly revealed the characteristic C3-symmetric shape of a P2X receptor (Supplementary Fig. 6a). However, the reconstructed map was difficult to interpret due to preferred orientations of the particles with a strong overrepresentation of views along the threefold symmetry axis (Supplementary Fig. 6b–d).

We reasoned that rare views might have been sorted out during the 2D classification procedure, and therefore decided to skip the 2D classification step. Instead, all 3.6 million extracted particles were fed into three iterative heterogeneous refinement steps using the 3D reconstruction of the P2X4 receptor and five pseudo-random reconstructions. In each step, the particles that were assorted to any of the pseudo-random reconstructions were discarded (Supplementary Fig. 6b). This procedure resulted in 183,957 particles that were further processed in two branches of the processing tree (Supplementary Fig. 6b). The overall branch led to a 3.67 Å C3-symmetric structure of residues 42–344 of the receptor. In the focused branch, further 3D classifications without particle alignment and local refinements were used to reconstruct the extracellular domain of the receptor at an overall resolution of 3.35 Å. Skipping the 2D classification and using multiple cycles of heterogeneous refinements led to a more balanced distribution of viewing directions than obtained initially (Supplementary Fig. 6e–g).

An AF3[57] model of the human P2X4 receptor was placed into the 3D volume, and ISOLDE[63] was used to fit the AF3 model to the experimental map. A significant portion of the transmembrane helices was visible in the overall reconstruction (Fig. 5, Supplementary Fig. 7). The final structure comprises residues 42–344 of the human P2X4 receptor (Fig. 6a). The overall architecture of our model reflects the typical trimeric arrangement of P2X receptors with a large extracellular domain and two transmembrane (TM) helices in each subunit. The interaction between the monomers is characterized by very large interfaces with a combined buried surface area of ~3500 Å$^2$ between each pair of monomers and distinct charge complementarity (Fig. 6b). Each monomer resembles the shape of a dolphin, consistent with previously reported P2X receptor structures[36,51] (Fig. 6).

Our reconstruction revealed the known N-glycosylation sites at positions Asn110 (upper body) and Asn208 (dorsal fin). The reconstruction was of sufficient quality to add one, respectively two N-acetylglucosamine units at these positions (Fig. 6). Additional density was also present at Asn153 in the head domain, but it was not well enough defined to build a model of the glycosylated asparagine side chain. Further N-glycosylations are known to occur at residues Asn75 and Asn184, but, as mentioned above, these residues were mutated to arginine in our expression construct[35].

We used the Foldseek Search Server[64] and manual PDB searches to compare our structural model to known structures of P2X receptors. Supplementary Fig. 8a–c show superpositions of our model with the closed and open state zebrafish P2X4 receptor ortholog (PDB ID: 4dw0, 4dw1), as well as the recently published human P2X4 receptor in the apo closed state and in the BAY-1797-bound form, which is overall very similar (PDB IDs: 9bqh, 9bqi)[51]. Our model most closely resembles the closed-state apo forms of the receptor, especially the central parts of the structures at the transition between the extracellular and the

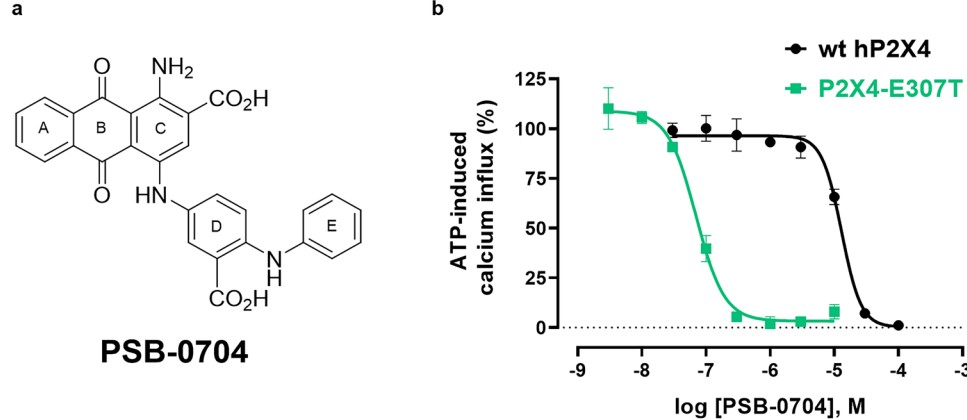

**Fig. 4 | Characterization of the anthraquinone derivative PSB-0704 used for cryo-EM. a** Chemical structure of PSB-0704 with the designation of aromatic rings (A–E). **b** Concentration-dependent inhibition of ATP-induced calcium influx by PSB-0704 at the wild-type (wt) human (h) P2X4 receptor (black) and the human P2X4-E307T receptor mutant (green), recombinantly expressed in 1321N1 astrocytoma cells, determined versus the $EC_{80}$ of ATP. Data represent means $\pm$ SEM of $n = 3$ biological replicates, performed in technical duplicates. Source data are provided as a Source data file.

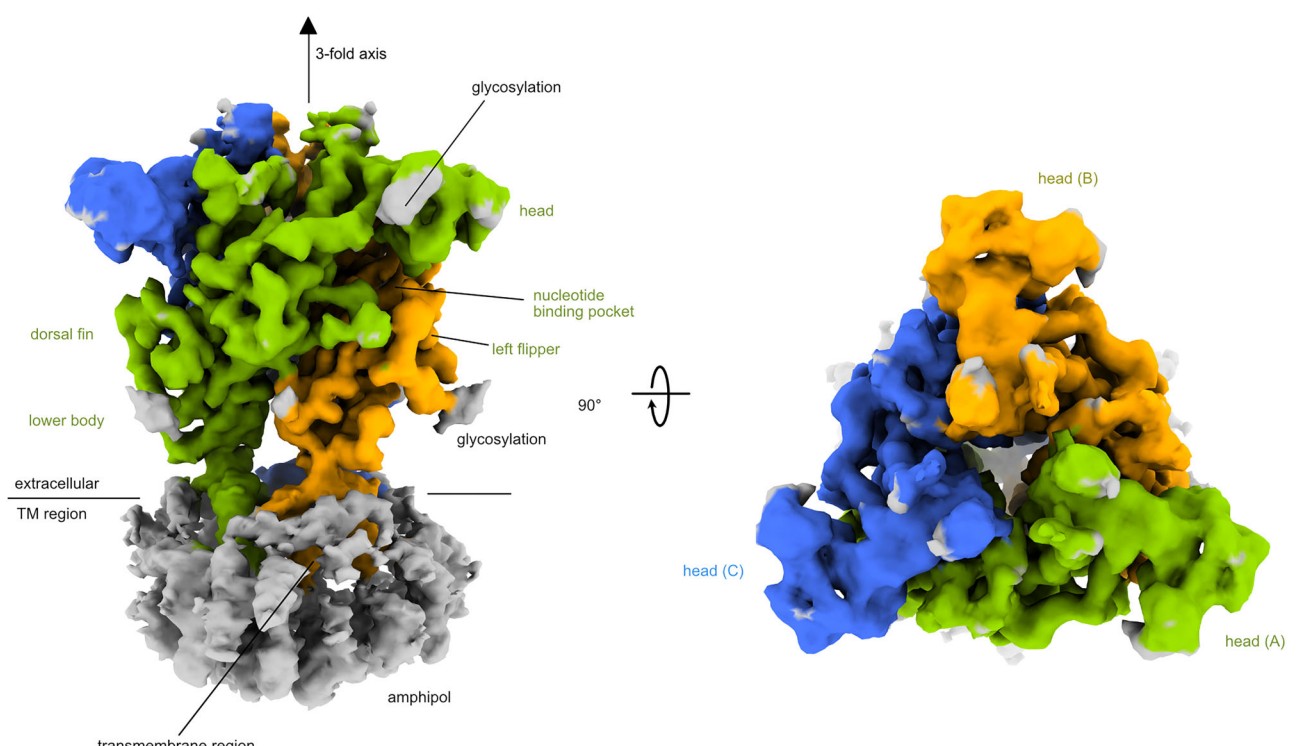

**Fig. 5 | Structure of the human P2X4 receptor.** Cryo-EM reconstruction of the human P2X4 receptor. The three protein chains of the receptor are colored in green, yellow, and blue. The C3 symmetry axis is indicated. The transmembrane region is colored gray. Side view, perpendicular to the cell membrane (left); top view from the extracellular side (right).

membrane domain (i.e., the lower body of the dolphin) align well (Supplementary Fig. 8a, c). Pairwise structural alignments (based on one chain of each model) resulted in an r.m.s.d. of 1.5 Å between 242 Cα atoms for the zebrafish apo closed-state structure, an r.m.s.d. of 1.3 Å between 242 Cα atoms for the human closed-state apo structure (9bqh), and an r.m.s.d. of 0.9 Å between 228 Cα atoms for the BAY-1797-bound structure of the human P2X4 receptor (9bqi). The r.m.s.d. between our structure and the ligand-bound open state of the zebrafish receptor was higher with 2.1 Å between 262 Cα atoms (Supplementary Fig. 8b). Since ATP was not added to our cryo-EM samples and was also not visible in our maps, we conclude that our structure represents the closed state of the receptor without ATP bound (Supplementary Fig. 8d). Furthermore, we noticed a concerted downward

movement (i.e., towards the membrane plane) of the head domain in our structure, compared to the zebrafish structures, but also compared to the human P2X4 structures (Supplementary Fig. 8a–c).

While building the structural model, we identified a large feature at the trimer subunit interface, at the back of the head domain. This feature was clearly visible in both the overall and focused reconstructions. The antagonist PSB-0704, which had been added during protein purification and again immediately before the cryo-EM grid preparation, was placed into this map feature by hand and refined with ISOLDE[63] (Fig. 7). Presumably due to the presence of the ligand, the β-hairpin loop containing R82 is partly disordered and bent backwards compared to the AF3 model of the human P2X4 receptor without a bound ligand, and the bound ligand is most likely also the reason

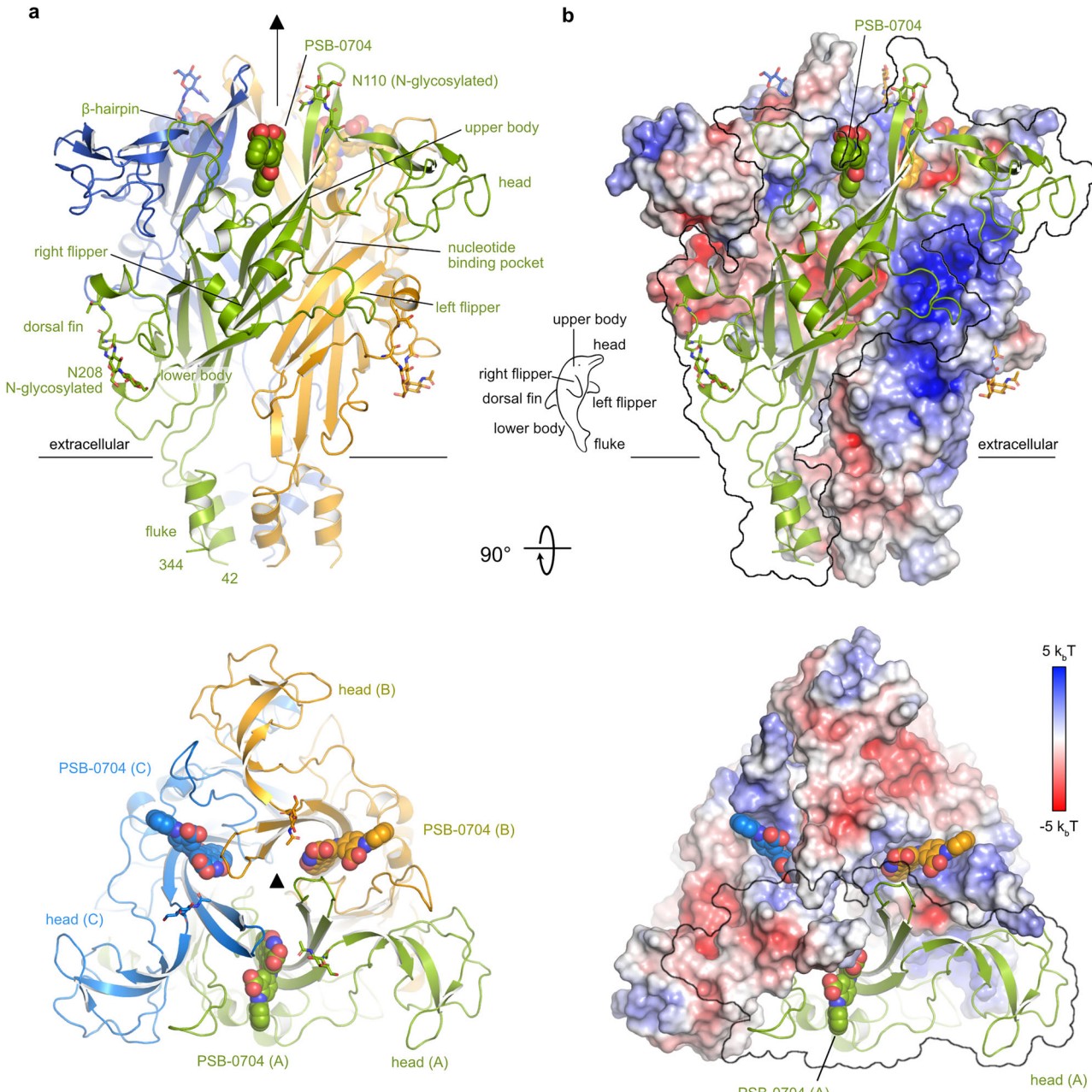

**Fig. 6 | Cryo-EM structure of the P2X4 receptor showing the allosteric binding sites. a** Cartoon model of the human P2X4 receptor. The antagonist PSB-0704 is shown as spheres. The C3 symmetry axis is indicated. N-glycosylations are shown as sticks. **b** Charge complementarity (calculated with the Adaptive Poisson-Boltzmann Solver (APBS[88])) of monomer interfaces in the extracellular domain stabilizes the trimeric receptor. The surface of one monomer is shown as a transparent outline to visualize the surface potential in the interface areas.

for the above-described downward movement of the head domains (Fig. 6a).

The antagonist PSB-0704 engages in many hydrophobic interactions of the aromatic rings A–E with the receptor and by ionic interactions between the two carboxylate groups in rings C and D with side chains Arg301 and Arg82, respectively (Fig. 7). Arg301 forms a bidentate salt bridge with the carboxylate of ring C, whereas Arg82 interacts with the carboxylate of ring D and additionally forms a cation-π stacking interaction with rings B and C (Fig. 7). The amino and carbonyl oxygen substituents of ring B and C are not in close contact to any polar groups of the receptor. Note that due to the location of the binding site at the interface between two monomers, the two arginine side chains come from two adjacent chains of the trimeric receptor (Fig. 7).

Mutagenesis studies had shown a significantly increased potency of Cibacron Blue and related anthraquinone derivatives, including PSB-0704, at the P2X4-E307T receptor mutant (Figs. 3a, b and 4b, Table 1). An AF3 model of the receptor together with our structure can rationalize this observation. Supplementary Fig. 8f shows a model of PSB-0704 binding to the human wt P2X4 receptor, based on the human P2X4-E307T receptor structure from this work, in which the threonine at position 307 was replaced by a glutamate residue using PyMOL, and Supplementary Fig. 8g shows a superposition of the AF3 model of the human wt P2X4 receptor with our cryo-EM structure. In the model of the wt P2X4 receptor, Glu307 of one subunit forms salt bridge interactions with both Arg82 and Lys298 of the adjacent subunit, flanked by a hydrogen bond between Arg301 of the first subunit with Asn110 of the adjacent subunit (Fig. 2a). The ionic intermolecular

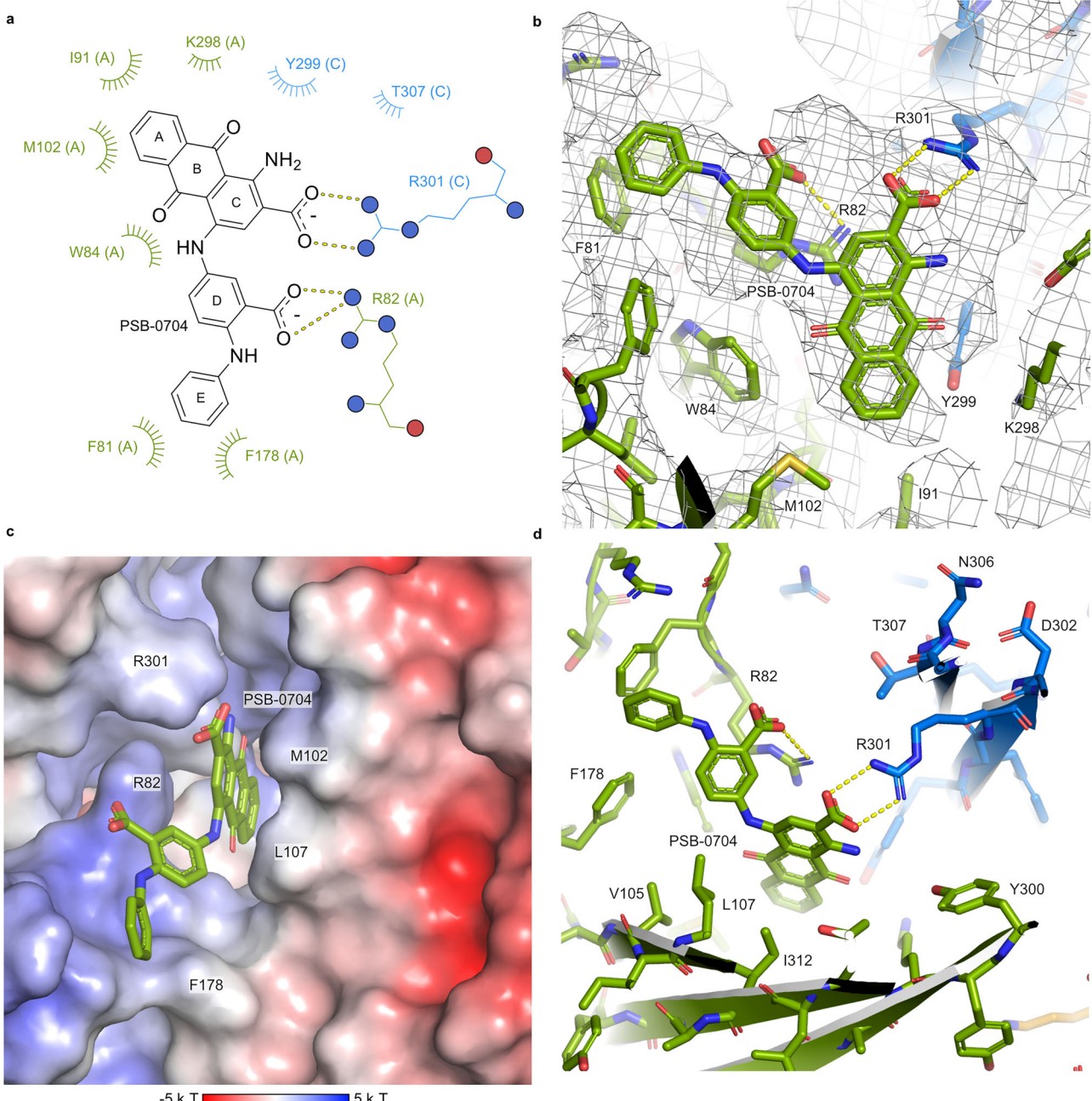

**Fig. 7 | Molecular interactions of the antagonist PSB-0704 binding to its allosteric site at the human P2X4 receptor. a** Schematic representation illustrating the interaction between PSB-0704 and the human P2X4 receptor. Polar interactions are shown as dashed lines and hydrophobic and van-der-Waals interactions as arcs. The letters in parentheses indicate the polypeptide chain of amino acid residues. **b** Close-up of the PSB-0704 binding site with the reconstructed volume shown as a black mesh. The color scheme is the same as in Fig. 6. **c** Surface electrostatic potential (calculated with APBS[88]) of the ligand binding site. **d** Close-up of the binding site.

interactions mediated by Glu307 block a significant part of the compound binding site (also see Supplementary Fig. 9).

In order to confirm crucial interactions of the anthraquinone derivative PSB-0704, and its potent close analog PSB-0739, in which the two carboxylate groups were replaced by sulfonate, we performed additional mutagenesis studies and performed TEVC studies on wt and mutant P2X4 receptor-expressing *Xenopus* oocytes (see below, Fig. 8).

**Two-electrode voltage clamp experiments and structure-based mutagenesis studies**
Amino acid residues to be exchanged were selected based on the determined cryo-EM structure and the observed interactions. The wt

P2X4 receptor and its mutants were expressed in *Xenopus* oocytes, and TEVC measurements were performed to determine inhibition of ATP-induced currents by PSB-0704 and its potent analog PSB-0739. The concentration-response curves of ATP determined at the wt and the E307T mutant P2X4 receptor were virtually identical (see Fig. 8a), and an $EC_{50}$ value of ca. 10 μM was determined. Subsequently, concentration-dependent inhibition of the ATP (10 μM)-induced current by PSB-0704 and PSB-0739 was measured at the P2X4-E307T receptor. Both compounds showed concentration-dependent inhibition with $pIC_{50}$ values of 5.79 (PSB-0704) and 6.49 (PSB-0739) (for current traces and inhibition curves see Fig. 8b, c). Thus, the inhibitory potency of the antagonists appeared to be

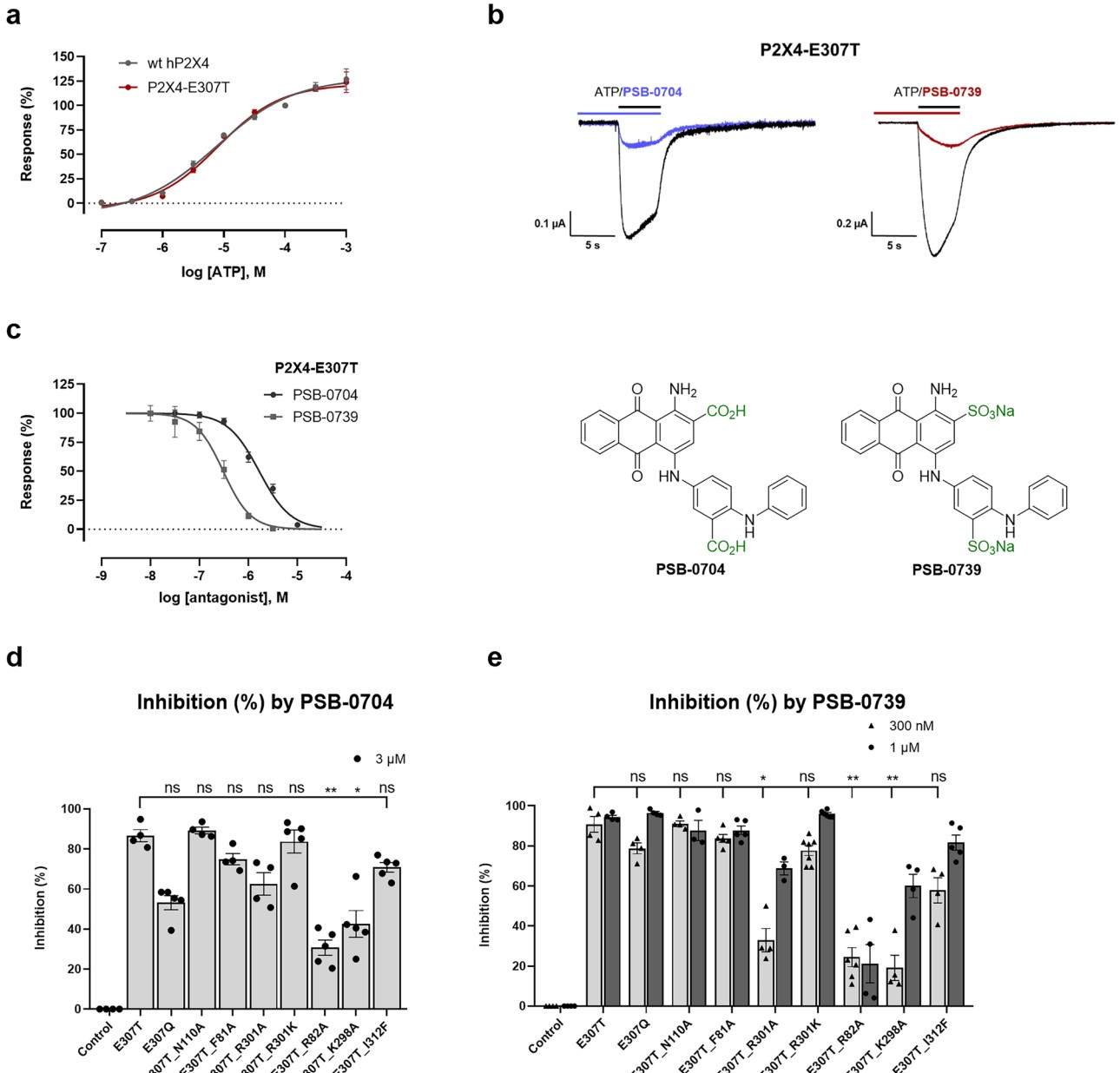

**Fig. 8 | Two-electrode voltage clamp (TEVC) experiments and structure-based site-directed mutagenesis studies. a** Concentration-response curves for ATP at the wild-type (wt) human (h) P2X4 receptor and the hP2X4-E307T receptor mutant. Receptors were expressed in *Xenopus laevis* oocytes and current responses to 5-s pulses of ATP were analyzed by TEVC at −60 mV. ATP test concentrations and a reference concentration of 100 μM ATP were alternately applied in 2 min intervals. Responses were calculated in relation to the reference concentration, fitted to the four-parameter Hill equation, and normalized to the maximal response. Data are presented as mean ± SEM, for wt, log ATP (M) [−7.0] *n* = 4, [−6.5] *n* = 3, [−6.0] *n* = 8, [−5.5] *n* = 8, [−5.0] *n* = 7, [−4.5] *n* = 6, [−4.0] *n* = 5, [−3.5] *n* = 5, [−3.0] *n* = 5; for E307T log ATP (M) [−7.0] n = 4, [−6.5] *n* = 3, [−6.0] *n* = 4, [−5.5] *n* = 4, [−5.0] *n* = 4, [−4.5] *n* = 7, [−4.0] *n* = 6, [−3.5] *n* = 6, [−3.0] *n* = 5. **b** Representative current traces recorded in oocytes expressing the P2X4-E307T receptor mutant. Colored lines indicate the presence of the indicated antagonist during a 2 min preincubation and during the application of 10 μM ATP (black line). 3 μM PSB-0704 and 300 nM PSB-0739 were used. **c** Concentraton-inhibition curves and chemical structures of the antagonists PSB-0704 and PSB-0739. Responses to 3-s pulses of 10 μM ATP were recorded after 20 s superfusion with the indicated antagonist concentrations and normalized to the preceding ATP response in the absence of antagonist. Data are represented as mean ± SEM, log PSB-0704 (M) [−7.5] *n* = 4, [−7.0] *n* = 4, [−6.5] *n* = 3, [−6.0] *n* = 3, [−5.5] *n* = 3, [−5.0] *n* = 4; log PSB-0739 (M) [−8.0] *n* = 4, [−7.5] *n* = 4, [−7.0] *n* = 6, [−6.5] *n* = 4, [−6.0] *n* = 5, [−5.5] *n* = 1. TEVC analysis of inhibitory effects

of **d** PSB-0704 3 μM (E307T *n* = 4, E307Q *n* = 5, E307T_N110A *n* = 4, E307T_F81A *n* = 4, E307T_R301A *n* = 4, E307T_R301K *n* = 5, E307T_R82A *n* = 5, E307T_K298A *n* = 5, E307T_I312F *n* = 5), and **e** PSB-0739 300 nM (E307T *n* = 4, E307Q *n* = 4, E307T_N110A *n* = 4, E307T_F81A *n* = 5, E307T_R301A *n* = 4, E307T_R301K *n* = 7, E307T_R82A *n* = 6, E307T_K298A *n* = 4, E307T_I312F *n* = 4, control only as a reference *n* = 4) and 1 μM (E307T *n* = 4, E307Q *n* = 4, E307T_N110A *n* = 3, E307T_F81A *n* = 5, E307T_R301A *n* = 3, E307T_R301K *n* = 6, E307T_R82A *n* = 4, E307T_K298A *n* = 4, E307T_I312F *n* = 5, control only as a reference *n* = 4) on the indicated single and double P2X4 receptor mutants expressed in *Xenopus laevis* oocytes. Inhibition of current responses evoked by 3-s pulses of 10 μM ATP after 2 min superfusion with antagonist is shown. Current responses evoked by co-application of ATP and antagonist were normalized to the preceding ATP responses in the absence of antagonist, and effects were represented as percent inhibition. Data are presented as mean ± SEM from at least 3 oocytes. The non-parametric Kruskal–Wallis test was applied, **d** *H* = 33.72, 8 d.f, *P* value < 0.0001; **e** *H* = 36.62, 8 d.f, *P* value < 0.0001, followed by Dunn's multiple comparisons test **d** E307T vs E307T_R82A, *P* = 0.0014; E307T vs E307T_K298A, *P* = 0.0025; E307T vs E307T_R301A, *P* = 0.0198; **e** E307T vs E307T_R82A, *P* = 0.0019; E307T vs E307T_K298A, *P* = 0.0180. The significance levels were defined as follows: **P* < 0.05, ***P* < 0.01, ****P* < 0.001, *****P* < 0.0001; ns not significant. For PSB-0739, the significance was evaluated for the lower concentration of 300 nM. Source data are provided as a Source data file.

much weaker in this assay under the applied conditions (pre-incubation for 20 s) than in the calcium influx assay, where the cells had been preincubated with the antagonists for 30 min. Subsequent preliminary experiments using different inhibitor concentrations, and increasing the incubation time (up to 2 min) indicated that prolonged preincubation led to an increase in potency, in particular for PSB-0704 (see Supplementary Fig. 10). A longer incubation time was not employed because current responses became unstable after longer periods without perfusion, and perfusion with antagonist for longer periods was limited by the availability of antagonist. Therefore, we subsequently tested the receptor mutants starting with antagonist preincubation for 2 min, followed by receptor activation with 10 μM of ATP (see Fig. 8). The more potent PSB-0739 was tested at two concentrations (300 nM and 1 μM) to observe concentration-dependency of the effects, while PSB-0704 was tested at a single, high concentration of 3 μM, based on preliminary dose-finding experiments.

The mutant E307Q was introduced into the wt P2X4 receptor for comparison with E307T. In both mutants, the antagonists PSB-0704 and PSB-0739 showed strong inhibition, confirming that the replacement of the acidic glutamate at this position by a neutral amino acid residue disrupted the ionic lock and enabled binding and inhibition by the allosteric antagonists. Several further double mutants were investigated containing a mutation of an amino acid directly or indirectly involved in ligand binding, in addition to the E307T substitution (Fig. 8). Both antagonists showed similar effects, indicating that they share the same binding mode and interactions. When the basic amino acid residues present in the allosteric binding site, R301, R82, and K298, were exchanged for alanine, the antagonists exhibited reduced potency at the resulting mutants. R82A had the largest effect—it forms an electrostatic interaction with the acidic group on ring D of PSB-0704—and pre-sumably also of PSB-0739, as well as a cation-π interaction with rings B and C (Fig. 7). At the R301A mutant the effect was moderate, only being significant for the lower concentration of PSB-0739, but not for the higher concentrations of both antagonists. Homologous replacement in the R301K mutant did not alter antagonist potency. According to the cryo-EM structure, R301 forms an electrostatic interaction with the acid residue in the 2-position (ring C) of the anthraquinone, which is consistent with our mutagenesis results. K298 likely forms an interaction with Y299 of an adjacent receptor monomer, thereby stabilizing the binding pocket, indirectly con-tributing to the interactions. The mutant N110A did not alter receptor inhibition by the antagonists. It was predicted to form a hydrogen bond with R301 in the wt P2X4 receptor in the absence of an antagonist, but it does not interact with the antagonist PSB-0704, while R301 binds to the antagonist. Finally, aromatic and lipophilic amino acid residues were exchanged. Unfortunately, the W84A and the F296Q mutants showed no proper expression. The F81A mutation had no significant effect, indicating that aromatic interactions with ring E of the anthraquinone derivatives are not essential, but may be substituted by the lipophilic alanine. The I312F mutation resulted in only a minor, non-significant decrease in antagonist potency; phenylalanine is larger than isoleucine, but still tolerated.

Finally, we studied the inhibitory potency of the standard P2X4 receptor antagonists BX430, BAY-1797, and 5-BDBD in ATP-induced calcium influx assays at the human P2X4-E307T receptor mutant expressed in 1321N1 astrocytoma cells. This was motivated by the fact that the allosteric binding site for anthraquinones discovered in the present work is adjacent to the previously discovered binding site for BX430 and BAY-1797. In fact, both compounds were several-fold more potent at the mutant as compared to the wt P2X4 receptor, while 5-BDBD was equally potent at both the mutant and the wt receptor (Table 1 and Supplementary Fig. 11).

## Discussion

The P2X4 receptor has emerged as a potential drug target, and its blockade was proposed to be beneficial for treating chronic pain, inflammatory diseases, and cancer[12,16,19–21]. Since the polar nature of the ATP-binding site and its similarity to other P2X receptor subtypes have impeded the discovery of orthosteric P2X4 receptor antagonists, drug research has been focusing on the development of negative allosteric modulators. However, up to now, only very few P2X4 receptor antagonists have been described, most of which are only moderately potent[4]. Structural biology of membrane proteins has recently made huge progress; in particular, cryo-EM has become the method of choice to determine structures of P2X receptors due to the large size (>100 kDa) of the trimeric ATP-gated cation channels. To date, 50 high-resolution structures of five P2X receptor subtypes in complex with different agonists and antagonists have been published (see Supple-mentary Table 1 for a complete list). Besides the characterization of the orthosteric ATP-binding site, this has led to the discovery of several allosteric sites for antagonists at the various P2X receptor subtypes. In Fig. 9, the locations of the discovered binding sites are depicted, including that of the anthraquinone derivative PSB-0704 determined in the present study.

The anthraquinone derivative Cibacron Blue was initially described as a moderately potent antagonist of human and rat P2X4 receptors in patch-clamp studies on receptors expressed in *Xenopus laevis* oocytes[65]. Later on, positive allosteric modulation was observed at lower con-centrations while inhibition was found to require higher Cibacron Blue concentrations, determined at the rat P2X4 receptor[31]. At rat P2X2 receptors, the compound acted as an antagonist, while at the human P2X3 receptor, it behaved as a positive allosteric modulator[31,32]. No direct effects were observed on P2X1 and P2X7 receptors[32]. Initially, we further characterized Cibacron Blue and performed SAR studies (see Table 1). Our interest in identifying the allosteric binding site of anthraquinone derivatives was also motivated by their physicochemical properties: in contrast to other P2X4 receptor antagonists, they are well water-soluble. Investigation of a series of truncated anthraquinone derivatives related to Cibacron Blue, bearing modifications in various positions, led to the discovery that smaller sulfo- or carboxy-substituted anthraquinone derivatives were still able to block the P2X4 receptor, and the amino group in the 1-position was dispensable (see Table 1). The sulfonate functions could be exchanged for carboxylate (in PSB-0704), which makes the molecules more drug-like.

A combination of different, complementary approaches was applied to identify the P2X4 receptor's binding site for anthraqui-nones. Chimeric receptors were studied[56], initially aimed at investi-gating and understanding the different behavior of Cibacron Blue at the P2X4 versus the P2X2 receptor. One of the P2X4(P2X2) receptor chimeras showed a strikingly different behavior in comparison to the wt receptors, exhibiting a markedly increased inhibitory potency of Cibacron Blue with an IC$_{50}$ value of 4.76 nM determined in calcium influx assays, a 2500-fold increase as compared to the wt P2X4 receptor (see Fig. 1e, f). Computational studies led to the hypothesis that the glutamate Glu307 forms an ionic lock with a nearby basic amino acid residue (likely Arg82 and Lys298, but Arg301 can also not be completely excluded), thereby preventing high-affinity binding of the anthraquinone derivatives by occluding part of its binding site (Table 1). This hypothesis was supported by the fact that small anthraquinone derivatives with only one acidic function (e.g., PSB-0826, PSB-25012, Table 1 and Supplementary Fig. 9) were several-fold more potent at the wt P2X4 receptor than the larger derivatives bearing two or three acidic groups (Cibacron Blue, PSB-0704, PSB-0739). A point mutation in the P2X4 receptor, in which Glu307 was exchanged for threonine, confirmed the hypothesis, Cibacron Blue being 680-fold more potent at the P2X4-E307T receptor mutant as compared to the wt receptor. The E307T mutation had no effect on the potency of the agonist ATP, as demonstrated in calcium influx and in

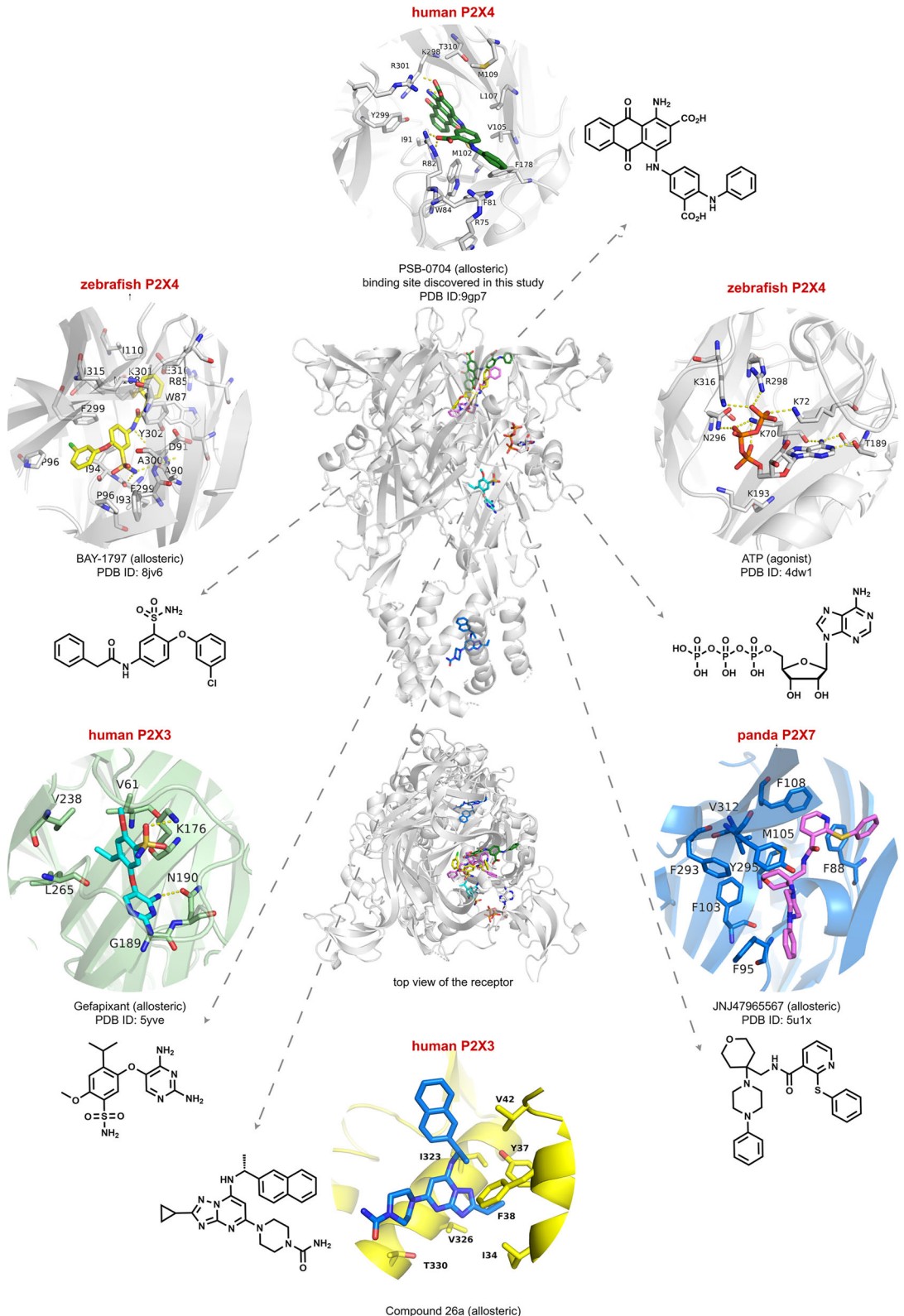

**Fig. 9 | Allosteric binding sites of P2X receptors.** Allosteric binding site of the anthraquinone derivative PSB-0704 at the human P2X4 receptor discovered in this study, compared to the orthosteric ATP-binding site and to the allosteric sites previously determined in various P2X receptor subtypes. P2X receptor side and top view showing the allosteric binding sites. In addition, close-up views of each binding site with the chemical structure of the allosteric antagonist (or ATP) and the PDB ID are depicted.

the voltage clamp assay (Fig. 8 and Supplementary Fig. 3). To gain conclusive insights into the binding pocket, determination of a high-resolution structure is indispensable.

Several high-resolution X-ray and cryo-EM structures of the zebrafish P2X4 receptor have been determined[35,36,38]. However, the zebrafish P2X4 receptor only shares 58% sequence identity with the human P2X4 receptor and exhibits a markedly different pharmacology. For instance, the potency of ATP at the zebrafish P2X4 receptor is about 700-fold lower than at the human P2X4 receptor[35,66]. Modifications were made to improve the crystallization behavior of the zebrafish P2X4 receptor, including truncation of its amino and carboxyl termini and insertion of two single-point mutations at the glycosylation sites (N78K/N187R)[35]. The receptor construct was shown to retain functionality in electrophysiological experiments, indicating that ATP is still able to activate the receptor (ATP concentration of 1 mM), although the peak current amplitudes were observed to be smaller than those recorded from the full-length receptor[35].

In contrast to these drastic modifications, our structure, as well as a recently published one[51], was obtained from the full-length human P2X4 receptor construct. The current sequence harbored only three single-point mutations, E307T, N75R, and N184R. The latter two are mutated glycosylation sites, the same as those described for the zebrafish P2X4 receptor structure[35].

The obtained structure exhibited a sufficiently high resolution of 3.35 Å to identify the anthraquinone's binding site and interactions. The receptor showed a trimeric structure, as expected, with three well-resolved allosteric binding sites for PSB-0704, located at the trimer subunit interface, at the back of the head domain. Even the glycosylation sites are well visible, at Asn110 in the upper body, Asn208 in the dorsal fin, and Asn153 in the head domain. The PSB-0704-bound receptor is in an inactive conformation, as clearly seen by comparison with active and inactive zebrafish X-ray structures and also compared to recent human P2X4 receptor structures[51] (see Supplementary Figs. 8a–c and 12). The structural comparisons revealed that the antagonist appears to induce a "downward" movement of the head domain towards the membrane plane. The β-hairpin loop containing Arg82 is bent backwards compared to the AF3 model of the human P2X4 receptor without ligand. The anthraquinone ring system of PSB-0704 forms hydrophobic and cation-π stacking interactions with the receptor.

The studied mutations support the determined structure: (1) Thr307 is located near two basic amino acids, Arg82 and Arg301, which form ionic interactions with the two carboxylate groups of the antagonist. In the wt receptor, there is a glutamate (Glu307) instead of threonine, which is predicted to form an ionic bridge with Arg82 and additionally with Lys298 according to the computational model (see Fig. 2a). Exchange of Glu307 for glutamine had virtually the same effect as its exchange for threonine (Fig. 8), both amino acid residues breaking the ionic lock. In the recent structure of the human P2X4 receptor in complex with the antagonist BAY-1797[51], which binds to a different allosteric site than PSB-0704, the ionic lock is closed, further confirming our hypothesis. Mutation of each of the involved basic amino acid residues, Arg82, Lys298, and Arg301, to alanine in the P2X4-E307T receptor led to a reduction in inhibitory potency of the potent anthraquinones PSB-0704 and PSB-0739, while a charge conservation in the R301K mutant had no consequences (Fig. 8). (2) The mutation Q308T (Fig. 3) and N110A (Fig. 8) had no effects on the potency of the anthraquinones, and indeed, the structure showed no interactions. These results suggest that the interaction between N110 and R301, predicted by AF3 and also observed in a recent cryo-EM structure of the closed apo state of the P2X4 receptor, does not play a role in the binding of the anthraquinone-derived antagonists. (3) The moderate effect of the D302I mutation (9-fold higher potency of Cibacron Blue, Fig. 3) can be explained by an interaction of Asp302 with Asn306, thereby potentially exerting conformational effects on the loop containing Arg301.

All of these and further mutations are in agreement with the determined structure, supporting our ionic-lock hypothesis on the wt P2X4 receptor. Together with the SARs at the wt and the E307T mutant receptor (Table 1), it can be concluded that the anthraquinone derivatives interact with the same binding site on both mutant and wt receptor (also see Supplementary Figs. 8 and 9 for docking of anthraquinone derivatives into the human wt and mutant P2X4 receptor). Smaller compounds and those with only one negative charge are better tolerated by the wt receptor than those bearing two acidic functions. The determined structure, although not obtained for the wt P2X4 receptor, is therefore useful for drug design. It rationalizes the SAR data showing, for example, that the amino group at ring C is not important for the interaction with the P2X4 receptor (Table 1). In our structure, the amino group is ~4 Å away from the hydroxyl group of Tyr300, which is too far to form a direct hydrogen bond (Fig. 7). Aromatic amines (anilines) are regarded as potential toxicophores[67], and therefore these insights can be used to improve the compounds not only with regard to potency, but potentially also with regard to toxicity. The oxygen atoms of the quinone ring B have not shown any polar interactions with the receptor and thus are likely also not required. This opens up the possibility to design antagonists without the redox-reactive quinone structure by replacing it with a more stable moiety.

Cibacron Blue is only weakly active at the wt P2X2 receptor, although the receptor lacks an ionic lock. This is likely due to the fact that the residues directly involved in ligand binding are not all conserved in the corresponding pocket of the human P2X2 receptor, and there are numerous smaller structural differences. In light of our data, the most obvious difference is the replacement of the very important Arg82 in the P2X4 receptor by the corresponding Lys91 in the P2X2 receptor. The Lys side chain is shorter and cannot form the cation-π interaction observed for Arg82 with rings B and C of PSB-0704's anthraquinone ring system. It should be pointed out that anthraquinone derivatives with relatively high affinity and selectivity for the P2X2 receptor, but not for other P2X receptor subtypes, have previously been developed[68], indicating that the P2X2 binding site can, in fact, well accommodate certain anthraquinone derivatives depending on the substitution pattern.

Since the hypothesis of ionic lock formation has been confirmed, this offers the possibility to design compounds that disrupt the intermolecular ionic lock by interacting with Glu307, to open up a larger, high-affinity binding site for antagonists. Introduction of a basic moiety into the structure of PSB-0704, e.g., attached to the anthraquinone 1-position instead of or linked via the anilinic amino group, might be a plausible strategy. However, the design of an antagonist that could break an ionic lock is certainly most challenging, even though the disruption of ionic locks is a common mechanism in drug-receptor interactions. For example, in G-protein-coupled receptors, ionic locks stabilize the inactive state of the receptor[69,70]. They typically act as molecular switches, which can be disrupted by drugs[71].

The moderately potent compounds BX430 and BAY-1797 are among the best investigated P2X4 receptor antagonists. Both have recently been shown to share the same allosteric binding site as determined by cryo-EM in complex with the zebrafish P2X4 receptor (see Supplementary Fig. 8e)[38]. The allosteric site discovered for the anthraquinone PSB-0704 in the present study is in the same region, but distinct from the BX430 and BAY-1797 binding sites, with which it only slightly overlaps. It is also distinct from any other allosteric binding site identified in P2X receptor subtypes so far, although the P2X7 receptor also harbors an allosteric binding site nearby (see Fig. 9 for an overview of orthosteric and allosteric P2X receptor binding sites). Compared to the P2X4 receptor antagonists BX430 and BAY-1797, the antagonist PSB-0704 is located further away from the membrane surface (Supplementary Fig. 8e). Interestingly, both compounds show increased affinity for the P2X4-E307T mutant as compared to the wt receptor (Table 1), indicating that the disruption of the ionic lock is also

beneficial for their binding interactions. This is not the case for another allosteric P2X4 receptor antagonist, 5-BDBD (Table 1), which can be expected to interact with a different binding site. The fact that anthraquinones and BX430/BAY-1797 have adjacent binding sites opens interesting opportunities to combine features of the different compound classes to design more potent antagonists. The design of bitopic or dualsteric ligands, substituting or replacing the benzoquinone moiety of the anthraquinones by a linker that connects the antagonist with the terminal phenylacetic acid amide moiety of BAY-1797, would be a promising strategy (Fig. 9 and Supplementary Fig. 8e).

Our results do not explain the biphasic behavior of Cibacron Blue at the human P2X4 receptor. It has been observed previously that some allosteric modulators can show receptor activation *or* inhibition depending on concentration[56]. Based on the present structural data, it is tempting to speculate that the degree of downward movement of the head domain induced by a single molecule of the relatively large Cibacron Blue could account for the potentiating effect observed with low concentrations of the ligand. Thus, only Cibacron Blue can induce a "pre-open" state of the receptor and consequently higher agonist efficacy, even if only two ATP molecules can bind. If, however, two or more antagonist molecules are bound, ATP cannot bind or induce the required conformational change, and the receptor is blocked. Another explanation could be the presence of different binding sites to which the modulator has different affinity, as postulated for some pentameric ligand-gated ion channels and their allosteric ligands[72]. This hypothesis is supported by the results obtained in radioligand binding studies using the stable ATP analog [$^{35}$S]ATPγS, in which Cibacron Blue showed a biphasic competition binding curve[30]. It has to be pointed out that this behavior is only observed for the large molecule Cibacron Blue, consisting of 6 rings, but not for any of the truncated analogs, including PSB-0704, all of which only exhibit antagonistic activity, in calcium influx assays at recombinant 1321N1 astrocytoma cells as well as in TEVC experiments at Xenopus oocytes.

Cibacron Blue and closely related analogs are often promiscuous compounds interacting with a large number of proteins with moderate affinity[33]. The high-affinity interaction of the mutated P2X4-E307T receptor with Cibacron Blue (IC$_{50}$ 17.8 nM) offers unique opportunities: this protein-ligand pair could be used for specifically elucidating P2X4 receptor functions. For example, knock-in animals expressing the P2X4-E307T receptor in organs or cell types of interest could be prepared. These would allow the specific blockade of the mutated receptor by suitable concentrations of antagonist without effects on the wt receptor expressed at other sites. Such an approach of chemical hijacking using an engineered protein-ligand pair has found broad application in basic research, in particular in chemical biology[73,74].

In conclusion, we discovered an allosteric binding site for anthraquinone derivatives in the human P2X4 receptor, along with important interactions. The binding site is adjacent to, but not identical with, a recently described allosteric P2X4 receptor binding site[38,51]. This could have relevance beyond the P2X4 receptor subtype since some anthraquinones, e.g., Cibacron Blue, have been found to also interact with several other P2X receptor subtypes[31,32,68]. The presented results, along with recent literature data, provide important insights into the interaction of anthraquinone derivatives with the human P2X4 receptor. Our findings constitute a basis that could support future drug development efforts.

## Methods

### Chimeric P2X4(P2X2) receptors
Chimeric P2X4(P2X2) receptor constructs were obtained through gene synthesis (Geneart, Life Technologies). All cDNAs were defined with attached EcoRI and BamHI restriction sites, and subcloned into the bicistronic lentiviral pLVX-IRES-mCherry plasmid (Clonetech)[56]. Verification of the constructed chimeric receptors was carried out by sequencing (GATC).

### Receptor mutants
Receptor mutants were generated by site-directed mutagenesis with mutagenesis primers, synthesized by Invitrogen.

### Transfection of 1321N1 astrocytoma cells
Two different expression systems were used for generating stable 1321N1 astrocytoma cell lines: lentiviral and retroviral expression[56].

For lentiviral expression, the genes of interest were subcloned into the pLVX-IRES-mCherry vector to coexpress the gene of interest and the mCherry gene. In brief, HEK293 T packaging cells were transfected using Xfect Reaction buffer, Lenti-X HTX Packaging-Mix, and Xfect-Polymer (Polymer) for virus production. Afterwards, non-transfected 1321N1 astrocytoma cells were infected with the produced viruses. After 48–72 h, the cells were selected for fluorescence emission at 610 nm with an excitation of 561 nm. As a negative control, non-transfected 1321N1 astrocytoma cells were measured. Cells with high expression of mCherry protein were single-sorted to 96-well plates using a BD FACS Aria III.

The second method used was retroviral transfection. In this case, GP+envAM-12 fibroblast-type mouse packaging cells were used to produce a recombinant helper virus. The gene of interest was subcloned into the pQCXIN vector. Briefly, GP+envAM-12 packaging cells were transfected with the plasmid containing the DNA of interest and the vesicular stomatitis virus G (VSV-G) protein DNA (in pcDNA3.1) to produce recombinant helper viruses. The non-transfected 1321N1 astrocytoma cells were then infected with the produced virus particles. The selection was initiated using geneticin G418 (800 μg/mL) 48–72 h after infection.

### Cultivation of cells
1321N1 Astrocytoma cells stably transfected with the human wt P2X4 receptor, the human wt P2X2 receptor, the chimeric P2X4(P2X2) receptors, and the receptor mutants as well as HEK293 T and GP+envAM-12 packaging cells were cultured in Dulbecco's Modified Eagle Medium (DMEM, Thermo Fisher Scientific) supplemented with 10% fetal calf serum (FCS, PAN Biotech), and 1% PenStrep (10,000 U/mL penicillin, 10 mg/mL streptomycin) (PAN Biotech). The medium for retrovirally-transfected cell lines contained G418 (800 μg/mL). Stably transfected 1321N1 astrocytoma cells were incubated at 37 °C with 10% CO$_2$, HEK293 T, and GP+envAM-12 packaging cells at 37 °C with 5% CO$_2$.

### Chemicals
Cibacron Blue 3GA was purchased from Sigma-Aldrich and used without further purification. The synthesis of all other anthraquinone derivatives, PSB-0704, PSB-0739, PSB-24039, PSB-0826, PSB-25041, PSB-0801, PSB-0711, PSB-25012, and PSB-09059 has been published[68,75–79]. In brief, compounds PSB-0704, PSB-0739, PSB-0826, PSB-25041, PSB-0801, PSB-0711, PSB-25012, and PSB-09059 were prepared via modified Ullmann coupling reaction under microwave irradiation starting from bromaminic acid, or 1-amino-4-bromo-2-methylanthraquinone, respectively, and the appropriate aniline derivative[75–77]. Compound PSB-24039 was obtained by deamination reaction[78] of compound PSB-0739.

### Calcium influx assays
To quantify intracellular Ca$^{2+}$ levels in transfected 1321N1 astrocytoma cells, fluorimetric calcium influx assays were performed according to previously established protocols[56,60].

Briefly, transfected cells were plated into 96-well plates and pre-incubated with the fluorescent Ca$^{2+}$-chelating dye Fluo-4 acetoxymethyl ester (AM) (Thermo Fisher Scientific), dissolved in Hanks' balanced salt solution (HBSS, Thermo Fisher Scientific) supplemented with 1% Pluronic® F127 (Sigma-Aldrich), for 1 h at room temperature. For experiments involving antagonist testing, series dilutions of antagonists (prepared either in dimethyl sulfoxide (DMSO) or in HBSS buffer) were

added to the cells, and incubation for 30 min was performed before the agonist ATP was added at its $EC_{80}$. Fluorescence imaging plate readers, Flex station (Molecular Devices), or NOVOstar (BMG Labtech GmbH), were used to measure fluorescence at an excitation wavelength of 485 nm and an emission wavelength of 520 nm.

## AlphaFold 3 models

The AF3 models used in this manuscript (P2X4 trimer (non-glycosylated and glycosylated)), the P2X4/P2X2 chimera (P2X4(P2X2) c10$^{R301-Q308}$) trimer, and the P2X2 trimer were created using the AF3 webserver (https://alphafoldserver.com/). The resulting model and quality indicators are summarized in Supplementary Fig. 12. The models, including their sequences together with the complete output of the AF3 server, are also provided for download as source data files.

## Design of the receptor construct

The full-length human P2X4 receptor gene was subcloned into the pFastBac1 vector (Thermo Fisher Scientific) with a polyhedrin promoter. To the C-terminus, a polyhistidine tag (10x-His tag) was attached, which included a preceding human rhinovirus 3C protease cleavage site. A flexible Gly-Ser-Ser-Gly (GSSG) linker was incorporated prior to the protease cleavage site. Two of the receptors' glycosylation sites were mutated (N75R and N184R), and at position 307, glutamate was mutated to threonine (E307T). No truncations were made for structure determination.

## Receptor expression and purification

The modified human P2X4 receptor protein was expressed in *Sf*9 insect cells (Expression Systems) using the Bac-to-Bac baculovirus expression system (Invitrogen). Freshly grown *Sf*9 insect cells were infected with recombinant baculoviruses (P1) at a density of $2-3 \times 10^6$ cells mL$^{-1}$ for 48 h at 27 °C and 150 rpm. Cell pellets were harvested by centrifugation ($4000 \times g$) and stored at −80 °C.

All subsequent steps were carried out on ice or at 4 °C. Cell membrane preparations were obtained by repeated homogenization using a Dounce glass homogenizer in a hypotonic buffer (10 mM HEPES (pH 7.5), 10 mM $MgCl_2$, 20 mM KCl) supplemented with EDTA-free complete protease inhibitor cocktail tablets (Roche), followed by centrifugation for 30–60 min at $57,000 \times g$ (twice). Then, membranes were washed in the same way using high osmotic buffer (10 mM HEPES (pH 7.5), 10 mM $MgCl_2$, 20 mM KCl, 1 M NaCl) (twice). Purified membranes were resuspended in 10 mM HEPES (pH 7.5), 10 mM $MgCl_2$, 20 mM KCl, 30% (v/v) glycerol, flash-frozen in liquid nitrogen, and stored at −80 °C.

Prior to receptor solubilization, cell membranes were thawed on ice and incubated in the presence of 37 μM of PSB-0704 dissolved in DMSO at 4 °C for 30 min. P2X4 receptors were then solubilized with 50 mM HEPES (pH 7.5), 100 mM NaCl, 1.5% (w/v) DDM (Anatrace), 0.3% (w/v) CHS (Sigma-Aldrich) for 3 h at 4 °C.

The solubilized fraction (supernatant) was collected after centrifugation for 30–60 min at $48,000 \times g$. Then, 20 mM of imidazole (pH 7.5) and 0.01 mL of TALON IMAC resin beads (Takara bio) per mL of supernatant were added and incubated for at least 3 h at 4 °C. Prior to use, TALON beads were washed with Millipore water (8 times) and stored in buffer (50 mM HEPES (pH 7.5), 0.5 M NaCl, 10% (v/v) glycerol) at 4 °C.

The resin was washed with 15–20 column volumes (CV) of wash buffer I containing 50 mM HEPES (pH 7.5), 100 mM NaCl, 0.1% (w/v) DDM, 0.02% (w/v) CHS, 10% (v/v) glycerol, 25 mM imidazole, 20 mM $MgCl_2$, and 13 μM PSB-0704, followed by washing with wash buffer II (10–15 CV) containing 50 mM HEPES (pH 7.5), 100 mM NaCl, 0.05% (w/v) DDM, 0.01% (w/v) CHS, 10% (v/v) glycerol, 40 mM imidazole, and 15 μM PSB-0704. Finally, the protein was eluted from the column with five CV of elution buffer (25 mM HEPES (pH 7.5), 100 mM NaCl, 0.025% (w/v) DDM, 0.005% (w/v) CHS, 10% (v/v) glycerol, 220 mM imidazole,

and 15 μM of PSB-0704) and concentrated using a 100-kDa molecular weight cut-off concentrator (Amicon™; Merck).

## Reconstitution into amphipol

Relevant information on method development and optimization was taken from previously published protocols[80]. For reconstitution into amphipol A8-35, the solubilized P2X4 receptor protein was mixed with A8-35 (Anatrace) dissolved in Millipore water at a ratio of 1:10 (w:w), and the mixture was incubated for at least 4 h at 4 °C while gently stirring. The appropriate amount of A8-35 was calculated based on the mass of the protein present in the sample. For detergent removal, Bio-Beads SM-2 (Bio-Rad) were added to the sample (20 × mass of detergent present in the sample). Prior to use, Bio-Beads were washed with 20% (v/v) ethanol and Millipore water. Detergent removal was performed overnight at 4 °C while gently stirring. After removing Bio-Beads using small chromatography columns (Bio-Rad), the reconstituted protein was concentrated using a 100-kDa molecular weight cut-off concentrator (Amicon™; Merck). For final protein purification, size-exclusion chromatography (SEC) was performed on a Superose 6 Increase 3.2/300 column (Cytiva) pre-equilibrated with SEC buffer (20 mM HEPES (pH 7.5) and 100 mM NaCl). The respective peak fractions of the trimeric human P2X4 protein reconstituted in A8-35 were collected and concentrated using a 100-kDa molecular weight cut-off concentrator (Vivaspin, GE Healthcare). Finally, PSB-0704 dissolved in DMSO was added at a concentration of 50 μM before grid preparation for cryo-EM studies.

## Protein expression, purification, and reconstitution into nano-discs for radioligand binding studies

The human P2X4 receptor construct used for radioligand binding studies was the same as the one used for cryo-EM studies, except for an additional N-terminal GFP tag and a C-terminal Rho1D4 tag fused by a GSSG linker, respectively.

The trimeric P2X4 receptor protein, purified in 0.05% DDM and 0.005% CHS, was incubated with membrane scaffold protein 1E3D1 (MSP1E3D1), nanodiscs (Cube Biotech), biobeads (Bio-Rad), cholate, and 1-palmitoyl-2-oleoyl-*sn*-glycero-3-phosphocholine (POPC) lipids (Avanti Polar lipids) in a 1:50 (P2X4:nanodisc) ratio. After the assembly reaction, an SEC on a Superose 6 increase column (Cytiva) was performed in 100 mM NaCl and 20 mM Tris-HCl buffer (pH 7.4) to remove empty assembled nanodiscs. Gel filtration of the solubilized GFP-tagged P2X4 receptor was performed on a Superdex 200 10/300 GL column (Cytiva). The buffer contained 10% glycerol, 150 mM NaCl and 20 mM $Na_2PO_4$ (pH 7.5).

## Radioligand binding studies

Radioligand binding studies were performed according to a published procedure[60]. The radioligand [$^{35}$S]ATPγS (1250 Ci/mmol; $4.63 \times 10^{13}$ Bq/mmol) was obtained from Perkin Elmer. Homologous competition binding assays were performed with 0.2 nM [$^{35}$S]ATPγS in 50 mM Tris-HCl buffer containing 1 mM EDTA (pH 7.4). Non-specific binding of [$^{35}$S]ATPγS was determined in the presence of ATP (100 μM, Appli-Chem). Purified human P2X4-E307T-GFP receptor reconstituted in nanodiscs (various dilutions), or membrane preparations of 1321N1 astrocytoma cells expressing human wt P2X4 receptors were added to the samples. After an incubation of 1 h at room temperature, the incubation was terminated by rapid vacuum filtration through GF/B glass fiber filters using a Brandel 48-well harvester (Brandel). The filters were rinsed three times with ice-cold Tris-HCl buffer (50 mM, pH 7.4, 3–4 mL each) to separate unbound from bound radioligand. After punching out the filters and transferring them into scintillation vials, scintillation cocktail (2.5 mL, Ultima Gold®) was added. The samples were measured after 6 h of preincubation using a liquid scintillation counter for 1 min each.

## Data analysis

All data from calcium influx assays and radioligand binding studies were analyzed using Microsoft Excel. The calculation of $EC_{50}$, $EC_{80}$, and $IC_{50}$ values from calcium influx assays was performed with GraphPad Prism (version 4.0 or 8.0, GraphPad Software, Inc., San Diego, CA, USA) by nonlinear regression with a sigmoidal dose-response equation. $K_D$ and $B_{max}$ values from radioligand binding assays were calculated by using the "one site-homologous" equation, implemented in GraphPad Prism version 8.0. The level of significance was calculated by a one-way ANOVA with Dunnett's post hoc test, in case there was a control group, and multiple comparison was required. Significance was expected if the $P$ value was ≤0.05. The significance levels were defined as follows: $*P < 0.05$, $**P < 0.01$, $***P < 0.001$, $****P < 0.0001$.

## Electron microscopy

The purified A8-35-reconstituted P2X4 receptor protein was applied onto glow-discharged EM grids (Quantifoil, Au R1.2/1.3) and plunge-frozen in liquid ethane cooled by liquid nitrogen using a Vitrobot Mark IV plunge freezer (Thermo Fisher Scientific). Grids were transferred to a Glacios electron microscope (Thermo Fisher Scientific) equipped with a Gatan K2 direct electron detector. For data acquisition of A8-35-reconstituted P2X4 receptor protein in complex with PSB-0704, two datasets were acquired, one with 2994 images and another with 5710 images, collected with an acceleration voltage of 200 kV, a magnification of 45,000×, a pixel size of 0.901 Å/pixel, and a defocus varied from −1.0 to −2.5 μm. The average dose rate was 46.7 e/Å². The dataset was processed with CRYOSPARC 4.0-4.7[81] as detailed in Supplementary Fig. 6. Briefly, particles were picked with a blob picker job, followed by 2D classification to create templates for a template picker job, resulting in 3.6 million particles. An initial reconstruction clearly showed the P2X4 receptor but was plagued by severe resolution anisotropy. Junk particles were sorted out by heterogeneous refinement jobs, offering the initial reconstruction and five junk classes. All particles assorted to the junk classes were discarded, and the procedure was repeated 2 times. This procedure was followed by 3D classification without alignment, non-uniform refinement[82] and local refinement jobs leading to a focused reconstruction at 3.35 Å resolution with local resolutions in the range 2.5 Å to 12.2 Å. In a second branch of the processing tree (Supplementary Fig. 6), we aimed to obtain an optimal overall reconstruction of the molecule and therefore refrained from using local refinements. This led to an overall reconstruction of 3.67 Å resolution (Supplementary Fig. 6).

The maps were sharpened using DeepEMhancer[83]. An AF3 model of the human P2X4 receptor, including the ligand, was placed into the 3D reconstruction and refined with ISOLDE[63] and ChimeraX 1.9[84]. The ligand density was clearly visible in both reconstructions. We placed the compound in the extra density and tested different orientations with respect to the protein. The binding mode shown in the manuscript was clearly the best explanation for the map feature and the only one that led to reasonable interactions with the protein. We parametrized the ligand with PHENIX.elbow[85] and used ISOLDE[63] to refine its ligand position. B-factors of the model were refined with PHENIX[85], and the model was validated with PHENIX and MOLPROBITY[86].

## Frogs and oocyte preparation

*Xenopus laevis* frogs were from Nasco International (Fort Atkinson, WI, USA) and kept at the Core Facility Animal Models (CAM) of the BMC of LMU Munich, Germany (Az: 4.3.2−5682/LMU/BMC/CAM) in accordance with the EU Animal Welfare Act. Frogs were anaesthetized with MS222 and ovaries were surgically extracted (ROB_54-011-AZ 2532.Vet_02-23-166), treated with collagenase (Nordmark, Uetersen, Germany 1.0−1.5 mg/mL, ≥2h at RT) and defolliculated (15-min treatment in $Ca^{2+}$-free oocyte Ringer (90 mM NaCl, 1 mM KCl, 2 mM $MgCl_2$, 5 mM HEPES)). Fifty nanoliters of cRNA (0.5 μg/μL) were injected using a Nanoject II injector (Science Products, Drummond, USA) and oocytes

were kept in ND96 (96 mM NaCl, 2 mM KCl, 1 mM $MgCl_2$, 1 mM $CaCl_2$, 5 mM HEPES, pH 7.4−7.5) supplemented with 5 μg/mL gentamicin at 16 °C.

## cDNAs, cloning, mutagenesis, and cRNA synthesis

cDNA encoding human P2X4 receptor (NM_002560.3) was obtained from BioCat GmbH (Heidelberg, Germany) and cloned into the pNKS2 vector[87] using Gibson Assembly (E2621L, New England Biolabs, USA) according to the manufacturer's instructions. Point mutations were introduced by site-directed mutagenesis using KLD Enzyme Mix (M0554S, New England Biolabs, USA). All constructs were verified by full-length cDNA sequencing (Eurofins Genomics, Germany). Plasmids were linearized with NotI-HF (R3189S, New England Biolabs, USA) and purified using the MinElute Reaction Cleanup Kit (Qiagen, Hilden, Germany). Capped cRNA was synthesized using the mMESSAGE mMACHINE™ SP6 Transcription Kit (Thermo Fisher Scientific, Schwerte, Germany) and dissolved in nuclease-free water at a final concentration of 500 ng/μL. Injected oocytes were incubated at 16 °C for at least 36 h to allow for protein expression.

## Electrophysiological recordings

TEVC recordings were performed at room temperature in ND96 at a holding potential of −60 mV using a Turbo Tec 05X Amplifier (npi electronic, Tamm, Germany). Currents were filtered at 1 kHz, and digitized at 1 kHz using CellWorks software (npi electronic, Tamm, Germany). Solutions were automatically applied with a VC3-8xP valve system (ALA scientific instruments, USA) and perfusion speed was regulated by air pressure using a PR-10 analog pressure regulator (ALA scientific instruments, USA). Oocytes were placed in a 200 μL Teflon bath (Automate Scientific, USA) and solutions were applied via 1 mm diameter Teflon tubing and a Teflon micro-manifold to minimize ligand binding to surfaces. Oocytes were continuously perfused at 20 μL/s, and ligand-containing solutions were applied at ~250 μL/s. All measurements were performed with oocytes from at least two different frogs.

## Concentration-response analysis

To determine agonist concentration-response relationships, stable current responses were first established by applying 100 μM ATP for 5 s via solution exchange at 2-min intervals until currents stabilized. Once stable, alternating applications of the reference concentration and increasing ATP test concentrations were performed. For low test concentrations, an additional 2-s application of reference concentration was delivered immediately afterwards to maintain comparable fractions of desensitized channels. Each test concentration was normalized to the mean of the preceding and following reference concentrations.

To investigate allosteric antagonist effects, 10 μM ATP was repeatedly applied via solution exchange at the indicated intervals until stable current responses were obtained. Oocytes were then perfused with antagonist-containing solutions for the indicated pre-incubation time (20 s, 1 min, or 2 min) before currents were activated by 10 μM ATP (for 3 s) supplemented with the same antagonist concentration. Responses were normalized to the preceding reference concentration without antagonist.

## Data analysis and visualization of electrophysiological recordings

Concentration-response analyses were performed with GraphPad Prism version 9, and dose-response curves were fit to the data using the inbuilt Hill equation normalized response = (bottom + (top-bottom))/(1 + ($IC_{50}^{nH/c}$)) for antagonists and normalized response = (bottom + ($c^{nH}$ • (top-bottom)))/($c^{nH}$ + $EC_{50}^{nH}$) for agonists with $nH$ = Hill slope, $c$ = concentration (μM) and bottom and top constraint to 0 and 1, respectively. Statistical analysis was performed with GraphPad Prism version 10 software, and data are presented as

means ± SEM. Data were assessed for normality using the Shapiro–Wilk test and for homogeneity of variances using Levene's test. If both assumptions were met, a one-way or two-way ANOVA was conducted, followed by Tukey's post hoc test for multiple comparisons. In cases where the assumptions of normality or homoscedasticity were violated, the non-parametric Kruskal–Wallis test was applied, followed by Dunn's multiple comparisons test. The significance levels were defined as follows: $*P < 0.05$, $**P < 0.01$, $***P < 0.001$, $****P < 0.0001$; ns not significant.

## Ethics statement

Female *Xenopus laevis* frogs were from Nasco International (Fort Atkinson, WI, USA) and kept at the Core Facility Animal Models (CAM) of the Biomedical Center (BMC) of LMU Munich, Germany (Az:4.3.2–5682/LMU/BMC/CAM) in accordance with the EU Animal Welfare Act. Ovaries were surgically extracted (ROB_54-011-AZ 2532.Vet_02-23-166). Experiments were approved by the District Government of Upper Bavaria.

## Data availability

Source data are provided with this paper. The cryo-EM reconstructions generated in this study have been deposited in the Electron Microscopy Data Bank (EMDB) with access number EMD-51502. The coordinates of the human P2X4-E307T receptor in complex with PSB-0704 were deposited in the Protein Data Bank (PDB) under accession code 9gp7. The coordinate data used in this study are available in the PDB database under accession codes 4dw0, 4dw1, 8jv6, 8jv5, 9bqh, 9bqi. Additional data are provided in the Supplementary Information.

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

## Acknowledgements

C.E.M. is grateful to the German Federal Ministry of Education and Research (BMBF) for support (Biopharma Neuroallianz, 0315606B). C.E.M., J.S.-M. and A.N. were supported by the Deutsche Forschungsgemeinschaft (SFB 1328, projects A11 and A15) and the COST Action CA21130 "P2X receptors as a therapeutic opportunity (PRESTO)." The authors thank the Bonn International Graduate School of Drug Sciences (BIGS DrugS) for travel support. Y.B. is grateful to the Deutscher Akademischer Austauschdienst (DAAD) for a scholarship. J.T. is supported by a grant from the DFG (GE 976/16-1). G.H. and M.G. are members of the excellence cluster ImmunoSensation2, funded by the DFG under Germany's Excellence Strategy – EXC2151–390873048. The authors thank Christiane Bous for performing calcium influx assays and Marion Schneider for the determination of NMR and MS spectra. The authors are grateful for the expert technical assistance of Parisa Mohammadianazad for performing some of the calcium influx assays. Cryo-EM data were acquired at the Institut de Génétique, Biologie Moléculaire et Cellulaire in Illkirch-Graffenstaden, France, and were supported by an iNEXT-Discovery grant. The authors thank Alexandre Durand and Nils Marechal for cryo-EM data collection. The authors gratefully acknowledge the life science EM facility of the Ernst-Ruska Centre at Forschungszentrum Jülich (FZJ). We thank Thomas Heidler and Daniel Mann at FZJ for excellent help and advice on cryo-EM data acquisition and processing. We acknowledge access to the cryo-EM infrastructure of StruBiTEM (Cologne, funded by DFG Grant INST 216/949-1 FUGG), and help by Monika Gunkel with plunge-freezing.

## Author contributions

S.W., J.N., A.A. and L.T. performed the pharmacological studies under the supervision of C.E.M. J.K., B.M. and C.E.M. designed and optimized the initial P2X4 receptor construct. J.N., V.J.V., J.G.S., J.T. and T.C. optimized methods for P2X4 receptor expression and purification. J.N. expressed the P2X4 receptor protein, reconstituted it into amphipols, and purified it for cryo-EM measurements. J.S.-M. and A.N. performed TEVC studies at wt and mutant P2X4 receptors using *Xenopus* oocytes. Y.B. and H.A.M.A.M. synthesized and purified anthraquinone derivatives. V.N. and G.H. performed the computational studies. J.N. prepared the samples for the cryo-EM measurements supervised by C.E.M., M.G. and G.H. G.H. and M.G. analyzed the cryo-EM data. J.N., G.H. and C.E.M. wrote the manuscript with support from and approval by all coauthors. C.E.M. designed the study with input from V.N., S.W. and J.N. and supervised it.

## Funding

## Competing interests

The authors declare no competing interests.
