## [Transparent Peer Review File · Nature Communications]

Discovery of an allosteric binding site for anthraquinones at the human P2X4 receptor

Corresponding Author: Professor Christa Mueller

Version 0:

Reviewer comments:

Reviewer #1

(Remarks to the Author)

The manuscript by Nagel et al. presents investigation in a proposed hitherto unidentified allosteric site in the P2X4 receptor targeted by anthraquinones. The authors perform a small SAR study based on Cibacron Blue, a relatively weak allosteric modulator of all P2X receptor and based on docking studies and mutagenesis they identify an intermolecular lock formed by Glu307 and basic residues in P2X4 to be responsible for the moderate potencies of these modulator at the P2X4, as the compounds are found to be potent allosteric inhibitors of a P2X4-E307T mutant. The authors then obtain cryo-EM structure of the human P2X4-E307T extracellular domain (ECD) in complex with the compound PSB-0704, and this co-structure is compared with other P2X structures and used to extract information of the binding mode of this compound series to P2X4. The authors also point out that this is one of few human/mammalian P2X receptor structures, and finally they propose that the findings in this work could form the basis for rational drug design in the field.

While this reviewer is not an expert in the P2X field, the study and its findings are communicated in a largely clear and understandable manner. The data presented in the manuscript appear to be original and novel, the chosen techniques and assays to obtain the results applied appear to be sound and performed well, and the obtained data appear to have been analyzed correctly. In most cases interpretations of the results are sound and reasonable based on the available data. However, some important conclusions drawn by the authors would greatly benefit from or straight out require additional experimentation, and some proposals made in the manuscript do not seem to be supported by data and/or would require more discussion.

In conclusion, this is a nice study that provides a high-resolution structure of the human P2X4 receptor ECD in complex with an allosteric modulator, identifies a novel allosteric site in the receptor and provides a plausible explanation for the modest potencies of modulators targeting this site in the wild-type receptor. However, not all steps in an overall interesting and nicely structured story are equally well supported by data and alternative interpretations of the obtained data could have been better presented in some cases. Moreover, the use of a high-affinity binding receptor mutant for the cryo-EM structure and the interpretations for the wildtype receptor based on that also comes with a price, and this reviewer is not as convinced about the informative value of some findings for future drug design efforts in this field as the authors seem to be. Thus, this reviewer is not convinced that the study in its present form warrants publication in Nature Communications.

SPECIFIC COMMENTS

“Introduction”

- Introduction overall: The Introduction section nicely provide the background for the study. However, the section could perhaps be a little improved in terms of the fluidity of the presentation, where the different subsections are coming a little abruptly and perhaps could be more seamlessly linked together.
- Lines 48-51: While it is relevant to mention the marketed P2X3 drug, it cannot be claim to demonstrate the therapeutic potential in all P2X receptors.
- line 65: Reference 27 (D'Antongiovanni et al., 2022) has been retracted by the journal it was originally published in, and thus it should not be used in this manuscript.
- line 74: The structures probably confirm the “trimeric” structure of the P2X receptors, not the “homotrimeric”.
- Lines 83-86: Repeatedly in the manuscript, the authors make a point out of their structure being of the human P2X4 ECD. If

they want to stress this point, it is important to provide some context with regard to how important this actually is in light of the numerous other P2X structures published. For example, what is the homology between the human and the zebrafish P2X4 receptor sequences, and what are the homologies of the human P2X4 receptor with the mammalian (human, rat, panda) P2X receptors for which structures have been published? Is this first structure of the human P2X4 ECD really a major step forward in itself?

- Lines 97-98: As commented on in more detail later, I am not convinced that this work provide a lot to the foundation of rational ligand design for P2X4 or other P2X receptors. The use of the word “may” in the phrasing here does not really promise that either, but if the authors wish to claim this they would need to provide more substance to this claim in their work.

“Results”

- Results overall: As for the Introduction section, also in the Results section, the authors could be better of creating a red thread through the section, i.e. connecting the subsections. For example, the jump from section 118-126 to section 156-164 could be linked better.

- Lines 105-108 / Fig 1b,c: The authors find Cibacron Blue to exhibit a biphasic modulator profile at P2X4 and a “clean” NAM profile at P2X2, using an ATP EC80 at both receptors. It would have been appropriate also to study the effects of the drug at both receptors using a lower ATP concentration, for example EC20, both to elucidate the potentiation mediated by low concentrations of the drug at P2X4 better, but also to provide better possibility to spot potential “low-efficacious” potentiation at P2X2.

- Lines 118-126 / Fig 1e,f / Suppl. Tables 2-3:

- The authors characterize Cibacron Blue at P2X2/4 chimeras and present the findings for the drug as inhibitor of these chimeras. While I assume that this mean that the drug does not display a biphasic curve at any of these chimeras, this should be stated explicitly - the authors should also provide representative concentration curves for the drug at the other chimeras (in the supplementary information).

- It would be appropriate to provide data for ATP at all chimeric receptors as well as for WT P2X2 in Supplementary Table 2. In lines 124-126 the authors claim that the EC50 of ATP “remained unaltered” at the c10 chimera. I assume the authors means that it is unaltered compared to WT P2X4? If that is the case, that is not true, since there is a statistically significant difference between the EC50 at these receptors. Thus, I suggest that the authors provide all these data and are more precise with their summary in the text.

- Suppl. Table 3: What is the reason for the n=5 for Cibacron Blue at the c10 chimera compared to n=3 (or 2) for the other receptors? With these different n-numbers is it still warranted to do a one-way ANOVA?

- Lines 156-224 / Figs. 2-4:

- Line 157: It should be Lys298, not Lys289.

- In their focus on the Alpha3 model of P2X4, the authors describe two interactions: the proposed ionic lock between E307 and R82 and K289 and a putative interaction between R301 and N110. The second interaction is not studied at all by mutations, and there is no mentioning of it after this section. Could this putative R301-N110 interaction not be interesting to investigate in connection to the modulator binding as well? I understand that R301 may form interactions with the bound modulator in the P2X4-E307T mutant (structure), but it would still be informative to mutate either N110 or R301, or both in two single mutants.

- The use of the term “intermolecular interaction” for E307 and R82 and K298 is technically not completely correct. While E307 and R82 are in the same subunit, K298 is in another. Thus, E307 is acting through both intra- and intermolecular interactions.

- Lines 190-201 / Figs. 2-4:

- While I understand that the E307T mutation is to validate the proposed ion-lock, why was the two aromatic residues (R82 and K298) not mutated? I understand that R82 may form interactions with the bound modulator in the P2X4-E307T mutant (structure), but it would still be informative to mutate this residue. And K298 does not seem to be involved directly in modulator binding to the P2X4-E307T mutant, so a similar increase in modulator potency at a K289 mutant would further substantiate the importance of the ionic lock. Although K298 is conserved in P2X2, K298 in P2X4 could still be mutated to something else.

- Since this putative ionic lock E307/R82-K298 lock in P2X4 of proposed importance for the low modulator binding is unlikely be established in P2X2 (corresponding residues Thr/Lys-Lys), how does the authors then explain the low modulator potency at P2X2. While it may be because the residues directly involved in modulator binding in P2X4-E307T all not all conserved in the corresponding pocket of P2X2, this aspect would be appropriate to address either in the Results or Discussion section.

- Neither the rationale behind nor the subsequent conclusions from the D302I and Q308T mutations included in the study are not outlined in the manuscript. Both residues are located in the same v10 segment as E308, be other residues in this loop are also not conserved between P2X4 and P2X2, and none of the two residues are candidates for the ionic lock or for modulator binding interactions based on the alpha3 model. As for Q308T, it could argued to serve as a “control mutation” in comparison with the neighboring E307T mutation, but the rationale for the D302I is not clear. Firstly, it is not clear which putative role this residues is believed to have. If it as the neighbor residue to the R301 – then why not just mutate R301? Secondly, this D302I mutation can not be argued to be a directly replacement mutation, since the tripeptide DLA segment in P2X4 in P2X2 is a dipeptide IN segment, meaning that the spatial position of the isoleucine in P2X4-D201I is bound to be different that the Ile in the IN dipeptide in P2X2. Finally, the fact that also the D302I has a substantial effect on Cibacron Blue potency as well (ca. 9-fold increased potency) is merely just stated, not put into any context. For this reader, the inclusion of these two mutations in the study and the take-home messages for them is somewhat of an enigma.

- Lines 216-217 / Figs. 2-4: To sum up my comments to the two subsections above, the rationale for the mutants included in and the conclusions drawn from this mutagenesis study should be explained and outlined much better to the reader. There are several obvious mutations not included in the study that would be extremely valuable to include to substantiate conclusions, two of the three mutations tested are not presented or enterpreted for the reader, and in general I encourage

the authors to provide more experimental data to both substantiate but also to challenge the validity of their proposal about the importance of the ionic lock and/or whether other/additional molecular mechanisms could be at play for the low potency of these modulators at wild-type P2X4.

- Lines 217-224 / Table 1: The small SAR study performed presents some interesting SAR observations, but in some cases a more precise phrasing is needed, for example the SAR study should be focused on the five derivatives only, since Cibacron Blue cannot be compared directly to these because there are more than one structural difference.
- It is correct when the authors conclude that the two acidic groups in ring C and D are beneficial for modulator binding, this is clear from comparisons of 0739 with 0801 and 09059. The acidic group in ring D of Cibacron Blue is in another position than in the five derivatives.
- The authors can NOT conclude that the F ring of Cibacron Blue is dispensable based on the provided data, because the Cibacron Blue structure can not be compared 1:1 to those of the five derivatives, including that that of PSB-09059: 1) Cibacron Blue contains a D ring acidic group in another position than the derivatives, and 2) Cibacron Blue contains a different ring system as its E ring compared to the phenyl E ring in the derivatives.
- Are the authors confident that none of the five derivatives did not exhibit a biphasic modulator profile at WT P2X4? This should explicitly be stated in the Results, and it could also be interesting to discuss this difference between the compounds in the Discussion.
- Lines 332-333: In the obtained P2X4-E307T structure, only the ECD of the receptor could be modeled, whereas the TMD was to be disordered. While this is understandable, it does mean that the structure of this receptor domain can not be seen, and the ECD structure could be influenced by the structure of the TMD of the receptor. The authors should point this out and the potential implications of this should be discussed.
- Lines 341-347: The mutations of two asparagines in the cryo-EM construct is described in the methods section, it is mentioned here, and it is also mentioned in the Discussion. Maybe omit one of the two latter mentionings from the manuscript?
- Lines 357-359: I assume that that the authors do not see a density in the ATP binding site in the P2X4-E307T structure?
- Lines 359-361: This observed difference in compared to the zebrafish structures are mentioned but not discussed further. Do the author think it is the presence of the modulator in the binding site, the absence of the ionic lock, or something else?
- Lines 382-390: The authors should confirm/substantiate the proposed binding mode of PSB-0704 by mutagenesis. It would be obvious to introduce R301A and R82A mutations into P2X4-E308T and characterize PSB-0704 at these mutants. Some of the hydrophobic interactions with aromatic residues in the binding site could also be probed.
- Lines 392-403: Based on their comparison of the P2X4-E308T/PSB-0704 structure and the AF3 WT P2X4 model, the authors ascribe the low modulator binding to the WT receptor to the ionic lock E308/R82-K298 in the WT receptor that “blocks a significant part of the compound binding site”. However, these modulators do bind to and inhibit the WT P2X4, so clearly the modulators can access the binding site – or parts of it? Is the reduced potency of the modulators thus due to a less optimal binding conformation of them at this binding site in the WT receptor when they access it? The authors should be more precise as to what this comparison shows.

“Discussion”

- Discussion overall: The Discussion could benefit from a very extensive revision. A LOT of the subsections in the Discussion are characterized by containing a lot of summary of either things from the Results section or things better presented in the Introduction section. For example, much of lines 430-454, lines 456-474. The authors should review the section carefully for this, and make the Discussion section more concise and focused on discussion of the results.
- Fig. 10: 1) It would be helpful for the reader if the position of the different binding sites in the structure and the detailed images of each binding site were linked with arrows or boxes. The different coloring of the ligands is not sufficient to identify their positions in the structure. 2) If the authors want to make the point that the binding site for these modulators is indeed novel, a part of this figure should be a closer look at this region and the binding sites for PSB-0704 and the other modulators targeting this region (Suppl. 6d).
- Lines 526-538: The discussion of the biphasic profile of Cibacron Blue as a modulator does not lead to much insight. This reviewer is somewhat skeptic about the first proposed scenario (1-2 modulator molecules bound leads to potentiation, three leads to inhibition) explaining this concentration-dependent effect. It would require that low concentrations of Cibacron Blue would bind but not saturate all three sites of the receptor, but only at higher concentrations would be able to bind all three sites – i.e. that the binding site for the second/third of the three sites in the trimer was significantly lower than the first one or two sites – a sort of reverse-allosteric phenomenon? Or how does the author envision this scenario?
- Lines 540-546: I am not quite sure what the authors are suggesting here. It is true that the P2X4-E307T/PSB-0704 pair could be used in explorations of the P2X4's role in physiology, for example by construction of a P2X4-E307T knock-in mice and the use of the compound (provided that the bioavailability and PK properties of PSB-0704 is suited for in vivo use). However, it would have to be a modified system, such as a knock-in animal, to enable such studies. The authors should try to be more specific about what they mean.

“Conclusions”

- I am not as convinced as the authors that this study will be useful for drug design. Even if it is argued that this study has demonstrated the importance of an ionic lock for the low potencies of modulators targeting this site in the WT receptor, the strategy proposed by the authors to use this insight for drug design is highly speculative: the design of compounds that “disrupts the intermolecular ionic lock by interacting with Glu307” and thus can allow the “regular” modulators to do their job. Even if one were able to develop compounds that potently can bind and disrupt this ionic lock, it would mean that these compounds would bind very close to the modulator binding site, and they would have to do so within changing the ability of this adjacent site to bind the modulator. That seems like quite a tall ask...
While I appreciate that the authors wish to convey potential possibilities in their findings, I think this proposal is too theoretical and speculative, and I would suggest the authors that they refrain from going down that road. The claim that these

results “provide a basis that will enable and promote future drug research” is quite a stretch.

Reviewer #2

(Remarks to the Author)

In this work the authors have investigated the structure-activity relationship (SAR) of Cibacron Blue (CB) and anthraquinone derivatives at human P2X₄, showing that CB exhibits a biphasic concentration-response, and showing that the P2X₄-P2X₂ receptor chimera (with region R301-Q308 swapped), and subsequently the mutant E307T, display significantly improved CB potency. The authors use AlphaFold modelling to propose that the residue E307 may form an ‘ionic lock’ with basic amino acid residues which may prevent the binding of anthraquinone derivatives to P2X₄ with high affinity to explain their findings. They go on to describe the cryoEM structure of the extracellular domain of the human P2X₄ receptor mutant E307T with the anthraquinone derivative PSB-0704 bound, delineating a novel allosteric pocket, which partially overlaps with the BAY-1797 pocket described for zebrafish P2X₄. The location of the pocket is consistent with functional data, suggesting that PSB-0704 forms ionic interactions with R301 and R82, and that in wild-type P2X₄, E307 prevents access of PSB-0704 to the binding pocket.

In my opinion this is a very interesting study, and the work describing CB mode of action, SAR and the potential molecular interpretation is carefully performed. The 3D-structure of the hP2X₄ E307T extracellular domain bound to PSB-0704 is relatively low-resolution, and although it appears plausible, I have reservations about the cryoEM analysis and interpretation, and how useful it might be for drug discovery. My specific major and minor comments are listed below.

Major comments

1. It is not clear from reading the abstract or introduction that the structure reported is the extracellular domain only (even though the full-length protein was used for structure determination), or that the structure represents a mutation (E307T) introduced to increase antagonist potency (see my minor comments 1 and 2 below). Both these important points should be emphasised.
2. Results, page 16-17: The reported cryoEM structure lacks information in the transmembrane region, which to me is surprising, and is also in contrast to several recent cryoEM structures of P2X receptors [Additionally, a human P2X₄ cryoEM structure (reported on bioRxiv in July 2024; doi: 10.1101/2024.07.25.605151) is in preparation at a reported resolution of 2.3 Å which appears to show well-defined structure in both the transmembrane and cytoplasmic domains]. The only comment from the authors on this surprising result in this manuscript is that ‘The transmembrane helices were disordered and could not be modelled.’ What does this say about the quality of the particle dataset (or the subsequent analysis), and are the authors not concerned about this? I appreciate that preferred orientations were an issue, but what steps were taken in sample preparation to address this? I also appreciate that local refinement was used to generate the maps enabling fitting of the extracellular domain, and that this is the location of the binding site for the antagonist and thus the focus of data interpretation, but how confident can the authors be about this structure where a key part of it (that surely one would expect to be relatively ordered in the same way as the extracellular domain if the protein is folded properly) is absent in the reconstruction?
3. Results, page 17: The resolution of the reported structure is relatively low (compared to recent cryoEM structures of other P2X receptors), at 3.35 Å. However, the authors state ‘The antagonist PSB-0704, which had been added during protein purification and again immediately before the cryo-EM grid preparation, could be placed into this map feature with confidence’. How was the PSB-0704 ‘placed’ into the map – was this fitted unambiguously by software, or manually by the authors with reference to their mutagenesis data? Is there more than one possible conformation for this molecule that would fit the density, and how were alternate conformations excluded? Was there any potential bias in the fitting process? There is no description of how the PSB-0704 was fitted to the structure in the Methods.
4. Supplementary Figure 4: Several of the 2D class averages appear to contain more than one particle – there are clear separate blobs of density. Presumably the reprojections from the 3D structure will not match these classes as the reconstruction was done in a different way, but it would be good to see how well they correlate. Were the 2D classes in Panel a all used to generate the map with preferred orientations that was difficult to interpret?
5. Results, page 9: Is there a reason why the authors did not also make and test the R301K mutation?
6. Discussion, page 23: ‘our structure was obtained of the full-length human P2X₄ receptor construct...’ I think it is very important to state that you used the full-length protein, but your structure is only of the extracellular domain.
7. Discussion, page 24: ‘Sufficiently high overall resolution’. This is not the overall resolution of the full-length protein, just of the extracellular domain, and it is also a low resolution compared to recent cryoEM structures of P2X receptors. I think it is important to discuss the potential shortcomings of the structure obtained, and of fitting ligands at this relatively low resolution (relates to my point 3 above). There is a general lack of detail on these points in the Methods, Results and Discussion.

Minor comments

1. Abstract: ‘structure of full-length human P2X₄’. I don’t think it is reasonable to state that the determined structure is full-length, because of the issues with reconstruction and the disorder of the transmembrane domains. It might be better to state ‘structure of the extracellular domain of the human P2X₄ mutant E307T in complex with the anthraquinone derivative PSB-0704 was determined from cryoEM analysis of the full-length protein at a resolution of 3.35 Å.’
2. Introduction: I think it would be better to state ‘structure of the extracellular domain of the human P2X₄ mutant E307T, derived from cryoEM analysis of the full-length protein’.
3. Results, page 7: Why not also make an AF3 model of your chimeric receptor and analyse the intersubunit contacts in the same region?
4. Supplementary Table 1: It would be good to report the resolution of each structure in this table, and also provide details on the truncations and mutations introduced in each construct.

Reviewer #3

(Remarks to the Author)

Summary:

The manuscript explores the structural basis of P2X4 receptor antagonism, focusing on anthraquinone derivatives like PSB-0704. Using cryo-EM, mutagenesis, and SAR studies, the authors identify a novel allosteric binding site on the human P2X4 receptor, distinct from previously known sites. High-resolution cryo-EM data reveal ligand interactions and receptor conformations, while complementary approaches validate the findings. Insights into PSB-0704's molecular interactions offer a foundation for developing potent, selective P2X4 antagonists. The compounds used in the SAR study were synthesised and characterised in prior work by the research group.

Major Weaknesses:

- While the study mentions the biphasic nature of Cibacron Blue's activity, the underlying mechanisms remain speculative. Further experiments or additional discussion of this phenomenon would strengthen the manuscript.
- The study focuses primarily on engineered receptor constructs (e.g., E307T mutant). The authors should discuss how these findings translate to the wild-type receptor in physiological contexts.
- While the structural and SAR data are robust, additional functional assays, such as electrophysiological recordings for PSB-0704 at the wild-type receptor, would enhance the manuscript.
- A more in-depth comparison of PSB-0704 with other P2X4 antagonists (e.g., BX430, BAY-1797) in terms of binding affinity, potency, and pharmacokinetics would be beneficial.

Minor Suggestions:

- Provide additional details on cryo-EM data acquisition and processing, especially regarding the resolution and interpretation of the binding site.
- Specify the rationale for selecting specific anthraquinone derivatives for SAR analysis.
- Conclude with a more explicit discussion of how the findings can inform the design of hybrid compounds or other allosteric modulators.

Recommendation:

I recommend major revisions before acceptance. The study has substantial merit and presents novel findings that significantly advance the field of P2X receptor pharmacology. Addressing the concerns highlighted above would greatly improve the manuscript's impact and clarity.

Version 1:

Reviewer comments:

Reviewer #1

(Remarks to the Author)

In their revised manuscript, Nagel et al. have more or less addressed all of my comments to and concerns about the original manuscript. I thank the authors for their responsiveness to my comments, and I congratulate them with a nice piece of work. Thus, the following comments should mainly be seen as suggestions to the authors to consider in order to further improve their manuscript.

1. Most importantly from my perspective, the revised manuscript contain data for several additional mutants (of the residues R82, K298, R301 and N110) investigating the binding pocket for the anthraquinones in the P2X4 receptor. With the new data, the proposal of an ionic lock and its important for modulator binding to the site is much more substantiated and convincing.

A minor thing for the authors to consider in this respect is how to interpret the effects of the mutations of R301 and N110. While the R301A mutation is detrimental to antagonist potency, N110A is not. Thus, the authors may want to make it more clear in the manuscript that this could suggest that there is not an interaction between R301 and N110 important for antagonist binding, but maybe rather that the importance of R301 is rooted another mechanism, such as putatively interacting with E307? The authors could consider making this more clear in the manuscript, with the new data in hand?

2. In the revised manuscript it is noted that a very recent cryo-EM structure of the human P2X4 receptor (with modulator BAY-1797 bound) has been published. This has prompted the authors to study the effects of some of their mutants on this and two other modulators as well. However, I would suggest that the authors also use this new structure to compare with their own, and assess whether the overall structural aspects of two structures are similar and support each other.

3. One of the few comments where I am left a little underwhelmed by the authors' response is to the puzzling weak antagonist potency of the anthraquinones at P2X2. While the statement added by the authors to the revised manuscript certainly could be correct, i.e. that the binding site for the anthraquinones in P2X2 only facilitates low-affinity binding of them in contrast to the high-affinity binding to the P2X4-E307Q mutant. Nevertheless, in light of the compounds displaying comparable IC50 values at WT P2X2 and WT P2X4, which for the latter is converted to potent antagonism by the removal of the ion lock by the E307Q mutation, it is nevertheless strange that the binding sites in the two receptors lacking the ionic lock (WT P2X2 and P2X4-E307Q) are so different when it comes to interactions with the anthraquinones. I think the authors should consider going into more depth with this discussion, and in light of their structure maybe identify some of these

P2X2/P2X4 differences responsible for these vastly different antagonist properties.

4. As for the extent to which structures of targets can be used for medicinal chemistry development: I certainly share the authors' opinion on the benefits and insights provided by these structures for ligand design. My comment in this regard was more to the specific extent that the authors use these structures. I am certainly aware that ligand binding can break "ionic locks" in receptors and in this way influence receptor signaling, and I also agree with the authors that the insight into the ionic lock in P2X4 for modulator binding to this site is interesting and good to have. It is when the authors repeat their proposal of a solution for this "ionic lock" that I become skeptical (revised manuscript; lines 655-659). I think that is very speculative that the introduction of a basic moiety in the modulator to interact with E307 would break the ionic lock, at least in a way where the modulator could bind to the site. If the ionic lock prohibits the modulator to access the binding site, this basic moiety would have to interact with E307 from the outside and next allow for the modulator to enter the binding site without this ionic lock being reformed at any point? I think it sound very speculative.

Reviewer #2

(Remarks to the Author)

I thank the authors for their thorough revision of the manuscript, and for addressing all my comments.

Reviewer #3

(Remarks to the Author)

Thank you for your thorough revision. The changes you have implemented are clear, well-considered, and improve the overall quality and clarity of the manuscript. I am satisfied with the responses provided and find no further issues that require attention.

Version 2:

Reviewer comments:

Reviewer #1

(Remarks to the Author)

I thank the authors for the changes implemented in this second revision in response to my comments. I do not have any major objections to things in the revised manuscript.

Response to Editor and Reviewers

Dear Editor, dear Reviewers,

we are most grateful for your expert comments and critique, which we have tried to implement in the revised version of the manuscript as detailed below. We believe that your comments and suggestions have led to a considerable improvement of the manuscript. In particular, we have performed new, additional experiments (see below), which support our results and conclusions, and added Annette Nicke's group in Munich, an expert in P2X receptor ion channels, who performed two-electron voltage clamp studies on human wildtype and mutant P2X4 receptors expressed in *Xenopus* oocytes as an additional method for pharmacological characterization of antagonists and mutants. Further, we have revised the processing of our cryo-EM dataset, which was previously focused on the ligand binding site. We now added a new branch to our processing tree, where we refrained from local refinements and succeeded in reconstructing a significant part of the transmembrane domains.

Reviewer's Comments:

Reviewer #1 (Remarks to the Author)

The manuscript by Nagel et al. presents investigation in a proposed hitherto unidentified allosteric site in the P2X4 receptor targeted by anthraquinones. The authors perform a small SAR study based on Cibacron Blue, a relatively weak allosteric modulator of all P2X receptor and based on docking studies and mutagenesis they identify an intermolecular lock formed by Glu307 and basic residues in P2X4 to be responsible for the moderate potencies of these modulators at the P2X4, as the compounds are found to be potent allosteric inhibitors of a P2X4-E307T mutant. The authors then obtain cryo-EM structure of the human P2X4-E307T extracellular domain (ECD) in complex with the compound PSB-0704, and this co-structure is compared with other P2X structures and used to extract information of the binding mode of this compound series to P2X4. The authors also point out that this is one of few human/mammalian P2X receptor structures, and finally they propose that the findings in this work could form the basis for rational drug design in the field.

While this reviewer is not an expert in the P2X field, the study and its findings are communicated in a largely clear and understandable manner. The data presented in the manuscript appear to be original and novel, the chosen techniques and assays to obtain the results applied appear to be sound and performed well, and the obtained data appear to have been analyzed correctly. In most cases interpretations of the results are sound and reasonable based on the available data. However, some important conclusions drawn by the authors would greatly benefit from or straight out require additional experimentation, and some proposals made in the manuscript do not seem to be supported by data and/or would require more discussion.

In conclusion, this is a nice study that provides a high-resolution structure of the human P2X4 receptor ECD in complex with an allosteric modulator, identifies a novel allosteric site in the receptor and provides a plausible explanation for the modest potencies of modulators targeting this site in the wild-type receptor. However, not all steps in an overall interesting and nicely structured story are equally

well supported by data and alternative interpretations of the obtained data could have been better presented in some cases. Moreover, the use of a high-affinity binding receptor mutant for the cryo-EM structure and the interpretations for the wildtype receptor based on that also comes with a price, and this reviewer is not as convinced about the informative value of some findings for future drug design efforts in this field as the authors seem to be. Thus, this reviewer is not convinced that the study in its present form warrants publication in Nature Communications.

Thank you for your overall positive comments. As suggested, we now provide additional experiments to support our findings and conclusions, which we hope will convince you - for details, see below.

SPECIFIC COMMENTS

“Introduction”

- Introduction overall: The Introduction section nicely provide the background for the study. However, the section could perhaps be a little improved in terms of the fluidity of the presentation, where the different subsections are coming a little abruptly and perhaps could be more seamlessly linked together.

Thank you for pointing this out. We have now worked on the fluidity of the text to avoid abrupt changes of topics, rearranged part of the text, and better connected the paragraphs.

- Lines 48-51: While it is relevant to mention the marketed P2X3 drug, it cannot be claim to demonstrate the therapeutic potential in all P2X receptors.

We agree. The sentence has now been rephrased:

“Thus, P2X receptors have emerged as attractive drug targets; in fact, the P2X3 receptor antagonist Gefapixant has recently been approved for the treatment of chronic cough^{10,11}.”

- line 65: Reference 27 (D’Antongiovanni et al., 2022) has been retracted by the journal it was originally published in, and thus it should not be used in this manuscript.

Thank you for this important comment! We removed the reference.

- line 74: The structures probably confirm the “trimeric” structure of the P2X receptors, not the “homotrimeric”.

Agreed and corrected.

- Lines 83-86: Repeatedly in the manuscript, the authors make a point out of their structure being of the human P2X4 ECD. If they want to stress this point, it is important to provide some context with regard to how important this actually is in light of the numerous other P2X structures published. For example, what is the homology between the human and the zebrafish P2X4 receptor sequences, and what are the homologies of the human P2X4 receptor with the mammalian (human, rat, panda) P2X receptors for which structures have been published? Is this first structure of the human P2X4 ECD really a major step forward in itself?

Thank you for your comment. Species differences can in fact play an important role in many receptors including the P2X4 receptor. We have recently published a study, comparing potencies of selected agonists and antagonists at human, monkey, dog, guinea pig, pig, rabbit, rat, mouse, and zebrafish P2X4 receptors, in which we show and discuss species differences and compare the sequences (Nagel J, Bous C, Abdelrahman A, Schiedel AC, Müller CE. Species Differences of P2X4 Receptor Modulators. ACS Pharmacol Transl Sci. 2025 Jan 30;8(5):1320-1332. doi: 10.1021/acsptsci.4c00688).

We have now added the following sentence:

“However, significant P2X4 receptor species differences are known to exist (doi: 10.1021/acsptsci.4c00688), and very recently, the first cryo-EM structure of the human P2X4 receptor bound to BAY-1797 was published.”

- Lines 97-98: As commented on in more detail later, I am not convinced that this work provide a lot to the foundation of rational ligand design for P2X4 or other P2X receptors. The use of the word “may” in the phrasing here does not really promise that either, but if the authors wish to claim this they would need to provide more substance to this claim in their work.

Being a medicinal chemist, I have realized that structural information is of great importance for drug design and development. This is the main reason, why my group embarked on structural biology projects: We want to know where our lead compounds bind, which interactions between receptor and ligand /antagonist are formed, and whether there is space for additional interactions, that can be utilized by designing molecules that show increased potency. At a recent medicinal chemistry conference that I attended, structural biology was used as a basis or at least to support each of the presented drug development projects.

The more cautious “may” was replaced by “are expected to”:

“They are expected to provide a basis for the development of future drugs.”

We have now added more substance to the claim in the Discussion section, which has been thoroughly revised as suggested.

Some text in Discussion regarding structure-based drug design:

“All of these and further mutations are in agreement with the determined structure supporting our ionic-lock hypothesis on the wt P2X4 receptor. Together with the SARs at the wt and the E307T mutant receptor (Table 1), it can be concluded that the anthraquinone derivatives interact with the same binding site on both mutant and wt receptor (also see Supplementary Fig. 8 and 9 for docking of anthraquinone derivatives into the human wt and mutant- P2X4 receptor). The determined structure, although not obtained for the wt P2X4 receptor, is therefore useful for drug design. It rationalizes the SAR data showing, for example, that the amino group at ring C is not important for the interaction with the P2X4 receptor (Table 1). In our structure, the amino group is ~4 Å away from the hydroxyl group of Tyr300, which is too far to form a direct hydrogen bond (Fig. 9). Aromatic amines (anilines) are regarded as potential toxicophores⁶⁷, and therefore these insights can be used to improve the compounds not only with regard to potency, but potentially also with regard to toxicity. The oxygen atoms of the quinone ring B have not shown any polar interactions with the receptor and thus are likely also not required. This opens up the possibility to design antagonists without the redox-reactive quinone structure by replacing it with a more stable moiety.

Since the hypothesis of ionic lock formation has been confirmed, this offers the possibility to design compounds that disrupt the intermolecular ionic lock by interacting with Glu307, to open up a larger, high-affinity binding site for antagonists. Introduction of a basic moiety into the structure of PSB-0704, e.g. attached to the anthraquinone 1-position instead of or linked via the anilinic amino group, could be a plausible strategy. In fact, disruption of ionic locks is a common mechanism in drug-receptor interactions. For example, in G-protein-coupled receptors ionic locks stabilize the inactive state of the receptor^{72,73}. They typically act as molecular switches, which can be disrupted by drugs⁷⁴."

"Results"

- Results overall: As for the Introduction section, also in the Results section, the authors could be better of creating a red thread through the section, i.e. connecting the subsections. For example, the jump from section 118-126 to section 156-164 could be linked better.

Thank you for pointing this out. We have now better connected the sections.

- Lines 105-108 / Fig 1b,c: The authors find Cibacron Blue to exhibit a biphasic modulator profile at P2X4 and a "clean" NAM profile at P2X2, using an ATP EC80 at both receptors. It would have been appropriate also to study the effects of the drug at both receptors using a lower ATP concentration, for example EC20, both to elucidate the potentiation mediated by low concentrations of the drug at P2X4 better, but also to provide better possibility to spot potential "low-efficacious" potentiation at P2X2.

Excellent suggestion. We repeated the experiment at the wt P2X4 receptor with an EC₂₀ of ATP (59 nM) and observed a similar biphasic curve for Cibacron Blue. At the P2X2 receptor, using an EC₂₀ of ATP (345 nM), no clear activating effect was observed. We observed biphasic modulation at the chimeric P2X4(P2X2) receptor c6 and added the following information to the manuscript:

"At chimeric P2X4(P2X2) receptors c2-c5^{C116-T186}, c7^{I218-D224}, and c8^{R265-L269}, Cibacron Blue showed inhibitory potency in the micromolar range (Supplementary Table 3 and Supplementary Figs. 4 and 5). The antagonist displayed a biphasic modulation of the chimeric P2X4(P2X2) receptor c6^{N208-S216}, similar to the modulation observed at the P2X4 receptor (Supplementary Table 3 and Supplementary Fig. 5), while it showed monophasic inhibition at the other investigated chimeric receptors."

"In contrast to Cibacron Blue, none of the other investigated anthraquinone derivatives showed a biphasic curve at the wt P2X4 receptor in calcium influx assays under the applied conditions."

Discussion:

Our results do not explain the biphasic behavior of Cibacron Blue at the human P2X4 receptor. It has been observed previously that some allosteric modulators can show receptor activation or inhibition depending on concentration⁵⁶. Based on the present structural data it is tempting to speculate that the degree of downward movement of the head domain induced by a single molecule of the relatively large Cibacron Blue could account for the potentiating effect observed with low concentrations of the ligand. Thus, only Cibacron Blue can induce a "pre-open" state of the receptor and consequently higher agonist efficacy, even if only two ATP molecules can bind. If, however, two or more antagonist molecules are bound, ATP cannot bind or induce the required conformational change and the receptor is blocked. Another explanation could be the presence of different binding sites to which the modulator has different affinity, as postulated for some pentameric ligand-gated ion channels and their allosteric ligands⁶⁹. This hypothesis is supported by the results obtained in radioligand binding studies using the

stable ATP analog [³⁵S]ATPgS, in which Cibacron Blue showed a biphasic competition binding curve³⁰. It has to be pointed out that this behavior is only observed for the large molecule Cibacron Blue consisting of 6 rings, but not for any of the truncated analogs, including PSB-0704, all of which only exhibit antagonistic activity, in calcium influx assays at recombinant 1321N1 astrocytoma cells as well as in TEVC experiments at *Xenopus* oocytes.

- Lines 118-126 / Fig 1e,f / Suppl. Tables 2-3:

- The authors characterize Cibacron Blue at P2X2/4 chimeras and present the findings for the drug as inhibitor of these chimeras. While I assume that this mean that the drug does not display a biphasic curve at any of these chimeras, this should be stated explicitly - the authors should also provide representative concentration curves for the drug at the other chimeras (in the supplementary information).

We have now included representative concentration-inhibition curves for Cibacron Blue at the chimeric receptors c2-c5, c6, c7, and c8 (Supplementary Fig. 5), as suggested, and inserted the statement above into the relevant paragraph.

- It would be appropriate to provide data for ATP at all chimeric receptors as well as for WT P2X2 in Supplementary Table 2. In lines 124-126 the authors claim that the EC₅₀ of ATP "remained unaltered" at the c10 chimera. I assume the authors means that it is unaltered compared to WT P2X4? If that is the case, that is not true, since there is a statistically significant difference between the EC₅₀ at these receptors. Thus, I suggest that the authors provide all these data and are more precise with their summary in the text.

We agree. As suggested, we now included data for ATP at all chimeric receptors as well as for the wt P2X2 receptor (Supplementary Figure 3 and Table 2)

The EC₅₀ value of ATP at the c10 receptor is 2.4-fold higher than at the wt P2X4 receptor. Thus, the potency can be considered to be in a similar range. We rephrased it to be more precise.

"The EC₅₀ value of the endogenous agonist ATP was found to be similar at the chimeric P2X4(P2X2) receptor (0.862 μM) c10^{R301-Q308} as compared to the wt P2X4 receptor (0.357 μM, Supplementary Fig. 3, Supplementary Table 2)."

- Suppl. Table 3: What is the reason for the n=5 for Cibacron Blue at the c10 chimera compared to n=3 (or 2) for the other receptors? With these different n-numbers is it still warranted to do a one-way ANOVA?

The experiments for Cibacron Blue at c10 were repeated more often because they were so surprising. We wanted to make absolutely sure that the value is correct. At the other receptor chimera, values were in the same range as at the wildtype receptor.

One-way ANOVA is generally suitable even with different numbers of experiments (unequal sample sizes) across groups if there are no severe outliers, which was not the case.

- Lines 156-224 / Figs. 2-4:- Line 157: It should be Lys298, not Lys289.

Corrected – thank you for spotting these errors.

- In their focus on the Alpha3 model of P2X4, the authors describe two interactions: the proposed ionic lock between E307 and R82 and K289 and a putative interaction between R301 and N110. The second interaction is not studied at all by mutations, and there is no mentioning of it after this section. Could this putative R301-N110 interaction not be interesting to investigate in connection to the modulator binding as well? I understand that R301 may form interactions with the bound modulator in the P2X4-E307T mutant (structure), but it would still be informative to mutate either N110 or R301, or both in two single mutants.

Thank you for your valuable suggestion. We have now investigated a number of additional mutants, including the suggested ones: R301A and R301K as well as N110A. These mutants were introduced into the hP2X4-E307T receptor, expressed in Xenopus oocytes and studied by two-electrode voltage clamp (TEVC) measurements (see below).

We investigated two different antagonists in these experiments, PSB-0704 (bearing two carboxylate functions) and its potent analog PSB-0739 (in which the carboxylate groups were replaced by sulfonate functions). Results on mutants were consonant for both antagonists.

While the antagonists showed decreased potency at the R301A mutant, potency was recovered in the R301K mutant, in which the basic amino acid Arg was exchanged for the basic Lys (homologous exchange).

In contrast, the antagonists retained their inhibitory potency at the N110A mutant.

These results are now reported in the manuscript, in addition to further investigated mutants and presented and discussed as follows:

“Two-electrode voltage clamp experiments and structure-based mutagenesis studies

Amino acid residues to be exchanged were selected based on the determined cryo-EM structure and the observed interactions. The wt P2X4 receptor and its mutants were expressed in Xenopus oocytes, and TEVC measurements were performed to determine inhibition of ATP-induced currents by PSB-0704 and its potent analog PSB-0739. The concentration-response curves of ATP determined at the wt and the E307T mutant P2X4 receptor were virtually identical (see Fig. 10a), and an EC₅₀ value of ca. 10 μM was determined. Subsequently, concentration-dependent inhibition of the ATP (10 μM)-induced current by PSB-0704 and PSB-0739 was measured at the P2X4-E307T receptor. Both compounds showed concentration-dependent inhibition with pIC₅₀ values of 5.79 (PSB-0704) and 6.49 (PSB-0739) (for current traces and inhibition curves see Fig. 10b,c). Thus, the inhibitory potency of the antagonists appeared to be much weaker in this assay under the applied conditions (preincubation for 20 s) than in the calcium influx assay, where the cells had been preincubated with the antagonists for 20 min. Subsequent preliminary experiments using different inhibitor concentrations, and increasing the incubation time (up to 2 min) indicated that prolonged preincubation led to an increase in potency, in particular for PSB-0704 (see Supplementary Fig. 10). A longer incubation time was not employed because current responses became unstable after longer periods without perfusion, and perfusion with antagonist for longer periods was limited by the availability of antagonist. Therefore, we subsequently tested the receptor mutants starting with antagonist preincubation for 2 min, followed by receptor activation with 10 μM of ATP (see Fig. 10). The more potent PSB-0739 was tested at two concentrations (300 nM and 1 μM) to observe concentration-dependency of the effects, while PSB-0704 was tested at a single, high concentration of 3 μM, based on preliminary dose-finding experiments.

Fig. 10 | Two-electrode voltage clamp (TEVC) experiments and structure-based site-directed mutagenesis studies. **a** Concentration-response curves for ATP at the wildtype (wt) human (h) P2X4 receptor and the hP2X4-E307T receptor mutant. Receptors were expressed in *Xenopus laevis* oocytes and current responses to 5s-pulses of ATP were analyzed by TEVC at -60 mV. ATP test concentrations and a reference concentration of 100 μ M ATP were alternately applied in 2 min intervals. Responses were calculated in relation to the reference concentration, fitted to the four-parameter Hill equation, and normalized to the maximal response. Data are presented as mean \pm SEM from 3–5 oocytes. **b** Representative current traces recorded in oocytes expressing the P2X4-E307T receptor mutant. Colored lines indicate the presence of the indicated antagonist during a 2 min preincubation and during the application of 10 μ M ATP (black line). 3 μ M PSB-0704 and 300 nM PSB-0739 were used. **c** Concentration-inhibition curves and chemical structures of the antagonists PSB-0704 and PSB-0739. Responses to 3-s pulses of 10 μ M ATP were recorded after 20 s superfusion with the indicated antagonist

concentrations and normalized to the preceding ATP response in the absence of antagonist. Data are represented as mean \pm SEM from 3–5 oocytes. **d, e** TEVC analysis of inhibitory effects of (d) PSB-0704 (3 μ M) and (e) PSB-0739 (300 nM and 1 μ M) on the indicated single and double P2X₄ receptor mutants expressed in *Xenopus laevis* oocytes. Inhibition of current responses evoked by 3s-pulses of 10 μ M ATP after 2 min superfusion with antagonist is shown. Current responses evoked by co-application of ATP and antagonist were normalized to the preceding ATP-responses in the absence of antagonist and effects were represented as percent inhibition. Data are presented as mean \pm SEM from at least 3 oocytes. The non-parametric Kruskal–Wallis test was applied, followed by Dunn’s multiple comparisons test. The significance levels were defined as follows: * $P < 0.05$, ** $P < 0.01$, *** $P < 0.001$, **** $P < 0.0001$; ns, not significant. For PSB-0739, the significance was evaluated for the lower concentration of 300 nM.

“The mutant E307Q was introduced into the wt P2X₄ receptor for comparison with E307T. At both mutants, the antagonists PSB-0704 and PSB-0739 showed strong inhibition, confirming that the replacement of the acidic glutamate at this position by a neutral amino acid residue disrupted the ionic lock and enabled binding and inhibition by the allosteric antagonists. Several further double mutants were investigated containing a mutation of an amino acid directly or indirectly involved in ligand binding, in addition to the E307T substitution (Fig. 10). Both antagonists showed similar effects indicating that they share the same binding mode and interactions. When the basic amino acid residues present in the allosteric binding site, R301, R82, and K298, were exchanged for alanine, the antagonists exhibited reduced potency at the resulting mutants. R82A had the largest effect – it forms an electrostatic interaction with the acidic group on ring D of PSB-0704 – and presumably also of PSB-0739. At the R301A mutant the effect was moderate, only being significant for the lower concentration of PSB-0739, but not for the higher concentrations of both antagonists. Homologous replacement in the R301K mutant did not alter antagonist potency. According to the cryo-EM structure, R301 forms an electrostatic interaction with the acid residue in the 2-position (ring C) of the anthraquinone, which is consistent with our mutagenesis results. K298 likely forms an interaction with Y299 of an adjacent receptor monomer thereby stabilizing the binding pocket, indirectly contributing to the interactions. The mutant N110A did not alter receptor inhibition by the antagonists. It was predicted to form a hydrogen bond with R301 in the wt P2X₄ receptor in the absence of an antagonist, but does not interact with the antagonist PSB-0704, while R301 binds to the antagonist. Finally, aromatic and lipophilic amino acid residues were exchanged. Unfortunately, the W84A and the F296Q mutants showed no proper expression. The F81A mutation had no significant effect indicating that aromatic interactions with ring E of the anthraquinone derivatives are not essential, but may be substituted by the lipophilic alanine. The I312F mutation resulted in only a minor, non-significant decrease in antagonist potency; phenylalanine is larger than isoleucine, but still tolerated.

Finally, we studied the inhibitory potency of the standard P2X₄ receptor antagonists BX430, BAY-1797 and 5-BDBD in ATP-induced calcium influx assays at the human P2X₄-E3207T receptor mutant expressed in 1321N1 astrocytoma cells. This was motivated by the fact that the newly discovered allosteric binding site for anthraquinones is adjacent to the previously discovered binding site for BX430 and BAY-1797. In fact, both compounds were several-fold more potent at the mutant as compared to the wt P2X₄ receptor, while 5-BDBD was equally potent at both the mutant and the wt receptor (Table 1 and Supplementary Fig. 11).”

Discussion

“Exchange of Glu307 for glutamine had virtually the same effect as its exchange for threonine (Fig. 10), both amino acid residues breaking the ionic lock. Mutation of each of the involved basic amino acid residues, Arg82, Lys298 and Arg301 to alanine in the P2X4-E307T receptor led to a reduction in inhibitory potency of the potent anthraquinones PSB-0704 and PSB-0739, while a charge conservation in the R301K mutant had no consequences (Fig. 10). (2) The mutation Q308T (Fig. 4) and N110A had no effects on the potency of the anthraquinones, and indeed, the structure showed no interaction. ...”

- The use of the term “intermolecular interaction” for E307 and R82 and K298 is technically not completely correct. While E307 and R82 are in the same subunit, K298 is in another. Thus, E307 is acting through both intra- and intermolecular interactions.

Yes, you are right – we apologize for this error. The description has now been corrected as follows:

“Based on the AF3 model, electrostatic interactions between Glu307 and basic amino acid residues Arg82, as well as Lys298 of an adjacent protein subunit, result in the formation of an intra- and intermolecular “ionic lock”, which is likely responsible for the moderate potency of Cibacron Blue and related anthraquinone derivatives at the wt P2X4 receptor.

• Lines 190-201 / Figs. 2-4:

- While I understand that the E307T mutation is to validate the proposed ion-lock, why was the two aromatic residues (R82 and K298) not mutated? I understand that R82 may form interactions with the bound modulator in the P2X4-E307T mutant (structure), but it would still be informative to mutate this residue. And K298 does not seem to be involved directly in modulator binding to the P2X4-E307T mutant, so a similar increase in modulator potency at a K289 mutant would further substantiate the importance of the ionic lock. Although K298 is conserved in P2X2, K298 in P2X4 could still be mutated to something else.

As suggested, we have now investigated a broader range of mutants (see above).

Among these were R82A (very strong reduction in antagonist potency), K298A (strong reduction in antagonist potency, but less than R82A). These data have now been included in the manuscript and are discussed. Thank you for these valuable suggestions.

- Since this putative ionic lock E307/R82-K298 lock in P2X4 of proposed importance for the low modulator binding is unlikely be established in P2X2 (corresponding residues Thr/Lys-Lys), how does the authors then explain the low modulator potency at P2X2.

While it may be because the residues directly involved in modulator binding in P2X4-E307T all not all conserved in the corresponding pocket of P2X2, this aspect would be appropriate to address either in the Results or Discussion section.

This is an excellent question! We created an overlay of our P2X4 receptor structure with an AF3 model of the hP2X2 receptor (not shown) and have now added the following discussion:

“Cibacron Blue is only weakly active at the wt P2X2 receptor although the receptor lacks an ionic lock. This is likely due to the fact that some of the residues directly involved in ligand binding are not present in the corresponding pocket of the human P2X2 receptor.”

- Neither the rationale behind nor the subsequent conclusions from the D302I and Q308T mutations included in the study are not outlined in the manuscript.

Both residues are located in the same v10 segment as E308, but other residues in this loop are also not conserved between P2X4 and P2X2, and none of the two residues are candidates for the ionic lock or for modulator binding interactions based on the alpha3 model. As for Q308T, it could be argued to serve as a “control mutation” in comparison with the neighboring E307T mutation, but the rationale for the D302I is not clear. Firstly, it is not clear which putative role this residue is believed to have. If it is as the neighbor residue to the R301 – then why not just mutate R301? Secondly, this D302I mutation can not be argued to be a direct replacement mutation, since the tripeptide DLA segment in P2X4 in P2X2 is a dipeptide IN segment, meaning that the spatial position of the isoleucine in P2X4-D201I is bound to be different than the Ile in the IN dipeptide in P2X2. Finally, the fact that also the D302I has a substantial effect on Cibacron Blue potency as well (ca. 9-fold increased potency) is merely just stated, not put into any context. For this reader, the inclusion of these two mutations in the study and the take-home messages for them is somewhat of an enigma.

The rationale was to replace amino acid residues in the human wt P2X4 receptor that were located in the variable region v10, in order to find out which residue(s) might be the reason for the high potency of Cibacron Blue at the chimeric P2X4(P2X2) c10 receptor.

We inserted the following explanation:

“The D302I and Q308T mutations essentially served as controls.”

“These results support our hypothesis that the amino acid residue Thr307 instead of glutamate at that position is particularly important for high-affinity binding of Cibacron Blue, while mutation of the other selected residues did not (Q308T) or only moderately contribute (D302I). It has to be mentioned that D302I in the P2X4 receptor does probably not align with Ile in the P2X2 receptor due to a gap in the sequence (see Fig. 3).”

- Lines 216-217 / Figs. 2-4: To sum up my comments to the two subsections above, the rationale for the mutants included in and the conclusions drawn from this mutagenesis study should be explained and outlined much better to the reader. There are several obvious mutations not included in the study that would be extremely valuable to include to substantiate conclusions, two of the three mutations tested are not presented or interpreted for the reader, and in general I encourage the authors to provide more experimental data to both substantiate but also to challenge the validity of their proposal about the importance of the ionic lock and/or whether other/additional molecular mechanisms could be at play for the low potency of these modulators at wild-type P2X4.

Thank you for your valuable comments. As suggested, we have now included an additional set of mutants and discuss them in detail. All of the data support our hypothesis and findings (for details see above).

- Lines 217-224 / Table 1: The small SAR study performed presents some interesting SAR observations, but in some cases a more precise phrasing is needed, for example the SAR study should be focused on the five derivatives only, since Cibacron Blue cannot be compared directly to these because there are more than one structural difference.

- It is correct when the authors conclude that the two acidic groups in ring C and D are beneficial for modulator binding, this is clear from comparisons of 0739 with 0801 and 09059. The acidic group in ring D of Cibacron Blue is in another position than in the five derivatives.

- The authors can NOT conclude that the F ring of Cibacron Blue is dispensable based on the provided data, because the Cibacron Blue structure can not be compared 1:1 to those of the five derivatives, including that that of PSB-09059: 1) Cibacron Blue contains a D ring acidic group in another position than the derivatives, and 2) Cibacron Blue contains a different ring system as its E ring compared to the phenyl E ring in the derivatives.

As you rightly point out, it is a small SAR study, and not all derivatives can be directly compared. Inspired by your comment, we extended the study by adding 4 more compounds, which makes the SAR analysis more systematic (the newly added compounds are highlighted in yellow in Table 1 below).

We also corrected the text and expanded it:

“To validate these results and to identify a potential binding mode, further anthraquinone derivatives were tested at the wt P2X4 receptor, the P2X4-E307T receptor mutant and the chimeric P2X4(P2X2) c10R301-Q308 receptor (Table 1). Structure-activity relationship (SAR) studies showed that at least one acidic function is required, as PSB-25041, a neutral molecule lacking any acidic group, is inactive. For all active compounds, a large potency increase is observed at the P2X4-E307T mutant and also at the chimeric P2X4(P2X2) receptor c10 (as far as investigated), in comparison to the wt P2X4 receptor. Besides Cibacron Blue, the most potent compounds of the present series at the mutant receptors are PSB-0739, bearing two sulfonate groups, one in position 2 (ring C), and one on ring D, and its derivative lacking the amino group in the 1-position (PSB-24039). Sulfonate or carboxylate groups at ring C and D are similarly well tolerated (Table 1). SARs are somewhat different at the wt P2X4 receptor as compared to its E307T mutant. The most potent antagonist at the wt P2X4 receptor is PSB-0826, bearing one sulfonate group (IC₅₀ 1.35 μM). The smaller compounds PSB-0711 and PSB-25012, lacking ring E, are more potent at the wt P2X4 receptor than many of the larger derivatives, but less potent at the mutant receptor. In contrast to Cibacron Blue, none of the other investigated anthraquinone derivatives showed a biphasic curve at the wt P2X4 receptor in calcium influx assays under the applied conditions.”

Table 1 | Potency of selected anthraquinone derivatives determined by measurement of calcium influx at the human wt P2X4 receptor, the P2X4-E307T receptor mutant, and the chimeric receptor P2X4(P2X2) c10^{R301-Q308}, stably expressed in 1321N1 astrocytoma cells.

	IC ₅₀ ± SEM (μM) ^a		
	wt P2X4	P2X4-E307T	P2X4(P2X2) c10 ^{R301-Q308}
Cibacron Blue 	12.1 ± 4.2	0.0178 ± 0.0047	0.00476 ± 0.00062
PSB-0704	12.5 ± 1.9	0.0724 ± 0.0102	n.d. ^b

PSB-0739 	8.91 ± 0.67	0.0353 ± 0.0047	0.00341 ± 0.00049
PSB-24039 	11.4 ± 0.3	0.0404 ± 0.0090	n.d. ^b
PSB-0826 	1.35 ± 0.22	0.164 ± 0.042	0.0922 ± 0.021
PSB-25041 	>10	>10	>10
PSB-0801 	25.3 ± 0.8	0.180 ± 0.007	0.0429 ± 0.0033
PSB-0711 	4.33 ± 1.29	0.289 ± 0.044	0.0619 ± 0.0103
PSB-25012	3.28 ± 0.80	0.204 ± 0.027	0.0448 ± 0.0073

PSB-09059 	20.0 ± 8.45	0.195 ± 0.014	0.266 ± 0.018

^aData represent means \pm SEM of at least three biological replicates, performed in technical duplicates. The inhibition of anthraquinone derivatives was determined in the presence of ATP at its respective EC₈₀. ^bn.d., not determined.

Discussion

*“Computational studies led to the hypothesis that the glutamate Glu307 forms an ionic lock with a nearby basic amino acid residue (likely Arg82 and Lys298, but Arg301 can also not be completely excluded) thereby preventing high-affinity binding of the anthraquinone derivatives by occluding part of its binding site (Table 1). This hypothesis was supported by the fact that small anthraquinone derivatives with only one acidic function (e.g. PSB-0826, PSB-25012, Table 1 and **Supplementary Fig. 9**) were several-fold more potent at the wt P2X4 receptor than the larger derivatives bearing two or three acidic groups (Cibacron Blue, PSB-0704, PSB-0739).”*

Discussion

“... Together with the SARs at the wt and the E307T mutant receptor (Table 1), it can be concluded that the anthraquinone derivatives interact with the same binding site on both mutant and wt receptor (also see Supplementary Fig. 8 and 9 for docking of anthraquinone derivatives into the human wt and mutant P2X4 receptor).”

We added Supplementary Fig. 9:

Supplementary Fig. 9 | Manual docking of the anthraquinone derivatives (a) PSB-25012 and (b) PSB-0826 into the P2X4-PSB-0704 structure. Both compounds showed high inhibitory potency at the wt human P2X4 receptor. Note that these compounds do not have a negatively charged substituent on ring D.

- Are the authors confident that none of the five derivatives did not exhibit a biphasic modulator profile at WT P2X4? This should explicitly be stated in the Results, and it could also be interesting to discuss this difference between the compounds in the Discussion.

Only Cibacron Blue, but none of the other investigated antagonists showed a biphasic curve. This is now stated in the Results section.

“In contrast to Cibacron Blue, none of the other investigated anthraquinone derivatives showed a biphasic curve at the wt P2X4 receptor in calcium influx assays under the applied conditions.”

- Lines 332-333: In the obtained P2X4-E307T structure, only the ECD of the receptor could be modeled, whereas the TMD was too disordered. While this is understandable, it does mean that the structure of this receptor domain can not be seen, and the ECD structure could be influenced by the structure of the TMD of the receptor. The authors should point this out and the potential implications of this should be discussed.

While we previously focused on the extracellular domain and the allosteric binding site for PSB-0704, we have now re-evaluated the data and refrained from using local refinements in this branch of the processing tree (see Supplementary Figure 6). We can now clearly see a significant portion of the TM helices, which align very well with the TM-helix conformation of PDB-id 4dw0 and 9bqh (both representing the closed state of the receptor). Please see Supplementary Figures 7 and 8.

- Lines 341-347: The mutations of two asparagines in the cryo-EM construct is described in the methods section, it is mentioned here, and it is also mentioned in the Discussion. Maybe omit one of the two latter mentionings from the manuscript?

We deleted it in the Discussion section.

- Lines 357-359: I assume that that the authors do not see a density in the ATP binding site in the P2X4-E307T structure?

Yes, you are right. This is now explicitly mentioned in the text.

“We therefore conclude that our structure represents the closed state of the receptor without ATP bound (Supplementary Fig. 8d), for which no electron density was seen, and which also had not been added during cryo-EM sample preparation.”

- Lines 359-361: This observed difference in compared to the zebrafish structures are mentioned but not discussed further. Do the author think it is the presence of the modulator in the binding site, the absence of the ionic lock, or something else?

We think, the effect is induced by the ligand. We included this information:

“The antagonist appears to induce a downward movement of the head domain in comparison with the zebrafish P2X4 structure and also compared to a recent human P2X4 receptor structure in the desensitized state⁵¹.”

- Lines 382-390: The authors should confirm/substantiate the proposed binding mode of PSB-0704 by mutagenesis. It would be obvious to introduce R301A and R82A mutations into P2X4-E308T and characterize PSB-0704 at these mutants. Some of the hydrophobic interactions with aromatic residues in the binding site could also be probed.

This has now all been done (see above and see revised manuscript).

- Lines 392-403: Based on their comparison of the P2X4-E308T/PSB-0704 structure and the AF3 WT P2X4 model, the authors ascribe the low modulator binding to the WT receptor to the ionic lock E308/R82-K298 in the WT receptor that “blocks a significant part of the compound binding site”. However, these modulators do bind to and inhibit the WT P2X4, so clearly the modulators can access the binding site – or parts of it? Is the reduced potency of the modulators thus due to a less optimal binding conformation of them at this binding site in the WT receptor when they access it? The authors should be more precise as to what this comparison shows.

The AF3 model suggests that critical regions of the PSB-0704 binding site adopt a different conformation in the wt P2X4 receptor in the ligand-free state. Since the anthraquinone derivatives still bind to the wt P2X4 receptor, although with lower affinity, this indicates a less optimal binding site, which is stabilized by the “ionic lock” formed by residues E308, R82, and K298. This clarification has been included into the revised manuscript.

In fact, smaller compounds appear to fit better into the binding site of the wt P2X4 receptor (see Table 1), and the SARs at the wt as compared to the mutant P2X4 receptor are not identical, indicating a different shape of the binding site.

Interestingly, the adjacent allosteric binding site to which the antagonists BAY430 and BAY-1797 are binding, is also affected by the breaking of the ionic lock in the P2X4-E307T receptor. We have now performed calcium influx assays on the P2X4-E307T receptor in comparison to the wt P2X4 receptor and determined blockade by three standard P2X4 receptor antagonists: BX430, BAY-1797, and 5--BDBD. While BX430 and BAY-1797 are more potent at the mutant receptor (5-fold, and 10-fold, respectively), the antagonist 5-BDBD shows the same antagonistic potency at both the wt and the E307T mutant receptor (see Table 1 and Supplementary Fig. 11). 5-BDBD binds to a different – still

unknown – allosteric binding site, while the other two antagonists bind in close vicinity to the anthraquinone binding site. Our results show that the conformational change in the E307T mutant is also favorable for these two antagonists, which now interact with their binding sites in a more optimal way. This is now explained as follows:

“Finally, we studied the inhibitory potency of the standard P2X4 receptor antagonists BX430, BAY-1797 and 5-BDBD in ATP-induced calcium influx assays at the human P2X4-E3207T receptor mutant expressed in 1321N1 astrocytoma cells. This was motivated by the fact that the newly discovered allosteric binding site for anthraquinones is adjacent to the previously discovered binding site for BX430 and BAY-1797. In fact, both compounds were several-fold more potent at the mutant as compared to the wt P2X4 receptor, while 5-BDBD was equally potent at both the mutant and the wt receptor (Table 1 and Supplementary Fig. 11).”

Discussion

“Compared to the P2X4 receptor antagonists BX430 and BAY-1797, the antagonist PSB-0704 is located further away from the membrane surface (Supplementary Fig. 8e). Interestingly, both compounds show increased affinity for the P2X4-E307T mutant as compared to the wt receptor (Table 1) indicating that the disruption of the ionic lock is also beneficial for their binding interactions. This is not the case for another allosteric P2X4 receptor antagonist, 5-BDBD (Table 1), which can be expected to interact with a different binding site.”

“Discussion”

- Discussion overall: The Discussion could benefit from a very extensive revision. A LOT of the subsections in the Discussion are characterized by containing a lot of summary of either things from the Results section or things better presented in the Introduction section. For example, much of lines 430-454, lines 456-474. The authors should review the section carefully for this, and make the Discussion section more concise and focused on discussion of the results.

We agree, and we have now extensively revised the Discussion as suggested.

- Fig. 10: 1) It would be helpful for the reader if the position of the different binding sites in the structure and the detailed images of each binding site were linked with arrows or boxes. The different coloring of the ligands is not sufficient to identify their positions in the structure. 2) If the authors want to make the point that the binding site for these modulators is indeed novel, a part of this figure should be a closer look at this region and the binding sites for PSB-0704 and the other modulators targeting this region (Suppl. 6d).

Thank you for your suggestion. We have now improved the figure and added arrows. In addition, Supplementary Figure 8g shows the binding site of PSB-0704 in comparison to BX430 and BAY-1797.

- Lines 526-538: The discussion of the biphasic profile of Cibacron Blue as a modulator does not lead to much insight. This reviewer is somewhat skeptic about the first proposed scenario (1-2 modulator molecules bound leads to potentiation, three leads to inhibition) explaining this concentration-dependent effect. It would require that low concentrations of Cibacron Blue would bind but not saturate all three sites of the receptor, but only at higher concentrations would be able to bind all three sites – i.e. that the binding site for the second/third of the three sites in the trimer was significantly lower than the first one or two sites – a sort of reverse-allostery phenomenon? Or how does the author envision this scenario?

We have now modified the discussion:

“Our results do not explain the biphasic behavior of Cibacron Blue at the human P2X4 receptor. It has been observed previously that some allosteric modulators can show receptor activation or inhibition depending on concentration⁵⁶. Based on the present structural data of the anthraquinone binding site it is tempting to speculate that the degree of downward movement of the head domain induced by a single molecule of the relatively large Cibacron Blue could account for the potentiating effect observed with low concentrations of the ligand. Thus, only Cibacron Blue can induce a "pre-open" state of the receptor and consequently higher agonist efficacy, even if only two ATP molecules can bind. If, however, two or more antagonist molecules are bound, ATP cannot bind or induce the required conformational change and the receptor is blocked.

*Another explanation could be the presence of different binding sites to which the modulator has different affinity, as postulated for some pentameric ligand-gated ion channels and their allosteric ligands⁶⁹. This hypothesis is supported by the results obtained in radioligand binding studies using the stable ATP analog [³⁵S]ATP γ S, in which Cibacron Blue showed a biphasic competition binding curve³⁰. It has to be pointed out that this behavior is only observed for the large molecule Cibacron Blue consisting of 6 rings, but not for any of the truncated analogs, including PSB-0704, all of which only exhibit antagonistic activity, in calcium influx assays at recombinant 1321N1 astrocytoma cells as well as in TEVC experiments at *Xenopus* oocytes.”*

- Lines 540-546: I am not quite sure what the authors are suggesting here. It is true that the P2X4-E307T/PSB-0704 pair could be used in explorations of the P2X4’s role in physiology, for example by construction of a P2X4-E307T knock-in mice and the use the compound (provided that the bioavailability and PK properties of PSB-0704 is suited for in vivo use). However, it would have to be a modified system, such as a knock-in animal, to enable such studies. The authors should try to be more specific about what they mean.

We have now better explained the strategy:

“For example, knock-in animals expressing the P2X4-E307T receptor in organs or cell types of interest could be prepared. These would allow the specific blockade of the mutated receptor by low concentrations of antagonist without effects on the wt receptor expressed at other sites. Such an approach of chemical hijacking using an engineered protein-ligand pair has found broad application in basic research, in particular in chemical biology^{72, 73”}

“Conclusions”

- I am not as convinced as the authors that this study will be useful for drug design. Even if it is argued that this study have demonstrated the importance of an ionic lock for the low potencies of modulators targeting this site in the WT receptor, the strategy proposed by the authors to use this insight for drug design is highly speculative: the design of compounds that “disrupts the intermolecular ionic lock by interacting with Glu307” and thus can allow the “regular” modulators to do their job. Even if one were able to develop compounds that potentially can bind and disrupt this ionic lock, it would mean that these

compounds would bind very close to the modulator binding site, and they would have to do so within changing the ability of this adjacent site to bind the modulator. That seems like quite a tall ask...

We agree, but we know that it is – at least in principle – possible. We observed in several structures of GPCRs (adenosine receptors) that we recently determined that agonists as well as antagonists can disrupt ionic locks upon binding to the receptors.

An example has now been added to support our claim:

“Since the hypothesis of ionic lock formation has been confirmed, this offers the possibility to design compounds that disrupt the intermolecular ionic lock by interacting with Glu307, to open up a larger, high-affinity binding site for antagonists. Introduction of a basic moiety into the structure of PSB-0704, e.g. attached to the anthraquinone 1-position instead of or linked via the anilinic amino group, could be a plausible strategy. In fact, disruption of ionic locks is a common mechanism in drug-receptor interactions. For example, in G-protein-coupled receptors ionic locks stabilize the inactive state of the receptor^{72,73}. They typically act as molecular switches, which can be disrupted by drugs⁷⁴.”

The design, synthesis and characterization of such a compound will be an interesting future project, but is beyond the scope of the present study.

While I appreciate that the authors wish to convey potential possibilities in their findings, I think this proposal is too theoretical and speculative, and I would suggest the authors that they refrain from going down that road. The claim that these results “provide a basis that will enable and promote future drug research” is quite a stretch.

We respect your opinion, but – being medicinal chemists – we value the insights structural biology provides as a basis for drug development, which should not be underestimated. Nevertheless, we reworded the sentence as follows:

“Our results provide a basis that could support future drug development efforts.”

Reviewer #2 (Remarks to the Author)

In this work the authors have investigated the structure-activity relationship (SAR) of Cibacron Blue (CB) and anthraquinone derivatives at human P2X₄, showing that CB exhibits a biphasic concentration-response, and showing that the P2X₄-P2X₂ receptor chimera (with region R301-Q308 swapped), and subsequently the mutant E307T, display significantly improved CB potency. The authors use AlphaFold modelling to propose that the residue E307 may form an 'ionic lock' with basic amino acid residues which may prevent the binding of anthraquinone derivatives to P2X₄ with high affinity to explain their findings. They go on to describe the cryoEM structure of the extracellular domain of the human P2X₄ receptor mutant E307T with the anthraquinone derivative PSB-0704 bound, delineating a novel allosteric pocket, which partially overlaps with the BAY-1797 pocket described for zebrafish P2X₄. The location of the pocket is consistent with functional data, suggesting that PSB-0704 forms ionic interactions with R301 and R82, and that in wild-type P2X₄, E307 prevents access of PSB-0704 to the binding pocket.

In my opinion this is a very interesting study, and the work describing CB mode of action, SAR and the potential molecular interpretation is carefully performed. The 3D-structure of the hP2X₄ E307T extracellular domain bound to PSB-0704 is relatively low-resolution, and although it appears plausible, I have reservations about the cryoEM analysis and interpretation, and how useful it might be for drug discovery. My specific major and minor comments are listed below.

Thank you for your expert opinion. We performed a number of additional experiments and hope to be able to fully convince you.

Major comments

1. It is not clear from reading the abstract or introduction that the structure reported is the extracellular domain only (even though the full-length protein was used for structure determination), or that the structure represents a mutation (E307T) introduced to increase antagonist potency (see my minor comments 1 and 2 below). Both these important points should be emphasised.

Thank you for pointing this out. We have now explained from the beginning, including the Abstract, that we determined the structure of the P2X₄-E307T mutant. We avoid talking about a full-length structure, as not all parts of the molecule were resolved. However, note that by reprocessing our dataset, we were now able to reconstruct a significant part of the TM domains, which aligns well with available structures (see below).

2. Results, page 16-17: The reported cryoEM structure lacks information in the transmembrane region, which to me is surprising, and is also in contrast to several recent cryoEM structures of P2X receptors [Additionally, a human P2X₄ cryoEM structure (reported on bioRxiv in July 2024; doi: 10.1101/2024.07.25.605151) is in preparation at a reported resolution of 2.3 Å which appears to show well-defined structure in both the transmembrane and cytoplasmic domains]. The only comment from the authors on this surprising result in this manuscript is that 'The transmembrane helices were disordered and could not be modelled.' What does this say about the quality of the particle dataset (or the subsequent analysis), and are the authors not concerned about this? I appreciate that preferred orientations were an issue, but what steps were taken in sample preparation to address this? I also appreciate that local refinement was used to generate the maps enabling fitting of the extracellular domain, and that this is the location of the binding site for the antagonist and thus the focus of data interpretation, but how confident can the authors be about this structure where a key part of it (that surely one would expect to be relatively ordered in the same way as the extracellular domain if the protein is folded properly) is absent in the reconstruction?

Your comments triggered us to re-process our cryo-EM dataset and try to get a better overall reconstruction of the receptor, instead of only focusing on the ligand binding site, which had been our initial strategy. Thank you for your significant contribution in improving our manuscript!

As shown in Supplementary Fig. 6, we added an additional branch to the processing tree, where we refrained from using local refinements. Indeed, we can now see a significant part of the TM helices of the receptor, and residues 42 to 344 are now included in our model. Reassuringly, our model aligns well with the TM domains of 4dw0 and 9bqh (apo closed states of the zebrafish and human P2X4 receptors, Supplementary Fig. 7 (also pasted below for your convenience)). Still, our reconstruction does not show every residue of the receptor. However, we hope that the referee appreciates that we have pointed out the limitations of our model and that it is of sufficient quality for the conclusions that we have drawn from it: 1) Localization of the ligand binding site. 2) The conformation of the receptor is very similar to the apo closed state of the P2X4 receptor (from 4dw0 and 9bqh).

3. Results, page 17: The resolution of the reported structure is relatively low (compared to recent cryoEM structures of other P2X receptors), at 3.35 Å. However, the authors state ‘The antagonist PSB-0704, which had been added during protein purification and again immediately before the cryo-EM grid preparation, could be placed into this map feature with confidence’. How was the PSB-0704 ‘placed’ into the map – was this fitted unambiguously by software, or manually by the authors with reference to their mutagenesis data? Is there more than one possible conformation for this molecule that would fit the density, and how were alternate conformations excluded? Was there any potential bias in the fitting process? There is no description of how the PSB-0704 was fitted to the structure in the Methods.

PSB-0704 is quite rigid and has a very bulky structure. As shown in the manuscript, the compound fitted well into the size of the extra density, which was observed at a position consistent with the pharmacological data (close to E307T). We placed the compound in the extra density and tested different orientations with respect to the protein. The binding mode shown in the manuscript was clearly the best explanation for the map feature and the only one that led to reasonable interactions with the protein. We parametrized the ligand with phenix.elbow and used Isolde to refine the ligand position. Therefore, we are indeed confident that the ligand was correctly placed by our "semi-automated" approach. We agree that we could have explained this more clearly and have now added this explanation to the manuscript.

Here is the text from the new Methods section:

“We placed the compound in the extra density and tested different orientations with respect to the protein. The binding mode shown in the manuscript was clearly the best explanation for the map feature and the only one that led to reasonable interactions with the protein. We parametrized the ligand with PHENIX.elbow⁸⁶ and used ISOLDE⁶⁴ to refine its ligand position. B-factors of the model were refined with PHENIX⁸⁶ and the model was validated with PHENIX and MOLPROBITY⁸⁷.”

Main manuscript text:

“While building the structural model, we identified a large feature at the trimer subunit interface, at the back of the head domain. This feature was clearly visible in both the overall and focused reconstructions. The antagonist PSB-0704, which had been added during protein purification and again immediately before the cryo-EM grid preparation, was placed into this map feature by hand and refined with ISOLDE⁶⁴ (Fig. 9).”

4. Supplementary Figure 4: Several of the 2D class averages appear to contain more than one particle – there are clear separate blobs of density. Presumably the reprojections from the 3D structure will not match these classes as the reconstruction was done in a different way, but it would be good to see how well they correlate. Were the 2D classes in Panel a all used to generate the map with preferred orientations that was difficult to interpret?

As shown in the processing diagram (Supplementary Fig. 6), the 427,934 particles after 2D classification led to two ab initio models. One (260,278 particles) clearly resembled the structure of P2X4, while the other (167,656 particles) did not, and the underlying particles were discarded. Thus, no, not all of the particles behind the 2D classes shown in Supplementary Fig. 6a (previous Supplementary Fig. 4) are part of the original reconstruction. As mentioned by the reviewer, the final reconstruction was prepared using a workflow based on 3D classification only, which gave much better results. Nevertheless, we decided to show the 2D classes in the processing tree because they clearly show the expected structure and it is common to do so.

5. Results, page 9: Is there a reason why the authors did not also make and test the R301K mutation?

We have now made a series of additional mutations including R301K. Two tested anthraquinone derivatives (PSB-0704 and PSB-0739) were similarly potent at the R301 receptor and its R301K mutant (no statistically significant difference) – see Fig. 10.

6. Discussion, page 23: ‘our structure was obtained of the full-length human P2X4 receptor construct,...’ I think it is very important to state that you used the full-length protein, but your structure is only of the extracellular domain.

Thank you for pointing this out, you are of course right. We now refrain from calling our structure a full-length structure and clearly state this in the manuscript:

“The final structure comprises residues 42-344 of the human P2X4 receptor (Fig. 8a). The overall architecture of our model reflects the typical trimeric arrangement of P2X receptors with a large extracellular domain and two transmembrane (TM) helices in each subunit. The interaction between the monomers is characterized by very large interfaces with a combined buried surface area of ~3,500 Å² between each pair of monomers and distinct charge complementarity (Fig. 8b). Each monomer resembles the shape of a dolphin, consistent with previously reported P2X receptor structures³⁶ (Fig. 8).”

7. Discussion, page 24: ‘Sufficiently high overall resolution’. This is not the overall resolution of the full-length protein, just of the extracellular domain, and it is also a low resolution compared to recent cryoEM structures of P2X receptors. I think it is important to discuss the potential shortcomings of the structure obtained, and of fitting ligands at this relatively low resolution (relates to my point 3 above). There is a general lack of detail on these points in the Methods, Results and Discussion.

We agree that our resolution is lower than in Shi et al. (<https://doi.org/10.1101/2024.07.25.605151>). However, a PDB query shows that this is a typical resolution achieved with single particle reconstructions since 1/1/2020.

Table. Depositions of EM reconstructions in the PDB since 1.1.2020.

Resolution of reconstruction (Å)	# depositions
<1	0
1-2	158
2-3	5.392
3-4 (resolution of our reconstruction)	10.626
4-5	1.859
5-6	287
6-7	288
7-8	194
8-9	116
>9	229

Furthermore, we strongly believe that our structure, in combination with the wealth of pharmacological data, provides important new insights and complements further structures including high-resolution structures, for example those of Shi et al.

We have now included a more detailed description of the ligand fitting procedure in our manuscript.

Minor

comments

1. Abstract: 'structure of full-length human P2X4'. I don't think it is reasonable to state that the determined structure is full-length, because of the issues with reconstruction and the disorder of the transmembrane domains. It might be better to state 'structure of the extracellular domain of the human P2X4 mutant E307T in complex with the anthraquinone derivative PSB-0704 was determined from cryoEM analysis of the full-length protein at a resolution of 3.35 Å.'

Yes, we agree and we have changed the wording, as discussed above.

2. Introduction: I think it would be better to state 'structure of the extracellular domain of the human P2X4 mutant E307T, derived from cryoEM analysis of the full-length protein'.

See above, we have now changed the wording.

3. Results, page 7: Why not also make an AF3 model of your chimeric receptor and analyse the intersubunit contacts in the same region?

Thank you for this excellent idea. We now included an AF3 model of the chimeric receptor and analyzed the intersubunit contacts in the same region (see Fig. 2). The AF3 model of the chimera aligns well with our interpretation.

4. Supplementary Table 1: It would be good to report the resolution of each structure in this table, and also provide details on the truncations and mutations introduced in each construct.

Very good suggestion. We have now included the data as suggested into Supplementary Table 1.

Reviewer #3 (Remarks to the Author):

Summary:

The manuscript explores the structural basis of P2X4 receptor antagonism, focusing on anthraquinone derivatives like PSB-0704. Using cryo-EM, mutagenesis, and SAR studies, the authors identify a novel allosteric binding site on the human P2X4 receptor, distinct from previously known sites. High-resolution cryo-EM data reveal ligand interactions and receptor conformations, while complementary approaches validate the findings. Insights into PSB-0704's molecular interactions offer a foundation for developing potent, selective P2X4 antagonists. The compounds used in the SAR study were synthesised and characterised in prior work by the research group.

Major Weaknesses:

-While the study mentions the biphasic nature of Cibacron Blue's activity, the underlying mechanisms remain speculative. Further experiments or additional discussion of this phenomenon would strengthen the manuscript.

We performed additional experiments using a lower ATP concentration (EC₂₀) as suggested by Reviewer 1. The results were similar. We have now extended the discussion:

*Our results do not explain the biphasic behavior of Cibacron Blue at the human P2X4 receptor. It has been observed previously that some allosteric modulators can show receptor activation or inhibition depending on concentration⁵⁶. Based on the present structural data it is tempting to speculate that the degree of downward movement of the head domain induced by a single molecule of the relatively large Cibacron Blue could account for the potentiating effect observed with low concentrations of the ligand. Thus, only Cibacron Blue can induce a "pre-open" state of the receptor and consequently higher agonist efficacy, even if only two ATP molecules can bind. If, however, two or more antagonist molecules are bound, ATP cannot bind or induce the required conformational change and the receptor is blocked. Another explanation could be the presence of different binding sites to which the modulator has different affinity, as postulated for some pentameric ligand-gated ion channels and their allosteric ligands⁶⁹. This hypothesis is supported by the results obtained in radioligand binding studies using the stable ATP analog [³⁵S]ATPγS, in which Cibacron Blue showed a biphasic competition binding curve³⁰. It has to be pointed out that this behavior is only observed for the large molecule Cibacron Blue consisting of 6 rings, but not for any of the truncated analogs, including PSB-0704, all of which only exhibit antagonistic activity, in calcium influx assays at recombinant 1321N1 astrocytoma cells as well as in TEVC experiments at *Xenopus* oocytes.*

-The study focuses primarily on engineered receptor constructs (e.g., E307T mutant). The authors should discuss how these findings translate to the wild-type receptor in physiological contexts.

This is an important and valid point.

We have now extended the SAR studies based on our findings and discovered that smaller anthraquinone derivatives are more potent at the wt P2X4 receptor than larger ones, which is in agreement with our data. We now discuss in more detail how our results can be further translated to the wt receptor.

"Computational studies led to the hypothesis that the glutamate Glu307 forms an ionic lock with a nearby basic amino acid residue (likely Arg82 and Lys298, but Arg301 can also not be completely excluded) thereby preventing high-affinity binding of the anthraquinone derivatives by occluding part of its binding site (Table 1). This hypothesis was supported by the fact that small anthraquinone derivatives with only one acidic function (e.g. PSB-0826, PSB-25012, Table 1 and Supplementary Fig. 9)

were several-fold more potent at the wt P2X4 receptor than the larger derivatives bearing two or three acidic groups (Cibacron Blue, PSB-0704, PSB-0739)."

...

"All of these and further mutations are in agreement with the determined structure supporting our ionic-lock hypothesis on the wt P2X4 receptor. Together with the SARs at the wt and the E307T mutant receptor (Table 1), it can be concluded that the anthraquinone derivatives interact with the same binding site on both mutant and wt receptor (also see Supplementary Fig. 8 and 9 for docking of anthraquinone derivatives into the human wt and mutant P2X4 receptor). The determined structure, although not obtained for the wt P2X4 receptor, is therefore useful for drug design. It rationalizes the SAR data showing, for example, that the amino group at ring C is not important for the interaction with the P2X4 receptor (Table 1). In our structure, the amino group is ~4 Å away from the hydroxyl group of Tyr300, which is too far to form a direct hydrogen bond (Fig. 9). Aromatic amines (anilines) are regarded as potential toxicophores⁶⁷, and therefore these insights can be used to improve the compounds not only with regard to potency, but potentially also with regard to toxicity. The oxygen atoms of the quinone ring B have not shown any polar interactions with the receptor and thus are likely also not required. This opens up the possibility to design antagonists without the redox-reactive quinone structure by replacing it with a more stable moiety."

"Since the hypothesis of ionic lock formation has been confirmed, this offers the possibility to design compounds that disrupt the intermolecular ionic lock by interacting with Glu307, to open up a larger, high-affinity binding site for antagonists. Introduction of a basic moiety into the structure of PSB-0704, e.g. attached to the anthraquinone 1-position instead of or linked via the anilinic amino group, could be a plausible strategy. In fact, disruption of ionic locks is a common mechanism in drug-receptor interactions. For example, in G-protein-coupled receptors ionic locks stabilize the inactive state of the receptor^{72,73}. They typically act as molecular switches, which can be disrupted by drugs⁷⁴."

-While the structural and SAR data are robust, additional functional assays, such as electrophysiological recordings for PSB-0704 at the wild-type receptor, would enhance the manuscript.

As suggested, we have now performed additional two-electrode voltage-clamp experiments (TEVC) using *Xenopus* oocytes expressing the wt or mutant P2X4 receptor (see Fig. 10 and Supplementary Fig. 10). These experiments were also used to study additional mutants. We found that the potency of the antagonists, in particular of PSB0704, depends on the duration of preincubation indicating a slow on-rate. Overall, the electrophysiological recordings are a valuable addition that further support our results and conclusions. Thank you for your suggestion.

Fig. 10 | Two-electrode voltage clamp (TEVC) experiments and structure-based site-directed mutagenesis studies. **a** Concentration-response curves for ATP at the wildtype (wt) human (h) P2X4 receptor and the hP2X4-E307T receptor mutant. Receptors were expressed in *Xenopus laevis* oocytes and current responses to 5s-pulses of ATP were analyzed by TEVC at -60 mV. ATP test concentrations and a reference concentration of 100 μ M ATP were alternatingly applied in 2 min intervals. Responses were calculated in relation to the reference concentration, fitted to the four-parameter Hill equation, and normalized to the maximal response. Data are presented as mean \pm SEM from 3–5 oocytes. **b** Representative current traces recorded in oocytes expressing the P2X4-E307T receptor mutant. Colored lines indicate the presence of the indicated antagonist during a 2 min preincubation and during the application of 10 μ M ATP (black line). 3 μ M PSB-0704 and 300 nM PSB-0739 were used. **c** Concentration-inhibition curves and chemical structures of the antagonists PSB-0704 and PSB-0739. Responses to 3-s pulses of 10 μ M ATP were recorded after 20 s superfusion with the indicated antagonist concentrations and normalized to the preceding ATP response in the absence of antagonist. Data are represented as mean \pm SEM from 3–5 oocytes. **d, e** TEVC analysis of inhibitory effects of (d) PSB-0704 (3 μ M) and (e) PSB-0739 (300 nM and 1 μ M) on the indicated single and double P2X4 receptor mutants expressed in *Xenopus laevis* oocytes. Inhibition of current responses evoked by 3s-pulses of 10 μ M ATP

after 2 min superfusion with antagonist is shown. Current responses evoked by co-application of ATP and antagonist were normalized to the preceding ATP-responses in the absence of antagonist and effects were represented as percent inhibition. Data are presented as mean \pm SEM from at least 3 oocytes. The non-parametric Kruskal–Wallis test was applied, followed by Dunn’s multiple comparisons test. The significance levels were defined as follows: * $P < 0.05$, ** $P < 0.01$, *** $P < 0.001$, **** $P < 0.0001$; ns, not significant. For PSB-0739, the significance was evaluated for the lower concentration of 300 nM.

-A more in-depth comparison of PSB-0704 with other P2X4 antagonists (e.g., BX430, BAY-1797) in terms of binding affinity, potency, and pharmacokinetics would be beneficial.

We have now prepared a table to compare these compounds. Unfortunately, available data are limited. We also added the anthraquinone derivative PSB-0826, which we have now included into the revised manuscript since its potency for the wt hP2X4 receptor is in the same range (only slightly weaker) as that of BX430 and BAY-1797. The largest set of data is available for BAY-1797, which was developed in industry. However, in contrast to the other antagonists, it does not fully inhibit the receptor, only by about 80%. We decided not to include the table into the manuscript since the data are limited and the compounds therefore not well comparable.

Table. Comparison of potencies and pharmacokinetic properties of standard P2X4 receptor antagonists and selected anthraquinone derivatives (as far as available)

	Potency (IC₅₀ value) at the human P2X4 receptor	Properties
PSB-0704	12.5 μ M (Ca ²⁺ , 1321N1 astrocytoma)	Water-soluble long preincubation enhances potency
PSB-0826	1.35 μ M (Ca ²⁺ , 1321N1 astrocytoma)	Water-soluble
BX430	0.54 μ M (patch clamp, HEK293) https://doi.org/10.1124/mol.114.096222 0.127 μ M (Ca ²⁺ , 1321N1 astrocytoma) https://doi.org/10.1016/j.lfs.2022.121143 4.5 μ M (SPR) ^b 10.1016/j.heliyon.2023.e21265	Lipophilic significantly weaker at mouse and rat than at human P2X4 receptors
BAY-1797	0.211 μ M (Ca ²⁺ , HEK293) doi: 10.1021/acs.jmedchem.9b01304. 0.320 μ M (Ephys Qpatch, 1321N1 astrocytoma) doi: 10.1021/acs.jmedchem.9b01304.	No complete inhibition (only partial inhibition) Stable in human and rat blood plasma; orally bioavailable;

	Similar potency at mouse and rat P2X4 receptors	in vivo $t_{1/2}$: 2.6 h (i.v. or p.o.)
--	---	--

^an.d., not determined

^bsurface plasmon resonance (SPR) competition assay

Minor Suggestions:

-Provide additional details on cryo-EM data acquisition and processing, especially regarding the resolution and interpretation of the binding site.

This comment aligns with questions asked by referee #2. As detailed above, we have now reprocessed our cryo-EM dataset to reveal a significant part of the TM domains and have added additional details about the ligand fitting procedure.

-Specify the rationale for selecting specific anthraquinone derivatives for SAR analysis.

We selected structurally diverse anthraquinone derivatives to study SARs. To expand the structural variations, we now included 4 additional derivatives (Table 1), which are discussed in the manuscript.

Importantly, we identified small anthraquinone derivatives that are more potent at the wt P2X4 receptor than the larger compounds (e.g. Cibacron Blue), which is consistent with the smaller binding site observed in the wt P2X4 receptor as compared to the E307T mutant, in which the ionic lock is disrupted and the binding site is enlarged.

-Conclude with a more explicit discussion of how the findings can inform the design of hybrid compounds or other allosteric modulators.

A more detailed discussion has now been included:

“All of these and further mutations are in agreement with the determined structure supporting our ionic-lock hypothesis on the wt P2X4 receptor. Together with the SARs at the wt and the E307T mutant receptor (Table 1), it can be concluded that the anthraquinone derivatives interact with the same binding site on both mutant and wt receptor (also see Supplementary Fig. 8 and 9 for docking of anthraquinone derivatives into the human wt and mutant P2X4 receptor). The determined structure, although not obtained for the wt P2X4 receptor, is therefore useful for drug design. It rationalizes the SAR data showing, for example, that the amino group at ring C is not important for the interaction with the P2X4 receptor (Table 1). In our structure, the amino group is ~4 Å away from the hydroxyl group of Tyr300, which is too far to form a direct hydrogen bond (Fig. 9). Aromatic amines (anilines) are regarded as potential toxicophores⁶⁷, and therefore these insights can be used to improve the compounds not only with regard to potency, but potentially also with regard to toxicity. The oxygen atoms of the quinone ring B have not shown any polar interactions with the receptor and thus are likely also not required. This opens up the possibility to design antagonists without the redox-reactive quinone structure by replacing it with a more stable moiety.”

“Since the hypothesis of ionic lock formation has been confirmed, this offers the possibility to design compounds that disrupt the intermolecular ionic lock by interacting with Glu307, to open up a larger, high-affinity binding site for antagonists. Introduction of a basic moiety into the structure of PSB-0704,

e.g. attached to the anthraquinone 1-position instead of or linked via the anilinic amino group, could be a plausible strategy. In fact, disruption of ionic locks is a common mechanism in drug-receptor interactions. For example, in G-protein-coupled receptors ionic locks stabilize the inactive state of the receptor^{72,73}. They typically act as molecular switches, which can be disrupted by drugs⁷⁴”

“The fact that anthraquinones and BX430/BAY-1897 have adjacent binding sites opens interesting opportunities to combine features of the different compound classes to design more potent antagonists. The design of bitopic or dualsteric ligands, substituting or replacing the benzoquinone moiety of the anthraquinones by a linker that connects the antagonist with the terminal phenylacetic acid amide moiety of BAY-1797 would be a promising strategy (Fig. 11 and Supplementary Fig. 8e).”

Recommendation:

I recommend major revisions before acceptance. The study has substantial merit and presents novel findings that significantly advance the field of P2X receptor pharmacology. Addressing the concerns highlighted above would greatly improve the manuscript's impact and clarity.

We hope that all questions have been answered to your satisfaction and that the manuscript is now suitable for publication.

With best regards,

Christa Müller

Response to Editor and Reviewers

Dear Reviewers,

thank you very much for your positive comments on our revised manuscript. In the further revised version R2, we have addressed the comments by Reviewer 1 as detailed below.

Reviewer's Comments:

Reviewer #1 (Remarks to the Author):

In their revised manuscript, Nagel et al. have more or less addressed all of my comments to and concerns about the original manuscript. I thank the authors for their responsiveness to my comments, and I congratulate them with a nice piece of work. Thus, the following comments should mainly be seen as suggestions to the authors to consider in order to further improve their manuscript.

We have to thank the reviewer for improving our manuscript with their excellent suggestions, comments and criticism, for which we are most grateful.

1. Most importantly from my perspective, the revised manuscript contain data for several additional mutants (of the residues R82, K298, R301 and N110) investigating the binding pocket for the anthraquinones in the P2X4 receptor. With the new data, the proposal of an ionic lock and its important for modulator binding to the site is much more substantiated and convincing. A minor thing for the authors to consider in this respect is how to interpret the effects of the mutations of R301 and N110. While the R301A mutation is detrimental to antagonist potency, N110A is not. Thus, the authors may want to make it more clear in the manuscript that this could suggest that there is not an interaction between R301 and N110 important for antagonist binding, but maybe rather that the importance of R301 is rooted another mechanism, such as putatively interacting with E307? The authors could consider making this more clear in the manuscript, with the new data in hand?

Answer

We agree that R301 is very important, while the N110A mutation had no obvious effect and does not play a role in antagonist binding. We have now made it clearer that the lacking effect of the mutation suggests that N110 does not play an important role in ligand binding:

Discussion:

The mutation Q308T (Fig. 4) and N110A had no effects on the potency of the anthraquinones, and indeed, the structure showed no interactions. These results suggest that the interaction between N110 and R301, predicted by AF3 and also observed in a recent cryo-EM structure of the closed-apo state of P2X4 receptor, does not play a role in binding of the anthraquinone-derived antagonists.

2. In the revised manuscript it is noted that a very recent cryo-EM structure of the human P2X4 receptor (with modulator BAY-1797 bound) has been published. This has prompted the authors to study the effects of some of their mutants on this and two other modulators as well. However, I would suggest that the authors also use this new structure to compare with their own, and assess whether the overall structural aspects of two structures are similar and support each other.

Answer:

This is a valid suggestion. We had already included a brief comparison with the closed-state apo structure of that study (9bqh), and we have now also added the BAY-1797 complex (9bqi) to this comparison (see Supplementary Figure 8, also Fig. 2) and text. We also discuss the state of the ionic lock in the BAY-1797 structure compared to ours.

“We used the Foldseek Search Server⁶⁵ and manual PDB searches to compare our structural model to known structures of P2X receptors. Supplementary Fig. 8a-c show superpositions of our model with the closed and open state zebrafish P2X4 receptor ortholog (PDB ID: 4dw0, 4dw1), as well as the recently published human P2X4 receptor in the apo closed state and in the BAY-1797-bound form, which is overall very similar (PDB IDs: 9bqh, 9bqi)⁵¹. Our model most closely resembles the closed-state apo forms of the receptor; especially the central parts of the structures at the transition between the extracellular and the membrane domain (i.e. the lower body of the dolphin) align well (Supplementary Fig. 8a, c). Pairwise structural alignments (based on one chain of each model) resulted in an r.m.s.d. of 1.5 Å between 242 Cα atoms for the zebrafish apo closed state structure, an r.m.s.d. of 1.3 Å between 242 Cα atoms for the human closed-state apo structure (9bqh), and an r.m.s.d. of 0.9 Å between 228 Cα atoms for the BAY-1797-bound structure of the human P2X4 receptor (9bqi). The r.m.s.d. between our structure and the ligand-bound open state of the zebrafish receptor was higher with 2.1 Å between 262 Cα atoms (Supplementary Fig. 8b).”

“Furthermore, we noticed a concerted downward movement (i.e. towards the membrane plane) of the head domain in our structure, compared to the zebrafish structures, but also compared to the human P2X4 structures (Supplementary Fig. 8a-c).”

“Presumably due to the presence of the ligand, the β -hairpin loop containing R82 is partly disordered and bent backwards compared to the AF3 model of the human P2X4 receptor without a bound ligand, and it is most likely also the reason for the above-described downward movement of the head domains (Fig. 8a).”

Discussion:

“The PSB-0704-bound receptor is in an inactive conformation as clearly seen by comparison with active and inactive zebrafish X-ray structures and also compared to recent human P2X4 receptor structures⁵¹ (see Supplementary Fig. 8a-c). The structural comparisons revealed that the antagonist appears to induce a “downward” movement of the head domain towards the membrane plane.”

“In the wt receptor, there is a glutamate (Glu307) instead of threonine, which is predicted to form an ionic bridge with Arg82 and additionally with Lys298 according to the computational model (see Fig. 2). Exchange of Glu307 for glutamine had virtually the same effect as its exchange for threonine (Fig. 10), both amino acid residues breaking the ionic lock. In the recent structure of the human P2X4 receptor in complex with the antagonist BAY-1797⁵¹, that binds to a different allosteric site than PSB-0704, the ionic lock is closed further confirming our hypothesis.”

The mutation Q308T (Fig. 4) and N110A (Fig. 10) had no effects on the potency of the anthraquinones, and indeed, the structure showed no interactions. These results suggest that the interaction between N110 and R301, predicted by AF3 and also observed in a recent cryo-EM structure of the closed apo state of the P2X4 receptor, does not play a role in binding of the anthraquinone-derived antagonists.

In revised Supplementary Figure 8, we added the recently published apo closed structure of the human P2X4 receptor in complex with BAY-1797 (9bqi) in addition to its apo closed state structure (9bqh).

Supplementary Fig. 8 | Comparison of the human P2X4 receptor structure with similar structures. a Superposition of the human P2X4 receptor structure (cartoon, color-coded as in the main text) with the apo closed-state structure (white cartoon) of the zebrafish ortholog; **b** as in (a) but with the open-state structure of the zebrafish ortholog; **c** as in (a) but with the desensitized state structure of the human P2X4 receptor. **d** Detail of (b) focused on the ATP-binding site. The ATP in 4dw1 (white) is shown as sticks, and the ligand PSB-0704 is shown as spheres. **e** Model of PSB-0704 binding to the wt human P2X4 receptor, based on the human P2X4-E307T receptor structure from this work. The threonine at position 307 was replaced by a glutamate residue in PyMOL. The rotamer of the side chain was chosen in such a way that no severe clashes occurred. **f** Cryo-EM structure of the human P2X4-E307T receptor (this work) superimposed with the AF3 model shown in Fig. 2 (white). **g** Cartoon model of the human P2X4 receptor. The antagonist PSB-0704 is shown as green ball-and-stick model. The relative positions of the antagonists BAY-1797 and BX430 are indicated. Note that alignments from the foldseek server were used to create the superposition. Fig. 2 was updated.

Fig. 1 / AlphaFold 3 (AF3) model of the trimeric apo human P2X4 receptor (left) and of the P2X4(P2X2) c10^{R301-Q308} variant (right). The three chains of the trimer are color-coded in blue, green and gold. In the wild type receptor (left) Glu307 is predicted to form ionic interactions with Arg82 and Lys298. Arg301 may interact with the N-glycosylated Asn110. The numbers are distances in Å.

- One of the few comments where I am left a little underwhelmed by the authors' response is to the puzzling weak antagonist potency of the anthraquinones at P2X2. While the statement

added by the authors to the revised manuscript certainly could be correct, i.e. that the binding site for the anthraquinones in P2X2 only facilitates low-affinity binding of them in contrast to the high-affinity binding to the P2X4-E307Q mutant. Nevertheless, in light of the compounds displaying comparable IC50 values at WT P2X2 and WT P2X4, which for the latter is converted to potent antagonism by the removal of the ion lock by the E307Q mutation, it is nevertheless strange that the binding sites in the two receptors lacking the ionic lock (WT P2X2 and P2X4-E307Q) are so different when it comes to interactions with the anthraquinones. I think the authors should consider going into more depth with this discussion, and in light of their structure maybe identify some of these P2X2/P2X4 differences responsible for these vastly different antagonist properties.

It is at first sight indeed surprising that the investigated anthraquinone derivatives are not more potent at the wt P2X2 receptor, which lacks an ionic lock. We have now compared the binding sites more thoroughly and discuss the differences in more detail. In fact, the P2X2 receptor was the first P2X2 receptor subtype, for which potent, selective anthraquinone-derived inhibitors had been developed (Baqi et al. doi: 10.1021/jm1012193.). However, the structure-activity relationships appear to be quite different between P2X2 and P2X4 receptors due to differences in their binding sites. The following discussion has now been added:

“Cibacron Blue is only weakly active at the wt P2X2 receptor although the receptor lacks an ionic lock. This is likely due to the fact that the residues directly involved in ligand binding are not all conserved in the corresponding pocket of the human P2X2 receptor and there are numerous smaller structural differences. In light of our data, the most obvious difference is the replacement of the very important Arg82 in the P2X4 receptor by the corresponding Lys91 in the P2X2 receptor. The Lys side chain is shorter and cannot form the cation- π interaction observed for Arg82 with rings B and C of PSB-0704’s anthraquinone ring system. It should be pointed out that anthraquinone derivatives with relatively high affinity and selectivity for the P2X2 receptor, but not for other P2X receptor subtypes, have previously been developed⁷⁹ indicating that the P2X2 binding site can in fact well accommodate certain anthraquinone derivatives depending on the substitution pattern.”

4. As for the extent to which structures of targets can be used for medicinal chemistry development: I certainly share the authors’ opinion on the benefits and insights provided by these structures for ligand design. My comment in this regard was more to the specific extent that the authors use these structures. I am certainly aware that ligand binding can break “ionic

locks” in receptors and in this way influence receptor signaling, and I also agree with the authors that the insight into the ionic lock in P2X4 for modulator binding to this site is interesting and good to have. It is when the authors repeat their proposal of a solution for this “ionic lock” that I become skeptical (revised manuscript; lines 655-659). I think that is very speculative that the introduction of a basic moiety in the modulator to interact with E307 would break the ionic lock, at least in a way where the modulator could bind to the site. If the ionic lock prohibits the modulator to access the binding site, this basic moiety would have to interact with E307 from the outside and next allow for the modulator to enter the binding site without this ionic lock being reformed at any point? I think it sound very speculative.

Answer:

We agree that our suggestion is highly speculative, although not entirely impossible, binding being a dynamic process. We have now changed the wording to better express how difficult it will be to design the desired compounds:

“Since the hypothesis of ionic lock formation has been confirmed, this offers the possibility to design compounds that disrupt the intermolecular ionic lock by interacting with Glu307, to open up a larger, high-affinity binding site for antagonists. Introduction of a basic moiety into the structure of PSB-0704, e.g. attached to the anthraquinone 1-position instead of or linked via the anilinic amino group, might be a plausible strategy. However, the design of an antagonist that could break an ionic lock is certainly most challenging, even though disruption of ionic locks is a common mechanism in drug-receptor interactions. For example, in G-protein-coupled receptors ionic locks stabilize the inactive state of the receptor^{72, 73}. They typically act as molecular switches, which can be disrupted by drugs⁷⁴.”

Conclusions was slightly modified, including additional citations:

“In conclusion, we discovered an allosteric binding site for anthraquinone derivatives in the human P2X4 receptor, along with important interactions. The binding site is adjacent to, but not identical with a recently described allosteric P2X4 receptor binding site^{38,51}. This could have relevance beyond the P2X4 receptor subtype since some anthraquinones, e.g. Cibacron Blue, have been found to also interact with several other P2X receptor subtypes^{31, 32,79}. The presented results, along with recent literature data, provide important insights into the interaction of anthraquinone derivatives with the human P2X4 receptor. Our findings constitute a basis that could support future drug development efforts.”

Reviewer #2 (Remarks to the Author):

I thank the authors for their thorough revision of the manuscript, and for addressing all my comments.

Reviewer #3 (Remarks to the Author):

Thank you for your thorough revision. The changes you have implemented are clear, well-considered, and improve the overall quality and clarity of the manuscript. I am satisfied with the responses provided and find no further issues that require attention.

We thank both reviewers for their considerate comments and kind assessment of the manuscript revision.

Additional

A few typos, missing spaces, and small additions and corrections were made.

The data availability statement was shifted to the end of the Methods section.

We provided **information on source data** and added the sentence: *“Source data are provided with this paper.”* Instead of: *“The datasets generated and analyzed during the current study are available from the corresponding author on reasonable request.”*

We hope that all questions have been answered to your satisfaction and that the manuscript is now suitable for publication.

With kind regards,

Christa Müller